# MIF is a 3′ flap nuclease that facilitates DNA replication and promotes tumor growth

Yijie Wang [1], Yan Chen[1], Chenliang Wang[1], Mingming Yang[1], Yanan Wang[1], Lei Bao[1], Jennifer E. Wang[1], BongWoo Kim[1], Kara Y. Chan[2], Weizhi Xu[2], Emanuela Capota[3], Janice Ortega [2], Deepak Nijhawan[3], Guo-Min Li [2], Weibo Luo [1,4] & Yingfei Wang [1,5 ✉]

How cancer cells cope with high levels of replication stress during rapid proliferation is currently unclear. Here, we show that macrophage migration inhibitory factor (MIF) is a 3′ flap nuclease that translocates to the nucleus in S phase. Poly(ADP-ribose) polymerase 1 co-localizes with MIF to the DNA replication fork, where MIF nuclease activity is required to resolve replication stress and facilitates tumor growth. MIF loss in cancer cells leads to mutation frequency increases, cell cycle delays and DNA synthesis and cell growth inhibition, which can be rescued by restoring MIF, but not nuclease-deficient MIF mutant. MIF is significantly upregulated in breast tumors and correlates with poor overall survival in patients. We propose that MIF is a unique 3′ nuclease, excises flaps at the immediate 3′ end during DNA synthesis and favors cancer cells evading replication stress-induced threat for their growth.

[1] Department of Pathology, UT Southwestern Medical Center, Dallas, TX, USA. [2] Department of Radiation Oncology, UT Southwestern Medical Center, Dallas, TX, USA. [3] Department of Internal Medicine, UT Southwestern Medical Center, Dallas, TX, USA. [4] Department of Pharmacology, UT Southwestern Medical Center, Dallas, TX, USA. [5] Department of Neurology, UT Southwestern Medical Center, Dallas, TX, USA. ✉email: Yingfei.Wang@UTsouthwestern. edu

D NA replication is a central event for cell proliferation and malfunction of the DNA replication machinery causes DNA replication stress. While DNA replication stress-induced genomic instability in normal cells has been thought as a key driver of tumorigenesis[1–4], the continuous abnormal proliferation-induced DNA replication stress may cause profound genomic instability in cancer cells and bring a threat to cancer cells viability[5,6]. However, cancer cells have evolved in certain ways to cope with DNA replication stress for their survival. Understanding and targeting these intrinsic DNA replication stress-resolving mechanisms in cancer cells may lead to developing a promising strategy to eliminate tumors.

The accuracy of DNA replication in eukaryotes requires the faithful DNA polymerases to copy DNA and also involves a timely precise DNA proofreading process to correct the replication errors. DNA polymerases (Pol) α, δ, and ε are three key polymerases contributing to DNA replication in eukaryotes[2,7,8]. Pol α does not have a proofreading function[9–12], but it is required for initiating DNA synthesis, which is subsequently carried over by Pol δ and Pol ε[9–12]. Although Pol δ and Pol ε are able to excise the mis-incorporated nucleotides at the 3′ end with their 3′ exonuclease activity to proofread replication errors, germline and somatic mutations within the exonuclease domain in human POLD1 (encoding Pol δ) and POLE (encoding Pol ε) have been identified in human cancers[1,13,14]. It is not yet completely understood how the DNA proofreading process is controlled in cancer cells while DNA polymerases keep incorporating nucleotides at such an amazingly high speed[2,5,6]. Furthermore, there is a clear discrepancy of polymerase-mediated nucleotide misincorporation rate between in vitro and in vivo studies. The nucleotide misincorporation rate of Pol δ and Pol ε in vivo is about $1/10^8$–$10^{10}$, which is much lower than their in vitro rate ($1/10^4$–$10^5$)[2]. Mismatch repair has been counted as one of the main contributors to correct in vivo replication errors that escape proofreading[2]. However, the efficiency of mismatch repair varies at different positions in the genome[2]. It is not known if additional proofreading mechanisms may be involved in replication to correct replication errors in vivo.

DNA replication produces both 5′ flap and 3′ flap DNA overhang structures, which are detrimental to cell proliferation. Resolving 3′ flap and 5′ flap DNA structures is equally important. A group of 5′ flap structure-specific nucleases, including Flap endonuclease 1 (FEN1) and DNA replication helicase/nuclease 2 (DNA2), have been implicated in the process of DNA incision to remove the 5′ flap structure in a 5′->3′ direction during DNA replication[15–18]. Pol δ and Pol ε have been well recognized for their functions in the removal of mis-incorporated nucleotides in a 3′->5′ direction. However, it is unknown if additional 3′ nucleases are required to cooperate with nuclease-proficient Pol δ and Pol ε to maintain high-speed elongation, or to proofread DNA elongated by Pol α or other 3′->5′ exonuclease-deficient polymerases-like translesion DNA polymerases or mutant Pol δ and Pol ε identified in certain human cancers, including colorectal cancer, breast cancer, and glioblastoma[14].

Our recent study identified microphage migration inhibitory factor (MIF), a previously known pleiotropic cytokine-like protein highly conserved in mammals, as a novel poly(ADP-ribose) polymerase 1 (PARP1)-associated nuclease (PAAN), which possesses a $Mg^{2+}$- and $Ca^{2+}$-dependent 3′ exonuclease activity[19]. MIF's 3′ exonuclease activity is involved in DNA fragmentation in ischemic stroke[19]. MIF is widely expressed in various cell types, including cancer cells, neurons, monocytes, macrophages, vascular smooth muscle cells, and cardiomyocytes, and plays an important role in inflammation, immune response, and tumor growth[20,21]. A large body of evidence has implicated a strong association of MIF in human cancers[22,23]. MIF is upregulated in human tumors and promotes tumor growth, which is independent of its tautomerase activity[22–24]. However, the role of MIF's nuclease activity in DNA replication and tumor growth remains completely unknown.

In this study, we identified MIF as a hitherto unrecognized 3′ flap nuclease involved in DNA replication. MIF recognizes and cleaves the 3′ flap at Y-shaped dsDNA. Genetic deletion of MIF or inhibition of its nuclease activity in cancer cells significantly increases mutation frequency, reduces DNA synthesis, causes cell cycle delay, inhibits colony survival, and attenuates the growth of breast tumors, glioblastoma, and colon tumors in mice. These findings uncover a possible mechanism of MIF-mediated tumor growth by removing the 3′ flap structures during DNA replication.

## Results

**MIF recognizes and cleaves mismatched nucleotides at the 3′ end of Y-shaped dsDNA.** To study whether MIF excises mis-incorporated nucleotides at the 3′ end of Y-shaped dsDNA, which mimics the intermediate products during DNA replication, we designed a series of "damaged" dsDNA substrates based on its stem-loop (SL) ssDNA substrate identified recently[19] but removed the loop and altered the length of mismatched nucleotides at the 3′ end with or without biotin labeling for in vitro nuclease assay (Fig. 1a). We found that MIF cleaved away unpaired nucleotides varying from 1 to 7 nt at the 3′ end of non-biotin-labeled dsDNA substrates, which had the Y-shaped structure but no loop structure (Fig. 1b, c). However, MIF could not excise the dsDNA substrate without unpaired nucleotides at the 3′ end (DS-0 nt, Fig. 1b, c). Using 3′-biotin-labeled Y-shaped dsDNAs as substrates, we discovered that MIF exhibited no cleavage or rather weak cleavage ability to 3′ biotin-labeled Y-shaped DNA substrates that had 3 or longer unpaired nucleotides at the 3′ end (Fig. 1d, e), which is consistent with previous findings that biotin labeling blocks the exonuclease activity[19]. Interestingly, with inhibition of its exonuclease activity, MIF still selectively cleaved away biotin alone (DS-0b) that served as an overhang, biotin +1 nt (DS-1b), and biotin +2 nt (DS-2b) unpaired nucleotides at the 3′ end (Fig. 1d, e). Its cleavage efficiency was negatively correlated with the length of unpaired nucleotides (Fig. 1d, e). Moreover, MIF cleaved the unpaired 3′ end nucleotide regardless of its sequence as "T", "A" or "C" (Supplementary Fig. 1a). These data indicate that MIF recognizes Y-shaped dsDNA as the substrate and possesses both 3′ exonuclease activity and 3′ flap endonuclease activity to selectively cleave away the short flap at the 3′ end, which depends on the substrate structure but not sequence.

The glutamate 22 (E22) residue is required for MIF's nuclease activity towards ssDNA with the stem-loop structure[19]. To determine whether MIF's nuclease activity is required for excision of the Y-shaped dsDNA substrate, we expressed and purified wild-type (WT) MIF and E22A-MIF mutant from bacteria (Fig. 1f). The purity of MIF protein was determined by Coomassie blue staining using bovine serum albumin (BSA, > 98% purity) as control (Fig. 1f) and further confirmed by mass spectrometry. No known bacterial or human nuclease contaminants were found in the MIF preparation (Supplementary Table 1). Next, MIF proteins were incubated with the biotin-labeled Y-shaped DNA substrate (DS-1b or 0b) in in vitro nuclease assay. We found that mutation of E22 into alanine clearly blocked MIF's nuclease activity towards DS-1b or 0b substrate in vitro (Fig. 1g). These results indicate that MIF cleaves away the unpaired nucleotides from the Y-shaped dsDNA substrate through its nuclease activity.

We further studied the nuclease kinetics of MIF by incubating purified MIF protein for 1–60 min with 1 nt overhanged

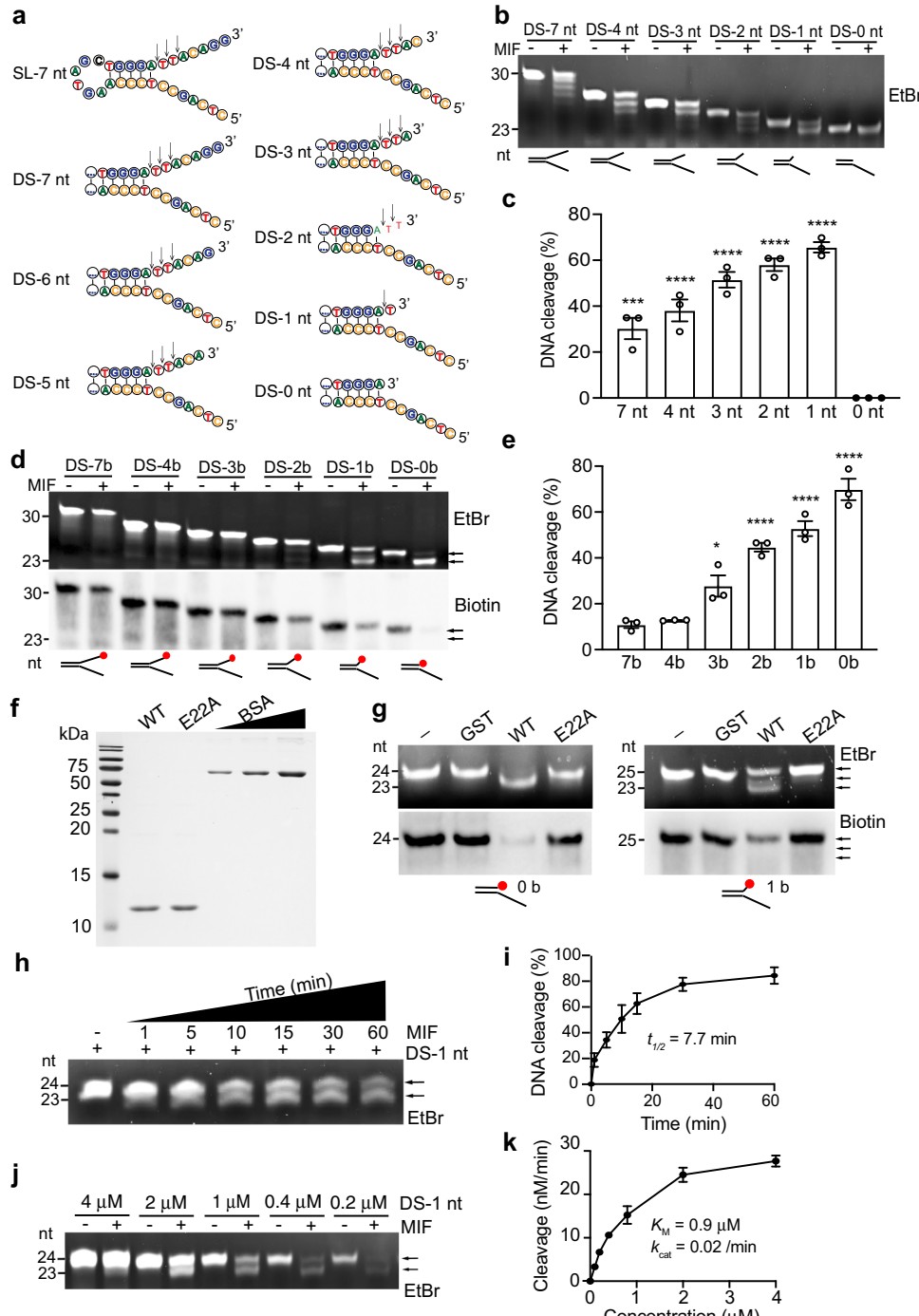

**Fig. 1 MIF recognizes Y-shaped dsDNA and cleaves 3′ unpaired nucleotides. a** Second structure of MIF substrates ssDNA with the stem-loop (SL) and Y-shaped dsDNA with different lengths of unpaired nucleotides at the 3′ end. **b**, **c** In vitro MIF (2 μM) nuclease cleavage assay using Y-shaped dsDNA as substrates. Data were quantified in **c** (mean ± SEM, $n = 3$ biologically independent experiments). ****$P < 0.0001$, ***$P < 0.001$; by one-way ANOVA Dunnett's multiple comparisons test vs 0 nt group. EtBr ethidium bromide. **d**, **e** In vitro MIF (2 μM) nuclease cleavage assay using 3′ biotin-labeled Y-shaped dsDNA as substrates. Data were quantified in **e** (mean ± SEM, $n = 3$ biologically independent experiments). ****$P < 0.0001$, *$P < 0.05$; by one-way ANOVA Dunnett's multiple comparisons test vs 7b group. Red dot indicates biotin (**d**). Arrow indicates the cleavage products. **f** Protein purification of wild-type (WT) MIF and nuclease-inactive E22A mutant shown by Coomassie blue staining. BSA (0.5–3 μg) was used as control. **g** In vitro nuclease assay of MIF and E22A mutant (2 μM) using 3′ biotin-labeled Y-shaped dsDNA (0.8 μM) as substrates. **h**, **i** MIF (2 μM) cleaves DS-1 nt substrate (0.8 μM) in a time (1, 5, 10, 15, 30, 60 min)-dependent manner. Representative image is shown in **h**. Data are quantified in **i** (mean ± SEM, $n = 3$ biologically independent experiments). **j**, **k** MIF (2 μM) cleaves DS-1 nt substrate in a concentration (0.2, 0.4, 1, 2, 4 μM)-dependent manner. Representative image is shown in **j**. Data are quantified in **k** (mean ± SEM, $n = 3$ biologically independent experiments).

Y-shaped dsDNA at concentrations varying from 0.2 to 4 μM. We found that MIF cleaved away 1 nt overhang at the 3' end of Y-shaped dsDNA (DS-1 nt) in a time-dependent manner with a $t_{1/2}$ of 7.7 min, and also in a concentration-dependent manner with an affinity for the substrate ($K_M$) of 0.9 μM and the catalytic efficiency ($k_{cat}$) of 0.02/min (Fig. 1h–k). Biotin-labeled Y-shaped dsDNA (DS-1b) was similarly tested. MIF cleaved DS-1b in a time- and concentration-dependent manner with a $t_{1/2}$ of 10.8 min, $K_M$ of 0.7 μM and $k_{cat}$ of 0.02/min (Supplementary Fig. 1b–e).

Collectively, these findings reveal that MIF has the known 3' exonuclease activity as well as a novel 3' flap endonuclease activity, which contributes to the excision of unpaired nucleotides at the 3' end of Y-shaped dsDNA in vitro.

**MIF nuclease activity is essential for nuclease-deficient polymerase-mediated DNA elongation and fidelity in vitro.** Y-shaped dsDNA mimics the replication fork during DNA replication. To directly test whether MIF plays a role in DNA proofreading during DNA replication, we next used three different nuclease-deficient polymerases as tools and performed in vitro DNA elongation assay with a 99-nt DNA as the template and a 5' biotin-labeled oligo containing 18 nt completely complementary to the DNA template as the primer #1 (Fig. 2a). For comparison, we also used three other primers (#2-4) containing 18 nt complementary to the DNA template and additional 1–3 unpaired nucleotides at the 3' end, respectively (Fig. 2a). The main eukaryotic DNA Pol δ and its proofreading-deficient mutant D402A (Pol δ-M)[25,26] were applied to elongate DNA in the presence or absence of MIF. We found that both Pol δ and its D402A mutant successfully completed the elongation and produced a 99-nt biotin-labeled DNA product regardless of the presence or absence of MIF when the primer #1 completely complementary to the DNA template was used (Fig. 2b). In contrast, when the primer (#2-4) contains 1–3 unpaired nucleotides at the 3' end, D402A Pol δ failed to complete DNA elongation to yield a 99-nt DNA product as Pol δ did (Fig. 2b), which was consistent with the previous observation that mismatched nucleotides significantly block Pol α and other nuclease-deficient polymerase-mediated DNA elongation[9–12]. Strikingly, in the presence of MIF, D402A Pol δ was able to produce a 99-nt DNA product as shown by both ethidium bromide (EtBr) staining and biotin staining (Fig. 2b), suggesting that MIF facilitates DNA elongation by removing the 3' unpaired nucleotides (Fig. 1). MIF itself did not possess the polymerase activity (Fig. 2b, lane 8). The amount of elongation product was gradually decreased along with the increased length of unpaired nucleotides at the 3' end (Fig. 2b, c), which was correlated with the cleavage efficacy of MIF on dsDNA with the different length of unpaired nucleotides at the 3' end (Fig. 1b, c). Pol α, which is another key eukaryotic DNA polymerase involved in DNA replication but lacks a 3'->5' nuclease activity[9–12], and Taq DNA polymerase without a 3' nuclease activity was also selected to elongate DNA in the presence or absence of MIF. Similar results were observed with these two 3' nuclease-deficient DNA polymerases (Supplementary Fig. 2a, b). Proliferating cell nuclear antigen (PCNA), which has been shown previously to enhance the nucleotide incorporation rate[27], did not obviously alter the effect of MIF on DNA elongation (Supplementary Fig. 2c).

Next, we studied whether the nuclease activity of MIF is required for DNA elongation using a nuclease-deficient E22A MIF mutant. Successful DNA elongation with primer #1 was observed in the presence of WT MIF or E22A MIF mutant (Fig. 2d and Supplementary Fig. 2d). However, unlike WT MIF, E22A MIF failed to ensure the success of D402A Pol δ mutant- or

Pol α-mediated DNA elongation with #3 and #4 primers, which contain 2–3 unpaired nucleotides at the 3' end (Fig. 2d and Supplementary Fig. 2e). The protein purity of POLA1 (Pol α) and POLD (Pol δ) was confirmed by Coomassie blue staining (Supplementary Fig. 2f, g). Together, these data indicate that MIF is able to cleave away mismatched nucleotides at the 3' end of dsDNA and coordinates with low-fidelity DNA polymerases to ensure the DNA elongation process.

To further determine the effect of MIF on the efficiency and fidelity of DNA synthesis, we performed M13mp18 DNA gap-filling and mutagenesis assay (Fig. 2e), as previously described[28]. The gapped M13mp18 substrate was generated by hybridizing ssM13mp18 with dsM13mp18 that had removed the LacZα-coding sequence by restriction digestions with Pst1 and Bsu36I[28]. The 283-nt gap was filled by Pol δ, D402A/L606M Pol δ mutant (Pol δ-DM) or 3' nuclease-deficient Taq DNA polymerase at 37 °C for 1 h in the absence or presence of MIF-WT (2 μM), nuclease-inactive MIF-E22A (2 μM), or negative control glutathione $S$-transferase (GST). The mutation frequency was determined by a blue-white screen in the presence of IPTG (1 mM) and X-gal (250 μg/ml) and all mutant M13mp18 DNAs were verified by Sanger DNA sequencing. Consistent with the previous report[25,26], D402A/L606M Pol δ mutant, which has a reduced 3' exonuclease activity but an increased polymerase activity, significantly increased the mutation frequency of the elongation products (76 mutants/$10^4$) in comparison with Pol δ (9 mutants/$10^4$) (Fig. 2f, g). However, in the presence of MIF-WT, the mutation frequency of D402A/L606M Pol δ mutant-mediated gap-filling assay was significantly reduced to 17 mutants/$10^4$, which was comparable to that of Pol δ. In contrast, nuclease-inactive E22A mutant and GST did not obviously alter the mutation frequency of D402A/L606M Pol δ mutant-mediated gap-filling assay (Fig. 2f, g). Moreover, Sanger sequencing analysis of mutant clones showed that point mutations were dominant in these nuclease-deficient polymerase groups and another portion of random mutations including insertion and deletion were also observed, whereas Pol δ only caused a very low frequency of single-point mutations (Fig. 2h). MIF-WT significantly reduced the point mutation frequency caused by D402A/L606M Pol δ mutant, although its mutation types were still more than those by Pol δ alone (Fig. 2h). We also examined the effect of MIF-WT on Pol δ-mediated mutagenesis and found that MIF did not further reduce the mutation frequency induced by Pol δ (Fig. 2f–h).

We next performed a gap-filling assay with another Taq DNA polymerase, which lacks 3'->5' nuclease activity. The mutation frequency in Taq + MIF-WT group was about 7 mutants/$10^4$ clones, which was much lower than those (14–22 mutants/$10^4$ clones) observed in the nuclease-inactive E22A mutant, GST, or Taq alone experimental group (Supplementary Fig. 2h). Moreover, point mutations were enriched in these three groups, whereas only several clones with a single G deletion were found in the MIF-WT group (Supplementary Fig. 2i). These data indicate that MIF is able to improve the efficiency and fidelity of nuclease-deficient polymerase-mediated DNA synthesis.

**MIF facilitates the DNA replication process.** To investigate whether MIF plays a role in DNA replication process at the cellular level, we established two MIF knockout (KO) MDA-MB-231 cell lines using CRISPR/Cas9 single-guide RNAs (sgRNAs) targeting two different regions of the *MIF* gene (Supplementary Fig. 3a) and the rescued cell lines by transducing MIF KO cells with lentivirus carrying Flag-tagged WT or nuclease-inactive E22A MIF mutant (Fig. 3a). Cell cycle progression was analyzed after 0–9 h-release from double thymidine-induced G1/S synchronization. Both MIF KO1 and

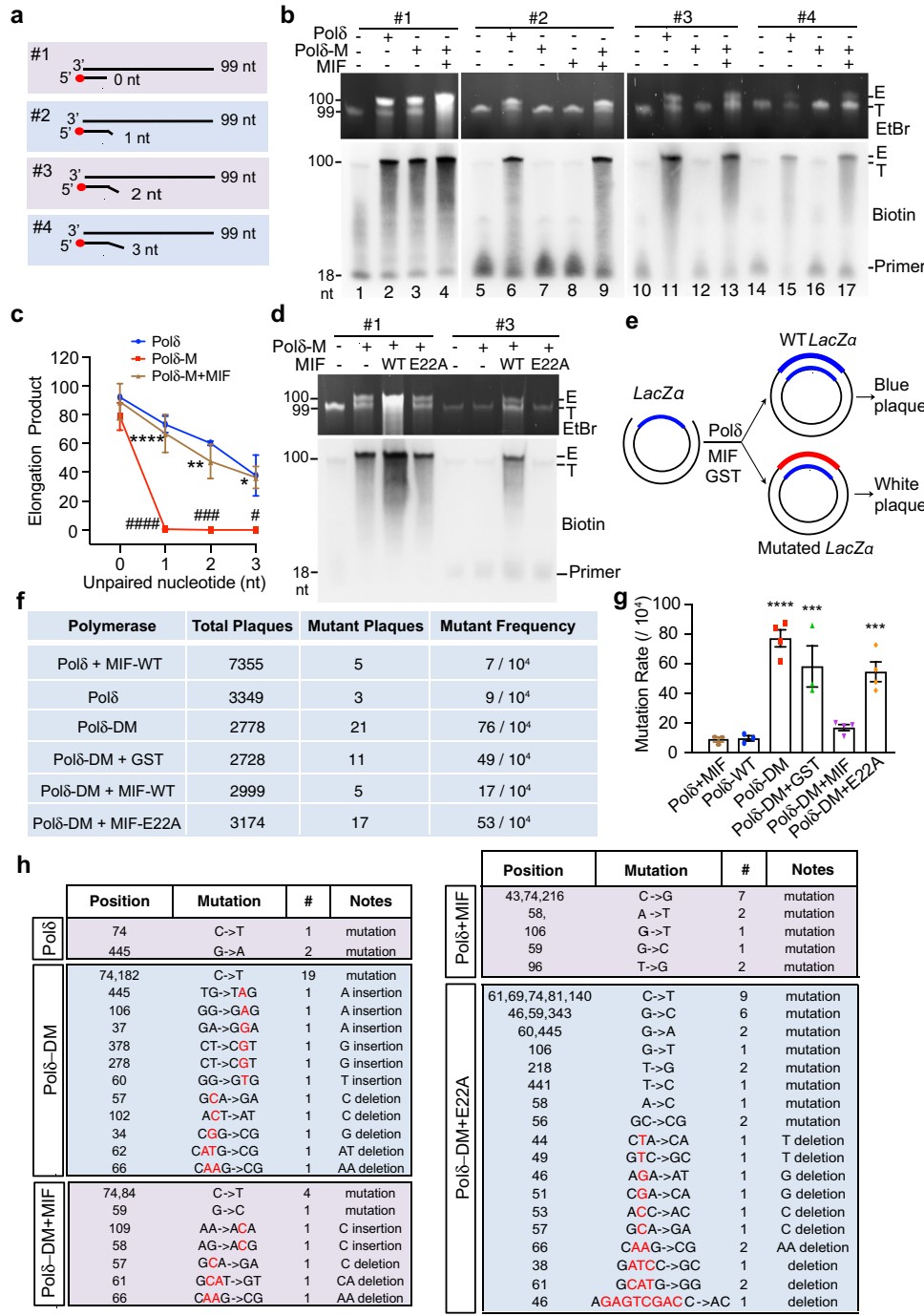

**Fig. 2 MIF coordinates with the nuclease-deficient Pol δ D402A mutant to facilitate DNA elongation. a** Scheme of four types of 5′ biotin-labeled short DNA primers complementary to the template DNA without or with 1–3 nt mismatched nucleotides at the 3′ end for in vitro DNA elongation assay. **b** In vitro DNA elongation assay mediated by Pol δ or 3′ nuclease-deficient Pol δ D402A mutant (Pol δ-M) (5 ng/μl) in the presence or absence of MIF (2 μM) using a 99-bp DNA template (0.4 μM) and short primers (0.4 μM) listed in **a**. DNA was visualized by EtBr staining and Biotin immunoblot. E represents the elongation product. T indicates the complex of the 99-bp DNA template and biotin-labeled primer. **c** Quantification of in vitro DNA elongation assay in **b** (mean ± SEM, *n* = 3 biologically independent experiments). ####*P* < 0.0001, ###*P* < 0.001, #*P* < 0.05 Pol δ-M vs Pol δ; *****P* < 0.0001, ***P* < 0.01, **P* < 0.05 Pol δ-M + MIF vs Pol δ-M; by two-way ANOVA Tukey's multiple comparisons. **d** Effects of WT and E22A MIF (2 μM) on Pol δ-M-mediated DNA elongation using a 99-bp DNA template and #1 or #3 primer (0.4 μM). DNA was visualized by EtBr staining or biotin immunoblot. **e** Schematic diagram of DNA gap-filling synthesis by Pol δ or Pol δ D402A/L606M mutant (Pol δ-DM) (5 ng/μl) in the presence or absence of GST or MIF proteins (2 μM) and mutagenesis screening. **f–h** Mutation frequency and mutation sequence analysis of gap-filling assay mediated by Pol δ or Pol δ-DM in the presence of GST, WT MIF, or E22A MIF proteins. Mutation rate was quantified in **g** and presented as mean ± SEM. *n* = 3 biological replicates for Pol δ + MIF, Pol δ-WT and Pol δ-DM + GST groups; *n* = 4 biological replicates for Pol δ-DM, Pol δ-DM + MIF, and Pol δ-DM + E22A groups. *****P* < 0.0001, ***P* < 0.001 vs Pol δ-WT, by one-way ANOVA Dunnett's multiple comparisons test. Source data are provided as Source Data file.

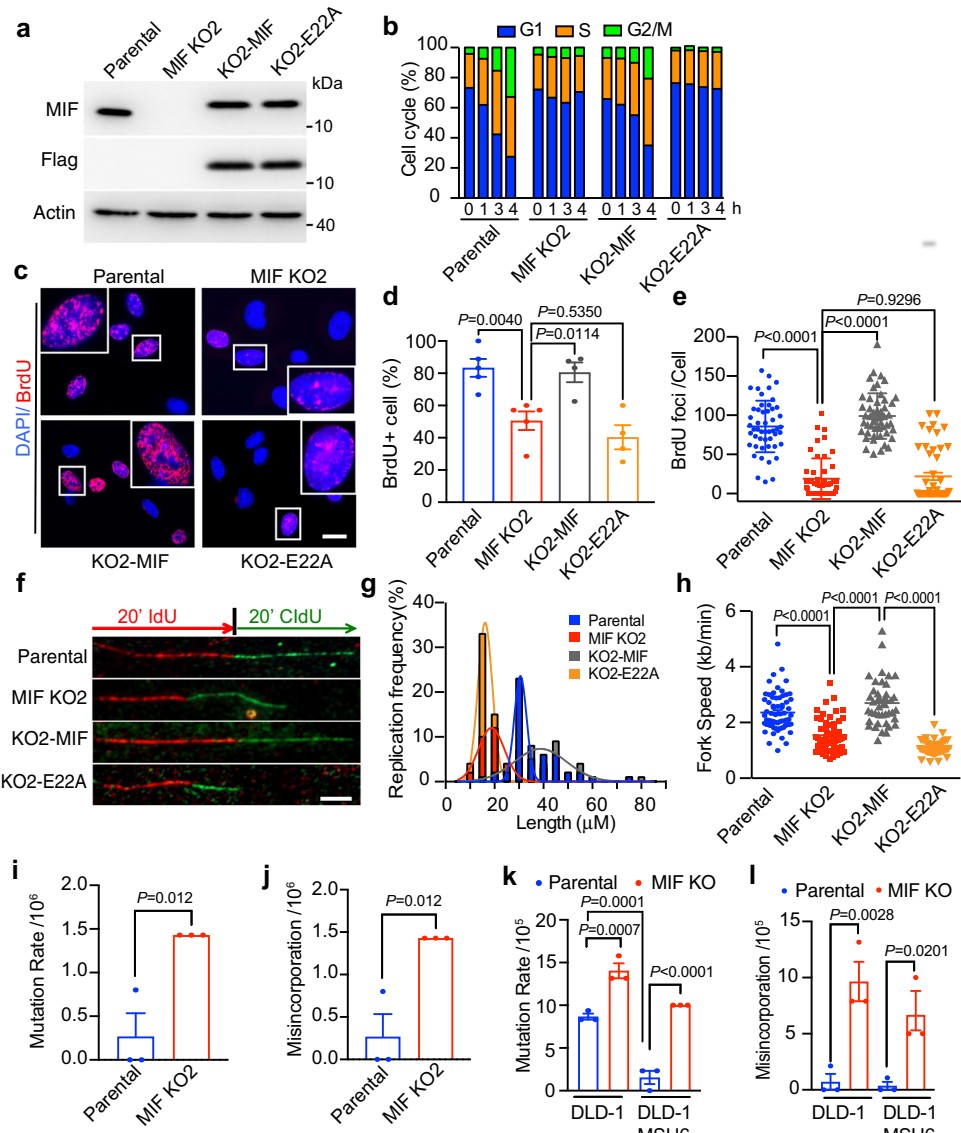

**Fig. 3 MIF and its nuclease activity are required for DNA replication and DNA synthesis in cancer cells. a** Expression of endogenous MIF and exogenous Flag-tagged WT or E22A MIF proteins in parental and MIF knockout KO MDA-MB-231 cells. **b** Cell cycle distribution analysis by flow cytometry in parental, MIF-KO2, KO2-MIF, and KO2-E22A MDA-MB-231 cells after double thymidine (2 mM) synchronization followed by 1, 3, and 4 h fresh medium incubation. **c** Representative images of BrdU (red) and DAPI (blue) staining in parental, MIF-KO2, KO2-MIF, and KO2-E22A MDA-MB-231 cells after double thymidine (2 mM) synchronization followed by 2 h fresh medium incubation. Scale bar, 20 μm. **d, e** Quantification of BrdU-positive cells (**d**) and BrdU foci numbers per cell (**e**). $n = 50$ cells in each group were counted over three biologically independent experiments. Foci numbers >20 were counted as BrdU-positive cells. Statistical significance was determined by one-way ANOVA Dunnett's multiple comparisons test. **f** Representative images of stretched DNA fibers in parental, MIF-KO2, KO2-MIF, and KO2-E22A MDA-MB-231 cells after IdU (25 μM, 20 min) and CIdU (250 μM, 20 min) incubation. Scale bar, 10 μm. **g, h** Distribution of the track lengths of CIdU and fork progression are indicated in different groups. $n = 66$ fibers in the parental group; $n = 56$ fibers in MIF-KO2 group; $n = 42$ fibers in the KO2-MIF group and $n = 54$ fibers in KO2-E22A group were analyzed from three independent experiments. Statistical significance was determined by one-way ANOVA Dunnett's multiple comparisons test. **i, j** Mutation frequency analysis including deletion, insertion, and point mutations (**i**) and misincorporation rate analysis (**j**) in MIF-WT and MIF KO MDA-MB-231 cells using HPRT mutation assay (mean ± SEM, $n = 3$ biologically independent experiments). Statistical significance was determined by two-tailed Student's $t$ test. **k, l** Mutation frequency (**k**) and misincorporation rate (**l**) determined by HPRT mutation assay in parental and MIF KO DLD1 and DLD1-MSH6 rescued cells (mean ± SEM, $n = 3$ biologically independent experiments). Statistical significance was determined by one-way ANOVA Dunnett's multiple comparisons test.

KO2 cells displayed a prolonged G1 phase and delayed S phase progression as compared with parental cells (Fig. 3b and Supplementary Fig. 3b, c). Restoring WT MIF expression in MIF-KO2 cells (KO2-MIF) rescued the cell cycle progression, whereas expression of E22A mutant in MIF-KO2 cells (KO2-E22A) failed to do so (Fig. 3b). To determine whether MIF controls DNA synthesis in cancer cells, we assessed bromodeoxyuridine (BrdU) incorporation in parental, MIF KO and its

rescued cells in the S phase. Both MIF KO1 and KO2 MDA-MB-231 cells showed a robustly reduced number of BrdU foci per cell as well as a lower percentage of BrdU-positive cells as compared with parental cells (Supplementary Fig. 3d–f). Expression of WT MIF in MIF-KO2 cells restored DNA synthesis, whereas expression of E22A MIF in MIF-KO2 cells had no detectable effects (Fig. 3c–e). Similar results were also observed in human colon cancer HCT116 cells that are defective

in mismatch repair (MMR) due to biallelic deletion in the *MLH1* gene (Supplementary Fig. 3g, h).

DNA fiber assay is a valuable tool to visualize and follow the spatial and temporal progression of DNA replication fork[29]. We labeled MDA-MB-231 cells with IdU (25 μM, 20 min) and CldU (250 μM, 20 min) to monitor the individual fork progression and found that the total tracks of IdU and CldU were much shorter in MIF-KO2 cells (18.7 μm) than parental cells (30.6 μm), indicating a slower speed of DNA synthesis (Fig. 3f–h). Expression of WT MIF, but not E22A MIF, restored DNA synthesis speed, which was comparable to that in parental cells (Fig. 3f–h). Taken together, our data indicate that MIF and its nuclease activity play a critical role in regulating DNA replication speed.

To study whether MIF KO cells are more sensitive to DNA replication stress, hydroxyurea (Hu, 2 mM for 2 h) was applied to parental and MIF KO MDA-MD-231 cells. As expected, hydroxyurea treatment decreased the replication frequency and length (21.2 μm) in parental cells, which were further reduced in MIF KO cells (10.6 μm, Supplementary Fig. 3i, j). The stalled replication fork number was significantly increased in MIF KO cells (Supplementary Fig. 3k), which was reversed by the expression of WT but not E22A MIF (Supplementary Fig. 3i–k). In line with this observation, the colony survival of MIF KO cells was significantly reduced in response to hydroxyurea varying at 25–100 μM as compared with those of parental cells, which was rescued by the expression of WT but not E22A MIF (Supplementary Fig. 3l). These data indicate that MIF-deficient cells are more sensitive to DNA replication stress.

The hypoxanthine-guanine phosphoribosyltransferase (*HPRT*) gene on the X chromosome has been used as a model gene to investigate mutability in mammalian cell lines[30,31]. To further explore the effect of MIF on replication fidelity, we established POLD1 knockdown (KD) and POLD1 KD/MIF KO MDA-MB-231 cells using CRISPR/Cas9 sgRNAs (Supplementary Fig. 3m) and performed HPRT mutation assay following 6-TG treatment (20 μM, 12 days). We found that the HPRT mutation rate was very low ($2.6 \times 10^{-7}$) in Pol δ-proficient MDA-MB-231 cells (Fig. 3i), whereas MIF KO increased the HPRT mutation rate ($1.43 \times 10^{-6}$), which was mainly due to the nucleotide misincorporation (Fig. 3i, j). Since POLD1 KD MDA-MB-231 cells were not able to form colonies for HPRT assay (Supplementary Fig. 3n), we instead established MIF KO in colorectal cancer cell line DLD1 cells, which have Pol δ mutations R689W and R506H in the conserved DNA polymerase III and exonuclease III motifs, respectively[32]. The overall mutation frequency in Pol δ-deficient DLD1 cells was about $8.68 \times 10^{-5}$, which was increased by 1.6-fold in MIF KO cells ($1.4 \times 10^{-4}$, Fig. 3k). It is of note that the nucleotide misincorporation rate in parental DLD1 cells was quite low ($7.1 \times 10^{-6}$) and MIF KO significantly increased the misincorporation rate by 13.6-fold ($9.65 \times 10^{-5}$, Fig. 3l). MIF KO DLD1 cells expressing WT MSH6 (DLD1 + MSH6) were also established, as DLD1 cells are deficient in MMR due to the mutation in MSH6 (Supplementary Fig. 3o, p). Restoring MSH6 expression in DLD1 cells reduced the overall mutation rate to $1.55 \times 10^{-5}$, as expected. However, MIF KO still significantly increased the overall mutation frequency ($1.0 \times 10^{-4}$, Fig. 3k). Notably, the misincorporation rate in the MMR-proficient DLD1 cells was only about $3.5 \times 10^{-6}$, which was increased by 19-fold in MIF KO cells ($6.67 \times 10^{-5}$, Fig. 3l). These findings indicate that MIF has a significant impact on correcting nucleotide misincorporation in vivo.

**MIF is recruited to the nucleus and locates at the DNA replication site in the S phase.** MIF primarily locates in the cytosol under physiological conditions. Next, we questioned when MIF is

translocated to the DNA replication site in the nucleus. To this end, we synchronized MDA-MB-231 cells at the G1/S boundary with double thymidine block (Fig. 4a) and examined MIF nuclear translocation at different time points varying from 0 to 8 h after synchronization. We found that MIF started to be translocated into the nucleus at 1 h-release after synchronization (Fig. 4b, c). MIF's nuclear translocation further increased and peaked at 2 h-release after synchronization, whereas its nuclear translocation dramatically decreased at 8 h-release after synchronization when most cells entered into the G2/M phase (Fig. 4b, c). These data reveal that MIF is translocated into the nucleus mainly in the S phase.

Next, we investigated whether MIF is recruited to the DNA replication site after its nuclear translocation. We performed iPOND (isolation of proteins on nascent DNA) assay and found that MIF was indeed associated with newly synthesized DNA at the replication forks (Fig. 4d). However, unlike PCNA, MIF persistently bound to newly replicated chromatin even after a thymidine chase. To further study if MIF locates at the DNA replication site, we examined whether MIF physically interacts with the DNA replisome protein PCNA, which serves as a scaffold protein to recruit other proteins involved in DNA replication. By immunoprecipitation (IP) with the antibody against either MIF or PCNA, we found that MIF and PCNA reciprocally interacted with each other in MDA-MB-231 cells (Fig. 4e, f). PCNA bound to both WT and E22A MIF with a similar binding affinity (Supplementary Fig. 4a). Immunostaining assay showed that the majority of MIF located in the nucleus and colocalized with EdU and PCNA in the S phase (Fig. 4g–i). Together, these findings reveal that MIF locates at the DNA replication site in the S phase.

PCNA-interacting proteins often contain a conserved PIP box (QxxLXXFF) that binds to PCNA[33]. We analyzed the MIF amino acid sequence across multiple species and identified a PIP box-like motif (Supplementary Fig. 4b). To determine whether MIF directly binds to PCNA through the PIP box-like motif, we mutated two conserved residues L47 and F49. We found that L47A, F49Y, or double-mutant L47A/F49Y failed to disrupt MIF-PCNA interaction by co-IP (Supplementary Fig. 4c), suggesting that MIF is likely to interact with PCNA through other proteins rather than the PIP box-like motif.

**PARP1 is required for MIF recruitment to the DNA replication site.** To understand the molecular mechanism of MIF recruitment to the DNA replication site, we immunoprecipitated MIF in HeLa cells with or without DNA damage induced by the treatment of alkylating agent N-Methyl-N′-nitro-N-nitrosoguanidine (MNNG, 50 μM) for 15 min followed by additional 4 h culture and then performed mass spectrometry analysis to identify MIF-interacting proteins (Fig. 5a). MIF-KO2 cells were used as a negative control (Fig. 5a). We identified 483 MIF-interacting proteins under physiological conditions and 1282 MIF-interacting proteins following DNA damage (Fig. 5b). Among them, 386 proteins including PARP1, POLD1, PCNA, the minichromosome maintenance protein complex (MCM) 2–7, XRCC5, XRCC6, FEN1, MSH2, RPA1, and Mus81, were overlapped (Fig. 5b). The functional annotation analysis using David bioinformatics resources revealed that the majority of these overlapped proteins were involved in either rRNA/mRNA processing or DNA repair/DNA replication/cell cycle regulation (Fig. 5c). To validate our findings, we performed co-IP experiments in MDA-MB-231 cells and found that endogenous MIF interacted with endogenous PARP1, PCNA, XRCC5, and Mus81 in MDA-MB-231 cells (Fig. 4e, f and Supplementary Fig. 5a–c).

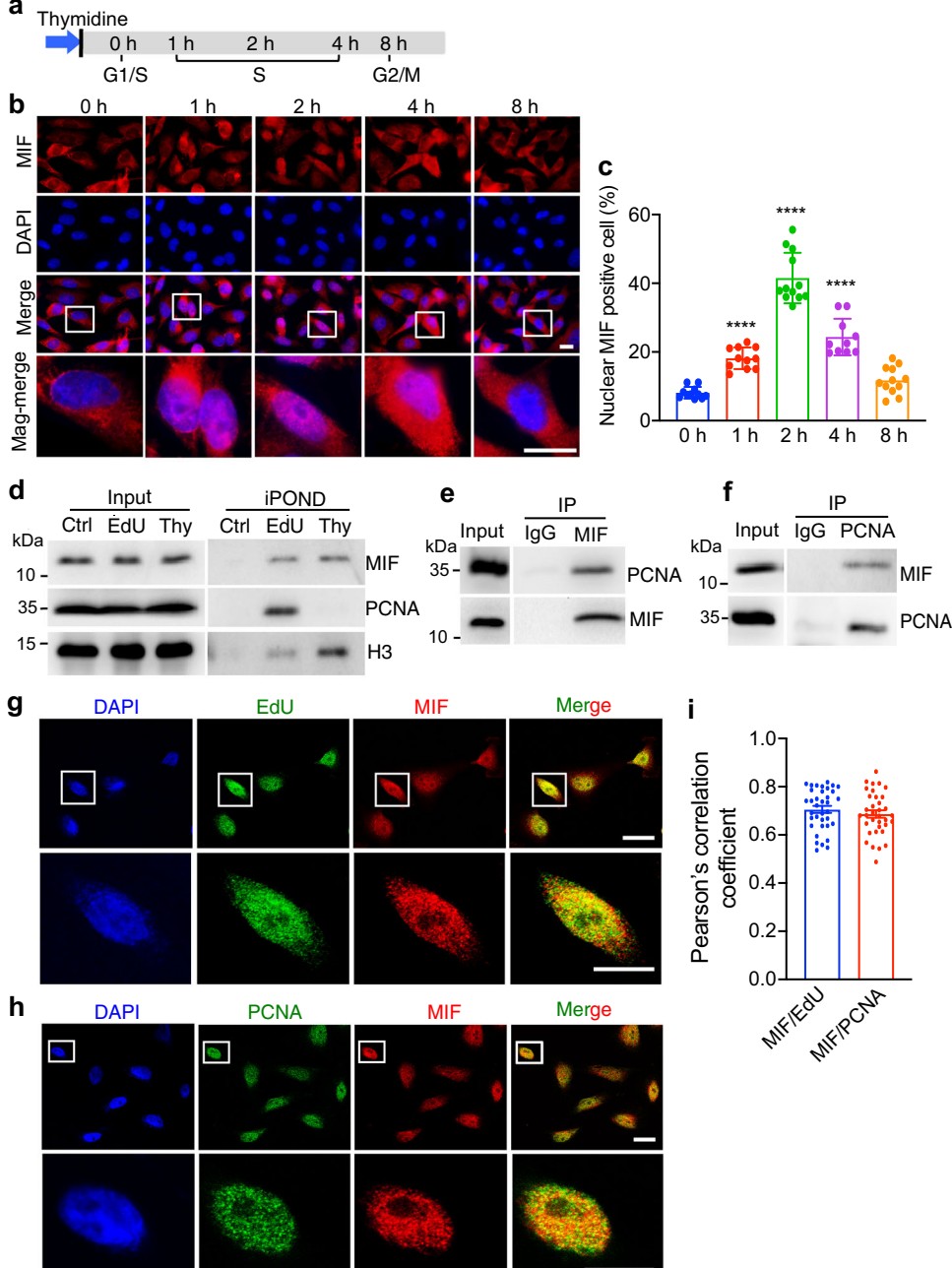

**Fig. 4 MIF is translocated to the nucleus and locates at sites of DNA replication in the S phase. a** G1/S synchronization workflow using double thymidine treatment. **b**, **c** Representative images of MIF nuclear translocation at different time points after double thymidine treatment. The percentage of cells with nuclear MIF was quantified and shown in **c** (mean ± SEM). $n = 11$ for 0 h and 1 h each group, $n = 12$ for 2 h group, $n = 10$ for 4 h group, and $n = 12$ for 8 h group were analyzed over three biological replicates. Scale bar, 20 μm. ****$P < 0.0001$ vs 0 h group, by one-way ANOVA Dunnett's multiple comparisons test. **d** MIF is recruited to the DNA replication fork determined by isolation of proteins on nascent DNA (iPOND) analysis with or without Thymidine (Thy, 10 μM) chase for 1 h after EdU treatment. Representative blots from three independent experiments are shown. **e**, **f** Reciprocal co-IP of endogenous MIF and PCNA in MDA-MB-231 cells. Representative blots from three independent experiments are shown. **g–i** Immunostaining of MIF and EdU (**g**) or PCNA (**h**) in MDA-MB-231 cells at 2 h after double thymidine synchronization. Representative images from three independent experiments are shown. Scale bar, 20 μm. Colocalization of MIF-EdU ($n = 36$ from three biologically independent experiments) and MIF-PCNA ($n = 35$ from three biologically independent experiments) were quantified as Pearson's correlation coefficient (mean ± SEM in **i**). Cells with Pearson's correlation coefficient ($r$) >0.5 is considered as the positive correlation of MIF-EDU or MIF-PCNA colocalization, $r > 0.7$ is considered as a strong correlation, and $r < 0.4$ is considered as a weak or no correlation.

PARP1 is a DNA damage sensor activated by ssDNA or dsDNA breaks. Recently, PARP1 activation was shown to be detected at DNA replication sites in the S phase[34,35]. Therefore, we hypothesized that PARP1 is activated by 3' unpaired DNA during DNA replication and then recruits MIF to the DNA

replication site. To test whether 3' unpaired DNA activates PARP1, we first performed an in vitro PARylation assay. As expected, without DNA, PARP1 was not activated even in the presence of NAD$^+$ (Supplementary Fig. 5d, e). Interestingly, dsDNA with -OH at both 3' and 5' ends activated PARP1 to

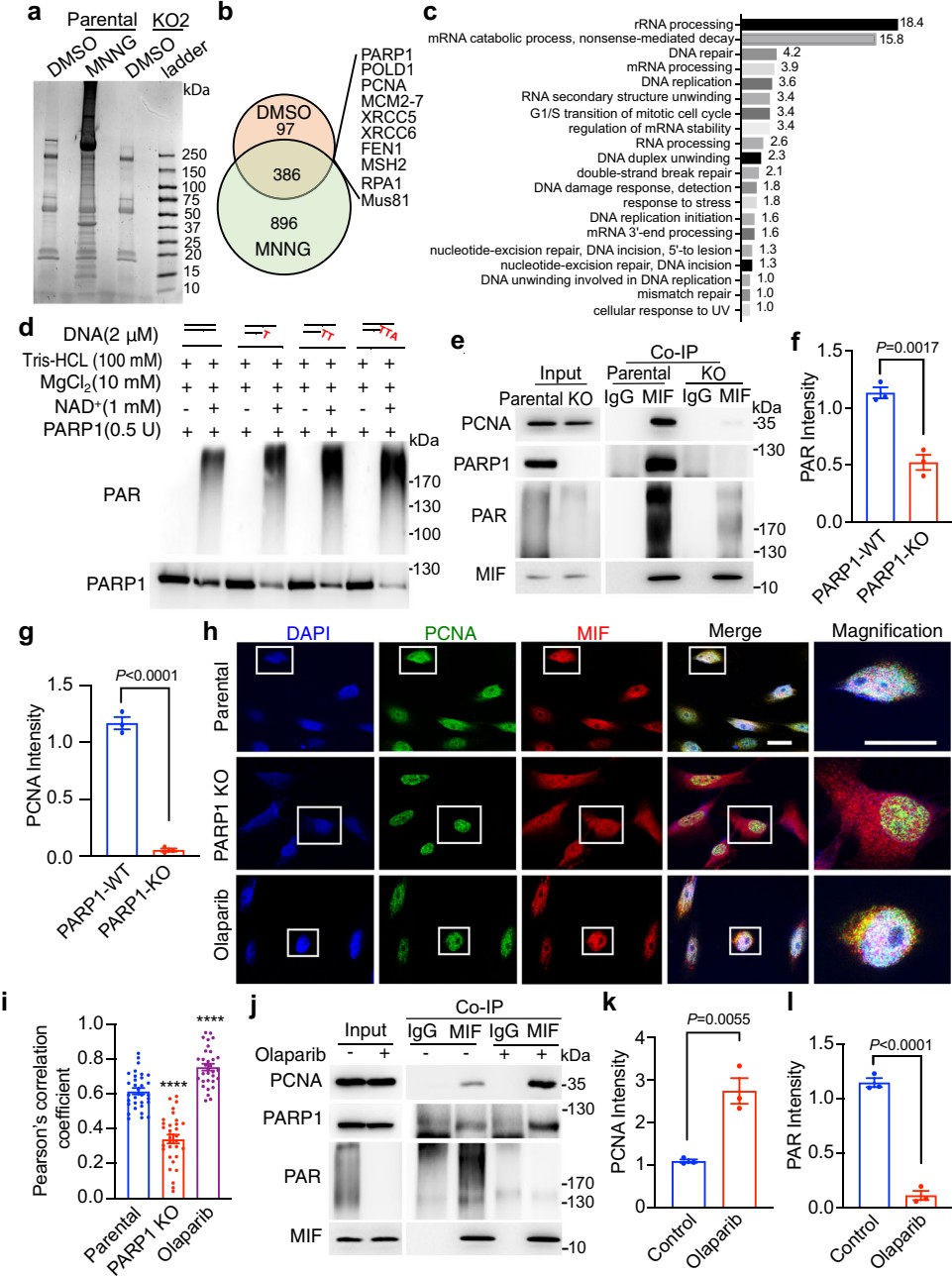

**Fig. 5 PARP1 is required for the recruitment of MIF to DNA replication sites. a** Immunoprecipitation of endogenous MIF in parental and MIF KO HeLa cells under physiological conditions (DMSO) as well as 4 h after DNA damage (MNNG, 50 μM, 15 min). **b** Mass spectrometry analysis of MIF-interacting proteins in HeLa cells under physiological conditions (DMSO) as well as 4 h after DNA damage. **c** Functional annotation of overlapped MIF-interacting proteins under both physiological and DNA damage conditions using David Bioinformatics Resources 6.8. Values indicate the % of proteins detected in the database. Pathways with $P < 0.05$ were listed. **d** In vitro PARylation assay by incubating PARP1 with dsDNA containing 0–3 mismatched nucleotides at the 3′ end in the presence of 1 mM NAD$^+$ and 10 mM MgCl$_2$. Representative blots from three independent experiments are shown. **e–g** Co-IP of MIF and PCNA in parental and PARP1 KO MDA-MB-231 cells. Relative PAR and PCNA intensities were quantified and shown in **f** and **g**, respectively (mean ± SEM, $n = 3$ biologically independent experiments). Statistical significance was determined by two-tailed Student's $t$ test. **h, i** Immunostaining of MIF and PCNA in parental, PARP1 KO and Olaparib-treated (5 μM, 2 h) MDA-MB-231 cells 2 h following double thymidine synchronization. White indicates colocalization. Representative images from three independent experiments are shown in **h**. Scale bar, 20 μm. The Pearson's correlation coefficient of MIF-PCNA colocalization was quantified and shown in **i** (mean ± SEM, $n = 32$ cells per group over three independent experiments). Cells with Pearson's correlation coefficient ($r$) >0.5 is considered as the positive correlation of MIF-PCNA colocalization, $r > 0.7$ is considered as a strong correlation and $r < 0.4$ is considered as a weak or no correlation. ****$P < 0.0001$ compared to parental cells, by one-way ANOVA Dunnett's multiple comparisons test. **j** Co-IP of MIF, PCNA, PAR, and PARP1 in MDA-MB-231 cells treated with or without Olaparib (5 μM, 2 h) following double thymidine synchronization. **k, l** Relative intensities of PCNA and PAR were quantified and shown in **k** and **l**, respectively (mean ± SEM, $n = 3$ biologically independent experiments). Statistical significance was determined by two-tailed Student's $t$ test.

produce a basal level of PAR (Fig. 5d and Supplementary Fig. 5d). dsDNA with 1–3 mismatched nucleotides at the 3' end clearly further increased PARP1 activity as evidenced by elevated levels of PAR (Fig. 5d, Supplementary Fig. 5d, e). The levels of PARP1 activation were not obviously correlated with the number of unpaired nucleotides (Supplementary Fig. 5d, e). These data indicate that PARP1 is activated by DNAs with 3' flap structure caused by mis-incorporated nucleotides.

To determine the direct effect of PARP1 on the recruitment of MIF to the DNA replication site, we generated PARP1 KO MDA-MB-231 cells using the CRISPR/Cas9 technique and compared MIF-PCNA interaction in both parental and PARP1 KO cells (Fig. 5e). The KO efficiency of PARP1 was confirmed by analysis of levels of PARP1 protein and PARP1 activation product PAR (Fig. 5e, f). Loss of PARP1 almost completely abolished the interaction of MIF and PCNA in MDA-MB-231 cells, as shown by co-IP (Fig. 5e, g). To further confirm if PARP1 directly binds to MIF, GST-MIF-bound Sepharose beads were incubated with purified PARP1, PCNA, or GST protein. We found that PARP1, but not PCNA or GST, directly bound to GST-MIF (Supplementary Fig. 5f). V5-tagged full-length (FL) PARP1 and a series of PARP1 truncates including ΔZn, ΔZn-BRCT, ΔWGR-CAT, ΔCAT were generated to systematically map MIF's binding domain on PARP1 by co-IP (Supplementary Fig. 5g). We observed that MIF mainly bound to the N-terminal zinc finger domain, as deletion of the zinc finger domain abolished PARP1-MIF interaction (Supplementary Fig. 5g, h). Immunostaining further showed that PARP1 KO blocked MIF-PCNA colocalization at the DNA replication site in MDA-MB-231 cells (Fig. 5h, i). To our surprise, the treatment of PARP inhibitor Olaparib (5 μM, 2 h) did not block MIF nuclear localization. Conversely, it even increased MIF-PCNA nuclear colocalization (Fig. 5h, i). In line with this, the treatment of Olaparib (5 μM, 2 h) following double thymidine synchronization clearly increased MIF-PCNA interaction in MDA-MB-231 cells (Fig. 5j–l).

To rule out the possibility that MIF is PARylated leading to inhibition of MIF-PCNA interaction, we performed an in vitro PARylation assay and found that MIF was not PARylated by incubating with the purified PARP1 protein in the presence of $NAD^+$ and activated DNA (Supplementary Fig. 5i). As expected, PARP1 itself was successfully PARylated. However, PARylated PARP1 was mainly in the supernatant and very little if any bound to GST-MIF-bound beads (Supplementary Fig. 5i). In line with this observation, the treatment of MNNG (50 μM, 15 min) increased PAR levels but decreased MIF binding to PARylated-proteins, which was determined by immunoprecipitation of PAR antibody (Supplementary Fig. 5j). In addition, MIF itself was not PARylated (Supplementary Fig. 5j). Taken together, these data indicate that PARP1 is required for MIF recruitment to the DNA replication site and PARylation of PARP1 suppresses MIF and PCNA interaction, thereby allowing MIF to dissociate from the DNA replication sites.

**MIF guards against DNA damage and genomic instability in cancer cells through its nuclease activity**. Our results above showed a critical role of MIF and its nuclease activity in regulation of DNA replication, we next studied the impact of loss of MIF or its nuclease activity on DNA damage and genomic stability. We found that MIF KO1 and KO2 remarkably increased the levels of DNA damage marker γH2AX in MDA-MB-231 cells (Fig. 6a). The DNA damage foci of γH2AX and 53BP1 were also significantly elevated in MIF-KO2 MDA-MB-231 cells (Fig. 6b, c and Supplementary Fig. 6a–d). Notably, expression of WT MIF, but not nuclease-inactive E22A mutant, reduced the number of DNA damage foci in MIF-KO2 MDA-MB-231 cells comparable to the

levels in parental cells (Fig. 6b, c and Supplementary Fig. 6a–d). In line with these observations in MDA-MB-231 cells, MIF KO significantly increased accumulation of γH2AX foci in LN229 cells as well as HCT116 cells (Supplementary Fig. 6e–j). This effect was reversed by re-expression of WT MIF but not nuclease-inactive E22A MIF (Supplementary Fig. 6e–j), indicating that MIF guards against DNA damage through its nuclease activity.

To further test the role of MIF in the maintenance of genomic stability, we studied the formation of micronuclei, nuclear bridge/budding as well as chromosome abnormality in parental, MIF-KO2, and rescued MDA-MB-231 cells by metaphase spread assay. MIF-KO2 cells exhibited a significant increase in abnormal micronuclei and nuclear bridge/budding (Fig. 6d–g). Moreover, loss of MIF increased chromosome aberrations, such as chromosome breaks and fusions, which could be reversed by expression of WT but not E22A MIF (Fig. 6h, i). These data indicate that MIF plays an important role in maintaining genomic stability in cancer cells.

**MIF promotes cancer cell growth in vitro and in vivo through its nuclease activity**. To determine if MIF promotes cancer cell growth, parental and MIF KO MDA-MB-231 cells were subjected to cell proliferation assay in vitro. MIF loss in Pol δ−proficient MDA-MB-231 cancer cells significantly reduced cell proliferation (Supplementary Fig. 7a). MIF KO similarly suppressed Pol δ-deficient cancer cell growth, although loss of Pol δ already significantly inhibited MDA-MB-231 cell growth (Supplementary Fig. 7a). Clonogenic assay showed that MIF KO significantly decreased colony survival of MDA-MB-231, LN229, and HCT116 cells (Supplementary Fig. 7b–d). Consistently, MIF KD by any of three independent short hairpin RNAs also reduced colony survival of HCT116 cells (Supplementary Fig. 7e–h). Notably, expression of WT MIF partially restored reduced colony formation conferred by MIF-KO2, whereas E22A MIF mutant failed to do so (Fig. 7a, b). These findings reveal that MIF promotes cancer cell survival and growth in vitro through its nuclease activity.

We next determined the role of MIF in normal cell growth. MIF was knocked out in a non-tumorigenic mammary epithelial cell line MCF-10A cells using two independent sgRNAs (Supplementary Fig. 7i) and the growth pattern was compared. We discovered that MIF KO1 or KO2 did not obviously alter MCF-10A cell proliferation, although they repressed MCF-10A colony survival (Supplementary Fig. 7j, k). Similar results were observed in mouse embryonic fibroblasts. Additional studies are required to explore if MIF is more vulnerable for growth of cancer cells than normal cells in future.

We then performed orthotopic implantation of parental and MIF KO1 and KO2 MDA-MB-231 cells into the mammary fat pad of NOD/SCID mice, respectively, to examine the effect of MIF on tumor growth in vivo. Consistent with in vitro findings above, MIF deletion robustly impaired tumor growth in mice (Supplementary Fig. 7l–o). Loss of MIF protein expression was verified in all KO tumors (Supplementary Fig. 7o). Similar results were observed in LN229 and HCT116 xenograft mouse models (Supplementary Fig. 7p–w). Reduced MDA-MB-231 tumor growth conferred by MIF-KO2 was partially rescued by overexpression of WT but not E22A MIF (Fig. 7c–e). To complement the loss-of-function studies, we conducted gain-of-function studies by implanting EV, WT MIF, and E22A MIF overexpressed MDA-MB-231 cells into the mammary fat pad of female NOD/SCID mice, respectively. Overexpression of WT MIF promoted tumor growth, whereas E22A MIF had no effect on tumor growth as compared with the EV group (Fig. 7f–i). These findings reveal that MIF promotes tumor growth in vivo through its nuclease activity.

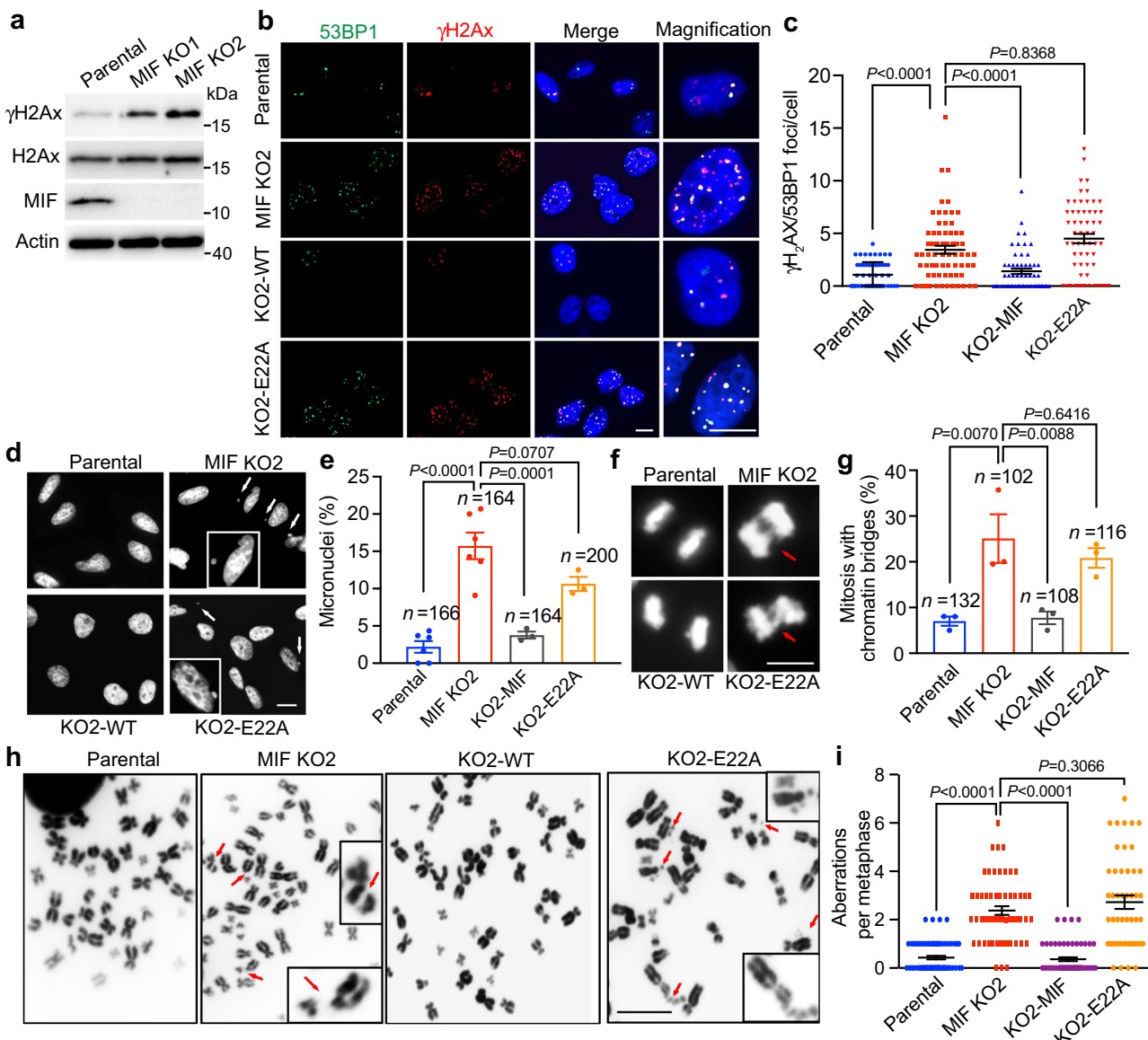

**Fig. 6 Loss of MIF and its nuclease activity caused genomic instability. a** The basal levels of γH₂AX protein in parental and MIF KO MDA-MB-231 cells. Representative blots from three independent experiments are shown. **b**, **c** Representative images of γH₂AX and 53BP1 foci in parental, MIF-KO2, KO2 + MIF-WT and KO2 + MIF-E22A MDA-MB-231 cells (**b**). Data were quantified and shown in **c** (mean ± SEM). Scale bar, 20 μm. n = 58 cells (parental), n = 67 cells (MIF-KO2), n = 57 cells (KO2-MIF), and n = 59 cells (KO2-E22A) from three independent experiments were analyzed. Statistical significance was determined by one-way ANOVA Dunnett's multiple comparisons test. **d**, **e** Representative images of micronuclei indicated by DAPI staining in parental, MIF-KO2, KO2 + MIF-WT, and KO2 + MIF-E22A MDA-MB-231 cells (**d**). Data were quantified and shown in **e** (mean ± SEM). Statistical significance was determined by one-way ANOVA Dunnett's multiple comparisons test. Scale bar, 20 μm. **f**, **g** Representative images of chromosome bridges indicated by DAPI staining in parental, MIF-KO2, KO2 + MIF-WT and KO2 + MIF-E22A MDA-MB-231 cells (**f**). Data were quantified and shown in **g** (mean ± SEM). Statistical significance was determined by one-way ANOVA Dunnett's multiple comparisons test. Scale bar, 10 μm. **h**, **i** Representative images of metaphase spread assay in parental, MIF-KO2, KO2 + MIF-WT, and KO2 + MIF-E22A MDA-MB-231 cells treated with colchicine for 2 h (**h**). Data were quantified and shown in **i** (mean ± SEM). n = 73 cells (parental); n = 56 cells (MIF-KO2); n = 58 cells (KO2 + MIF-WT) and n = 47 cells (KO2 + MIF-E22A) were analyzed from three independent experiments. Statistical significance was determined by one-way ANOVA Dunnett's multiple comparisons test. Scale bar, 10 μm.

To determine the clinical relevance of MIF in human cancers, we analyzed MIF mRNA expression in The Cancer Genome Atlas (TCGA) breast carcinoma dataset. MIF mRNA levels were significantly increased in primary and metastatic breast tumors as compared with normal breast tissues (Fig. 7j). Moreover, MIF mRNA upregulation was detected in breast tumors with histological stages 1–4 and luminal A, luminal B, HER2 +, basal-like subtypes (Fig. 7k, l). Kaplan–Meier analysis (GSE1456) revealed that overall survival in breast cancer patients with high

MIF expression was significantly shorter than that in patients with low MIF expression (Fig. 7m). Missense mutations of MIF, including P2R, M3I, N7H, R12P/L, S14F, V15M, G18W, A39T, D45V, R74H, S75F, L84M, A85V, R89P, N98I, Y99C, M102I, had been found in human cancer patients according to the COSMIC database. However, its overall mutation rate was 0.2% (59 out of 29614 in the COSMIC database). Similarly, the cBioPortal database showed that the overall mutation rate of *MIF* is relatively lower than its amplification rate in human tumors

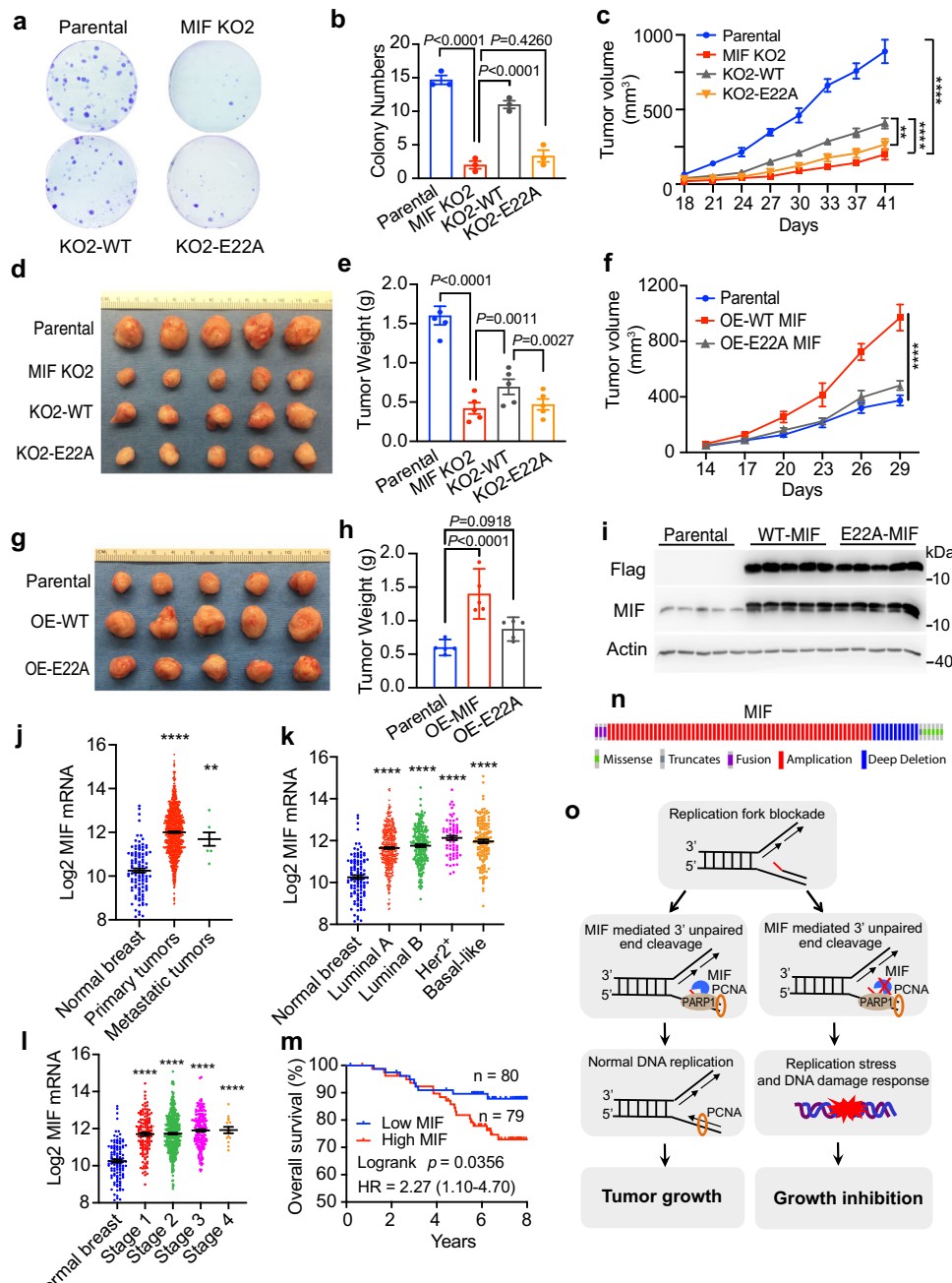

(Fig. 7n). Taken together, these data indicate that MIF is upregulated in human breast tumors and correlates with the poor clinical outcome in breast cancer patients.

## Discussion

MIF was recently identified as a $Mg^{2+}$- and $Ca^{2+}$-dependent 3′ exonuclease involved in ischemic brain injury[19]. Upon DNA damage, MIF binds to the 5′ unpaired arm of ssDNA with the stem loop structure and cleaves it at the 3′ end[19]. Here we discovered that MIF is a novel 3′ flap nuclease. It recognizes Y-shaped dsDNA and selectively cleaves 3′ unpaired flaps to facilitate DNA elongation. MIF's 3′ flap endonuclease activity is dramatically reduced with increased numbers of mismatched nucleotides (>2 nt) at the 3′ end. In contrast, its 3′ exonuclease activity leads to the cleavage of both shorter and longer nucleotides at the 3′ end, which is blocked by biotin labeling at

the 3′ end mimicking the 3′ end protection by DNA-binding proteins. The selectivity and potency of MIF's 3′ nuclease activity, especially its flap endonuclease activity that is not interfered by the shield protection at the 3′ end, make MIF an efficient guardian during DNA replication against 3′ flap structures, which can be produced not only by mis-incorporation of nucleotides, but also other factors like mis-incorporation of ribonucleotides, disruption of the balance between purine and pyrimidine, DNA damage, or DNA secondary structures[6,36,37].

Pol δ and Pol ε have the 3′ exonuclease activity and have been well recognized for proofreading DNA during replication. However, given the fact that increasing numbers of germline and somatic mutations within the exonuclease domain in human *POLD1* and *POLE* have been identified in human cancers[1,13,14] and that our in vitro nuclease assay, DNA elongation assay, and gap-filling assay clearly showed that MIF cooperates with Pol α

**Fig. 7 MIF promotes cancer cell growth in vitro and in vivo through its nuclease activity. a, b** Colony formation of parental, MIF-KO2, KO2-WT, and KO2-E22A MDA-MB-231 cells cultured for 14 days. Representative images from three experiments were shown in **a** and quantification of colony numbers was shown in **b** (mean ± SEM). Statistical significance was determined by one-way ANOVA Dunnett's multiple comparisons test. **c–e** Growth of parental, MIF-KO2, KO2-WT, and KO2-E22A MDA-MB-231 tumors in mice. Tumor volume, image, and weight were shown in **c**, **d** and **e**, respectively (mean ± SEM, $n = 5$ mice). $****P < 0.0001$ parental vs MIF-KO2, $****P < 0.0001$ MIF-KO2 vs KO2-WT, $**P = 0.0013$, KO2-WT vs KO2-E22A by two-way ANOVA Tukey's multiple comparisons (**c**) and statistical significance in **e** was determined by two-tailed Student's $t$ test. **f–h** Growth of parental, WT MIF-, and E22A-MIF-overexpressed (OE) MDA-MB-231 tumors in mice. Tumor volume, image, and weight were shown in **f**, **g**, and **h**, respectively (mean ± SEM, $n = 5$ mice per group). $****P < 0.0001$ by two-way ANOVA Tukey's multiple comparisons (**f**) and one-way ANOVA Dunnett's multiple comparisons test (**h**). **i** Immunoblot analysis of WT and E22A MIF protein levels in overexpressed tumors. **j–l** Analysis of MIF mRNA levels in human breast tumors and normal breast tissues. Data were retrieved from the TCGA dataset and presented as mean ± SEM. $n = 112$ normal breasts, $n = 1041$ primary tumors, $n = 7$ metastatic tumors (**j**); $n = 112$ normal breast, $n = 422$ Luminal A, $n = 194$ Luminal B, $n = 67$ Her2 +, $n = 142$ basal-like (**k**); $n = 112$ normal breast, $n = 133$ Stage 1, $n = 445$ Stage 2, $n = 175$ Stage 3, $n = 15$ Stage 4 (**l**). $**P < 0.01$, $****P < 0.0001$, versus normal breast, by one-way ANOVA Dunnett's multiple comparisons test. **m** Kaplan–Meier survival analysis for patients with breast cancer. Patients were divided by median expression levels of MIF mRNA. Data were retrieved from the GEO dataset (GSE1456). Statistical significance was determined by log-rank test. **n** MIF genetic alteration analysis in TCGA PanCancer Atlas studies (cBioPortal database). **o** A proposed model of MIF in the regulation of DNA proofreading and tumor growth. PARP1 interacts with MIF and recruits MIF to the DNA replication sites, where PARP1 detects DNA damage including 3′ flap structures and MIF excises unpaired flaps to facilitate DNA synthesis executed by polymerases in cancer cells, leading to cancer cell proliferation. Inhibition of MIF suppresses DNA replication and cancer cell growth.

and nuclease-deficient Pol δ to proofread DNA and ensure the success of DNA elongation, MIF is likely to play a role in proofreading in Pol δ- or Pol ε-deficient cancer cells. This is further supported by in vivo HPRT mutation analysis showing that MIF KO increases 13.6-fold nucleotide misincorporation rate in Pol δ-deficient DLD1 cells. Interestingly, our studies showed that MIF also plays an important role in nuclease-proficient MDA-MB-231 cancer cells that do not have *POLD1* and *POLE* mutations, as MIF KO increases misincorporation mutations and genomic instability, delays cell cycle, and inhibits DNA synthesis and cancer growth in vitro and in mice. Pol α plays an important role in initiation of DNA replication and Okazaki fragment on the lagging strand, but it lacks 3′ exonuclease activity for proofreading errors. Our data indicate that MIF may cooperate with Pol α to ensure the accuracy and success of the primer elongation in Pol δ- and Pol ε-proficient cancer cells. In addition, polymerase dissociation from DNA occurs when a second structure like R-loop, hairpin, stem-loop, G-quadruplex, fork reversal, or slippage is formed during replication[38–40]. Pol δ and Pol ε are also less stringent in discriminating dNTPs and rNTPs. Misincorporation of rNTPs occurs and causes replication stress[36]. Under these conditions, MIF may resolve second structures like hairpin and stem-loop and correct misincorporation, which is supported by our in vitro MIF nuclease assay and in vivo mutation analysis. Thus, MIF may also cooperate with nuclease-proficient Pol δ and Pol ε and allow them to maintain high speed of DNA replication. Further investigation is required to explore how MIF cooperates with polymerases during DNA replication.

DNA replication errors must be removed and repaired in an efficient way to ensure high fidelity of DNA synthesis. Many 5′ nucleases, including FEN1, DNA2, and EXO1, have been implicated to contribute to replication fidelity by removing replication errors using their 5′ flap endonuclease and 5′ exonuclease activities[41]. The MMR pathway is also shown to correct errors post DNA replication to maintain DNA fidelity[42]. However, to our knowledge, no 3′ flap nuclease other than polymerases has been reported so far to proofread DNA during replication. Here, we showed that MIF is a novel 3′ flap nuclease that monitors replication errors in a real-time with its 3′ nuclease activity to promote cancer cell growth. On the other hand, we found that the proofreading ability of MIF is not as precise as that of Pol δ does, because more mutations were generated in gap-filling reactions carried out by proofreading-deficient Pol δ supplemented with MIF than those conducted by Pol δ (Fig. 2h). Given the high level expression of MIF in cancer cells and that MIF promotes cancer

growth (Fig. 7g–n), the less-active proofreading activity of MIF may allow cancer cells not only to effectively deal with 3′ flap-induced replication stress for cell survival, but also to generate sufficient mutations in favor of cancer development and progression.

Our mass spectrometry analysis of MIF purification revealed very little if any protein contamination (Supplementary Table 1), which is supported by Coomassie blue staining. The purity and specificity of the MIF preparation is also confirmed by the parallel studies of nuclease-inactive mutant E22A and GST protein, which exhibited no obvious nuclease activity under the same experimental conditions. Moreover, the unique nuclease selectivity of purified MIF protein on various DNA substrates with different structure or length of nucleotides at the 3′ end reinforces the selectivity and potency of MIF's 3′ nuclease activity. Although the kinetic analysis of MIF activity revealed a relatively slow reaction with $t_{1/2}$ at 7.7–10.8 min, $K_M$ at 0.7–0.9 µM and $k_{cat}$ at 0.02/min, it is comparable to some well-recognized nucleases including MutSS2, exoribonuclease (ExoN), ribozyme, and EcoRI[43–47]. For example, the nuclease activity of *Thermus thermophilus* MutS2 (ttMutS2), which is known to play an important role in MMR, has been reported to have a $K_M$ at 290 nM and $k_{cat}$ at 0.041/min[44], which are comparable to those of MIF. The restriction nuclease EcoRI's $t_{1/2}$ is 9.7 min calculated by UV linear dichroism[46] and about 8 min monitored by fluorescence changes[47]. As with many nucleases, it is possible that the additional enhancer may help MIF bind and cleave DNA efficiently in vivo, which requires further investigation.

We showed that MIF primarily locates in the cytosol but is translocated to the nucleus when cancer cells enter S phase, which is supported by the previous clinical data showing that nuclear MIF protein is detected in a large number (79.7%) of human tumors including glioblastoma, bladder tumors, and lung adenocarcinoma[48–50]. Our biochemical and cell biology studies revealed that PARP1 is required for MIF recruitment to the DNA replication sites through their physical protein-protein interaction after MIF is translocated to the nucleus in the S phase. As a DNA damage sensor, PARP1 dynamically binds to DNA and recognizes DNA damage, thereby recruiting nucleases and DNA repair proteins to the sites of DNA damage to facilitate DNA repair[18]. Recent studies revealed that PARP1 is activated by unligated Okazaki fragments in S phase and promotes Okazaki fragment maturation[34]. In line with this report, our study illustrated that PARP1 senses DNA with 3′ flap structures and is required for MIF recruitment to the replication fork as deletion of PARP1

blocks MIF colocalization with DNA replication protein PCNA. However, PARylation of PARP1 disrupts the PARP1-MIF interaction and decreases the amount of MIF protein at the replication sites, which is supported by our data that PARP inhibitor treatment increases MIF-PCNA interaction and colocalization in the replication sites. Increase of MIF at the DNA replication sites is likely to increase replication speed as MIF loss decreases the DNA replication speed (Fig. 3f–h). Consistent with these findings, a recent study showed that PARylation controls the velocity of replication forks and that inhibition of PARylation by PARP inhibitors increases the speed of fork elongation[35]. Although our data support that PARP1 plays a pivotal role in recruiting MIF to DNA replication sites, the primary nuclear localization of PARP1 suggests that PARP1 may not mediate MIF translocation from the cytosol to the nucleus. Future studies are required to investigate the molecular mechanism underlying MIF nuclear translocation. Taken together, our findings support a possible working model that PARP1 interacts with MIF during S phase and forms a complex prior to binding to the damage sites. PARP1 as the DNA damage sensor detects DNA damage including 3′ flap structures during replication. Then MIF-PARP1 complex is recruited to the damage sites at the replication fork to resolve the replication stress, facilitating DNA synthesis and promoting cancer cell growth (Fig. 7o). Future in vivo studies are required to further support the proposed model.

Tumor cells suffer with high levels of DNA replication stress, which poses a threat to their viability. Our oncogenic studies clearly showed that MIF promotes cancer cell growth in vitro and in vivo through its nuclease activity, which provide the functional evidence that MIF's proofreading function helps cancer cells cope with DNA replication stress for their survival. However, MIF loss does not obviously alter the growth of non-tumorigenic epithelial cells MCF-10A or mouse embryonic fibroblasts, and MIF knockout mice are normal without obvious defects, suggesting that the expression of MIF is more favorable for cancer cell growth rather than normal cells. An early study reported that homozygous loss of MIF increases the development of B-cell lymphomas and multiple types of carcinoma in p53-null mice[51]. This finding, together with our current data, suggests that MIF's action on genomic stability in normal cells and cancer cells may have the distinct oncogenic consequences. This is consistent with the idea that genomic instability in normal cells has been thought to be a key driver of tumor initiation[41,52–54]. Previous studies showed that MIF has a variety of pleiotropic actions involved in inflammation and immune response. Further investigation is required to understand if MIF's nuclease activity contributes to inflammation and immune response and their impact on tumor growth.

In conclusion, we found that (1) MIF is a novel 3′ flap nuclease and specifically recognizes Y-shaped dsDNA to excise unpaired 3′ flaps. (2) MIF cooperates with nuclease-deficient polymerases (Pol α and mutant Pol δ) to proofread DNA and facilitates DNA elongation. (3) Loss of MIF nuclease activity increases mutation frequency, reduces the DNA replication speed, and causes cell cycle delay in cancer cells. (4) PARP1 senses DNA with 3′ flap structures and recruits MIF to the DNA replication sites in the S phase. (5) MIF facilitates cancer cell growth in vitro and in vivo via its nuclease activity. MIF is upregulated in human breast tumors and high levels of MIF are correlated with patients' overall poor survival. Thus, MIF's nuclease activity is a therapeutic vulnerability in cancers. Collectively, this study uncovers that MIF surveils DNA replication fidelity in cancer cells and identifies a hitherto unknown intrinsic mechanism to help cancer cells evade DNA replication stress for their survival (Fig. 7o). Because MIF is upregulated in many types of human cancers, including breast tumors, lung adenocarcinoma, hepatocellular carcinoma,

colorectal cancer, pancreatic ductal carcinoma, prostate cancer, head and neck squamous cell carcinoma, and bladder cancer[49,50,55–57], our findings here provide a strong rationale for targeting MIF and its nuclease activity to treat human cancers.

## Methods

**Mice.** In total, $2 \times 10^6$ breast cancer MDA-MB-231 cells were suspended in 100 μl PBS/Matrigel (1:1, Corning) and injected into the second left mammary fat pad of female NOD/SCID mice (6–8-weeks old, The Jackson Laboratory). In all, $2 \times 10^6$ glioblastoma LN229 or colorectal cancer HCT116 cells were subcutaneously injected into male NOD/SCID mice (6–8-weeks old, The Jackson Laboratory). Tumor volume was measured every three days starting on 11–18 days after cell implantation, and calculated with the formula: $V = 0.52 \times L \times H \times W$ ($V$: volume, $L$: length, $H$: height, $W$: width). Tumors were harvested when their volume reached ~1500 mm³. UT Southwestern Medical Center is fully accredited by the American Association for the Accreditation of Laboratory Animal Care (AAALAC). All research procedures performed in this study were approved by UT Southwestern Medical Center Institutional Animal Care and Use Committee (IACUC) in compliance with the Animal Welfare Act Regulations and Public Health Service (PHS) policy.

**Cell culture and transfection.** MDA-MB-231, LN229, HCT116, DLD1, and HEK293FT cells were cultured in DMEM or McCoy's 5a supplemented with 10% heat-inactivated FBS at 37 °C in a 5% CO₂/95% air incubator. All cell lines were mycoplasma-free and authenticated by short tandem repeat DNA profiling analysis.

**Plasmid constructs and virus production.** Mouse MIF (NM_010798) and MIF-E22A cDNAs were subcloned into an AgeI- and EcoRI-linearized lentiviral cFugw-Flag vector. MIF expression was driven by the human ubiquitin C (hUBC) promoter as previously described[19]. MIF sgRNA targeting was constructed by annealing DNA oligonucleotides and ligating into BsmBI-linearized lenti-CRISPRv2 vector (Addgene, #52961). Primers used for sgRNA constructs are listed in Supplementary Table 2. Human MIF shRNAs were designed using the online "tool "[http://katahdin.cshl.org/siRNA/RNAi.cgi?type=shRNA]. The program provided 97-nt oligonucleotide sequences for generating shRNAmirs. Using PacI-SME2 forward primer 5′ CAGAAGGTTAATTAAAAGGTATATTGCTGTTGAC AGTGAGCG 3′ and NheI-SME2 reverse primer 5′ CTAAAGTAGCCCCTTGC TAGCCGAGGCAGTAGGCA 3′, we then PCR amplified them to generate the second strand and added PacI and NheI restriction sites to clone the products into pSME2, a construct that is inserted an empty shRNAmir expression cassette in the pSM2 vector with modified restriction sites into the cFUGw backbone. This vector expresses GFP. V5-tagged full-length human PARP1 and its various truncated cDNAs were generated by PCR using pTY-U6-hPAPR1 (provided by W. Lee Kraus lab) as the template and subcloned into an XhoI/XbaI-linearized lentiviral Plvx-hUbc-Flag-C vector, which was modified by inserting Ubc promoter into pLVX-mCherry-N1 vector (Addgene). Primers for full-length PARP1 and truncates are shown in Supplementary Table 2. Lentivirus was produced by transient transfection of the recombinant cFugw vector into HEK293FT cells together with three packaging vectors: pLP1, pLP2, and pVSV-G (1.3:1.5:1:1.5). The viral supernatants were collected at 48 and 72 h after transfection and concentrated by ultra-centrifugation for 2 h at $50,000 \times g$.

**Generation of KO and KD cell lines.** MIF KO cell lines were generated using the CRISPR (Clustered regularly interspaced short palindromic repeat)/Cas9 technique. Briefly, cells were transiently transfected with sgRNA vector using PolyJet (SignaGen). Forty-eight hours post transfection, cells were treated with 1 μg/mL puromycin for 3 days, and a single cell was selected and verified by genotyping and immunoblot assays. HCT116 MIF KD cell lines were generated by infecting cells with lentivirus encoding MIF shRNA and knock down efficiency was verified by immunoblot assays.

**Protein expression and purification.** Human MIF (NM_002415) cDNA and their variants were digested by EcoRI and XhoI restriction enzymes and subcloned into GST-tagged pGex-6P-1 vector (GE Healthcare). The protein was expressed in *E. coli* and purified by glutathione sepharose as described previously[19]. The GST tag was then removed with PreScission Protease (GE Healthcare) cleavage. The purity of MIF protein preparations was confirmed by Coomassie blue staining and mass spectrometry ($n = 3$) to exclude the protein contamination (Supplementary Table 1). GST protein was used as a negative control in in vitro nuclease assay.

Pol α protein was expressed by infecting Sf9 insect cells with P2 baculovirus prepared using a pFastBac plasmid 6xHis-Tev-hPOLA1 containing the catalytic fragment of POLA1 corresponding to residues 335-1257 as described previously[58]. Cells (grown in SF-900 media) were incubated with the virus for 48 h at 27 °C, pelleted, and frozen. Frozen pellets were pulverized using a Retsch Cryomill. Pol α powder (5 ml) was resuspended in 50 ml lysis buffer containing 20 mM Tris, pH 7.8, 150 mM NaCl, 10 mM KH₂PO₄, 3% glycerol, 3 mM BME, 0.5 mM PMSF, 1× Sigma inhibitor tablet, passed through 22-G needle twice, and centrifuged at

$20,000 \times g$ for 30 min at 4 °C. Pol α protein was then purified with affinity chromatography on Ni-NTA column (Qiagen) followed by size exclusion chromatography on Superdex 200 (GE Healthcare) equilibrated in 10 mM Tris-HCl, pH 7.8, 500 mM NaCl, 1% glycerol, and 1 mM DTT. The purified protein sample was concentrated using a centricon filter (Millipore) and frozen as single-use aliquots in 10 mM Tris-HCl, pH 7.8, 50 mM NaCl, 1% glycerol, and 1 mM DTT.

Pol δ, Pol δ D402A, Pol δ D402A/L606M mutant proteins were purified using the baculovirus system as described previously[26]. Pol δ mutants D402A and D402A/L606M were defective in the 3'->5'proofreading nuclease activity, which was confirmed by in vitro elongation assays[26].

**In vitro nuclease assay**. dsDNA substrates were prepared by annealing ssDNA oligos with or without 3'-end biotin labeling (IDT). dsDNA substrates (0.8 μM) were incubated with WT MIF or catalytically inactive MIF at a final concentration of 2 μM in 10 mM Tris/HCl buffer (pH 7.0) containing 10 mM MgCl₂ for 1 h at 37 °C. The reaction was terminated with 2× formamide loading buffer containing 5 mM EDTA. Samples were boiled at 95 °C for 5 min and immediately separated on a 25% TBE-urea polyacrylamide (PAGE) gel. The gel was subsequently stained with 0.5 μg/ml EtBr followed by electrophoretic transfer to nylon membrane. Biotin-labeled DNA was further detected by chemiluminescence using a Chemi-luminescent Nucleic Acid Detection Module Kit (Thermo Fisher Scientific).

**In vitro DNA elongation assay**. A 99-nt DNA oligo was used as the template. The template-primer complex was prepared in the following reaction, including 4 μl of 100 μM 99-nt DNA template, 4 μl of 100 μM 5'-end biotin-labeled DNA primer, and 2 μl of 10× universal buffer (200 mM Tris/HCl, pH 7.8, 100 mM MgCl₂, 20 mM DTT, and 500 mM NaCl). The mixture was heated to 95 °C followed by gradually cooling down to the room temperature and then diluted at 1:10. The elongation was performed by incubating Pol δ, Pol δ D402A mutant, Pol α or Taq with the elongation reaction buffer, including 2 μl of the template substrate, 2 μl of 10 mg/ml BSA, 0.2 μl of 50 mM MgCl₂, 0.4 μl of 2.5 mM dNTPs in the presence or absence of WT or catalytically inactive MIF at 37 °C for 1 h. The reaction was terminated by adding 10 μl of 2× formamidine loading buffer and resolved on a 25% TBE-urea gel. The gel was subsequently stained with 0.5 μg/ml EtBr followed by electrophoretic transfer to nylon membrane. Biotin-labeled DNA was further detected by chemiluminescence using a Chemiluminescent Nucleic Acid Detection Module Kit (Thermo Fisher Scientific).

**DNA gap-filling and mutagenesis assay**. DNA gap-filling assay was performed as described previously[28]. In brief, a 283-nt gap of DNA substrate M13mp18 (100 ng) was filled in 10 μl reaction buffer containing 20 mM Tris/HCl (pH 8.0), 10 mM MgCl₂, 2 mM DTT, 250 μM dNTP, 2.5 U Taq DNA polymerase or Pol δ, or Pol δ D402A/L606M mutant proteins in the presence or absence of MIF protein, E22A, or GST at 37 °C for 1 h. The gap-filling reaction was terminated by adding 15 mM EDTA. DNA was then precipitated and transformed into XL-1 blue competent cells. The amplified transformed cells (30 μl) were mixed with 3 ml soft agar containing 20 μl X-gal (40 mg/ml), 4 μl IPTG (1 M) and 200 μl amplified XL-1-blue cells, and plated onto the agar plate. The mutation rate was calculated based on the ratio of white/total clones and the mutant plaques were selected for Sanger DNA sequencing.

**HPRT gene mutation assay**. HPRT gene mutation assay was modified and conducted as described previously[30,31]. DLD1 and DLD1 + MSH6 cells were cultured for several passages in RPMI-1640 + 10% FBS containing 100 μM hypoxanthine, 0.4 μM aminopterin, and 16 μM thymidine to pre-clean of the pre-existing mutants. Approximately $2 \times 10^5$ cells were seeded in triplicate onto 10-cm dishes treated with 20 μM freshly prepared 6-TG and incubated for about 12 days. Survived colonies were isolated and expanded to extract mRNA. The HPRT1 cDNA was amplified with primers (HPRT1-F- GCGCGCCGGCCGGCTCCGTT; HPRT1-R-GGCGATGTCAATAGGACTCCAGATG) targeting the entire coding region and sequenced. Plating efficiency was determined by seeding 100 cells in the absence of 6-TG. After 10 days-culture, colonies were visualized by staining with 0.1% crystal violet. The mutation frequency was determined by dividing the number of 6-TG resistant colonies by the total number of cells plated after being corrected for the colony-forming ability. Specifically, mutation frequency = number of mutant colonies/(cloning efficiency × cells seeded for mutation assay) and cloning efficiency = no. of colonies/no. of cells seeded.

**Cell cycle analysis**. Proliferating cells were synchronized at G1/S boundary with double thymidine treatment. Briefly, cells were first treated with 2 mM thymidine (Sigma) for 18 h. After 9 h release in the fresh medium, cells were further treated with 2 mM thymidine for another 15 h. Cells were then harvested at different time points varying from 0 to 9 h after release in the fresh medium and fixed in pre-cold 70% ethanol at −20 °C for at least 2 h. Subsequently, cells were stained in PI/Triton X-100 solution (PI: 2 μg/ml, Triton X-100: 0.1%(v/v)) with RNase A (0.1 mg/ml) at 37 °C for 30 min. Cell cycle distributions were assessed in a population of 20,000 cells by flow cytometry and analyzed with Flowjo software.

**BrdU staining assay**. Cells were released from double thymidine synchronization and cultured in fresh medium for another 3 h followed by BrdU (10 μM) incorporation for 30 min. Cells were fixed with fresh 4% paraformaldehyde at room temperature for 15 min and permeabilized with 0.2% Triton X-100 solution for 15 min. Subsequently, DNA was denatured by incubation for 30 min with 2 M HCl. Cells were then washed with PBS twice and blocked with 3% BSA at room temperature for 30 min. Cells were stained overnight with anti-BrdU (1:250) antibody at 4 °C followed by Alexa-568 secondary antibody incubation for 1 h at 37 °C. After washing with PBS, cells were stained with DAPI and mounted with anti-fade solution (Shandon). Immunofluorescence images were observed with Axio Observer Z1 microscope using Zen 2 software (Carl Zeiss).

**DNA fiber assay**. DNA fiber assay was performed using an adapted method as described previously[59]. Briefly, 25 μM IdU (Sigma-Aldrich, I7125) was added to asynchronously growing cells, and incubated for 20 min at 37 °C. After washing twice with PBS, cells were incubated with CldU (250 μM, Sigma-Aldrich, C6891) for 20 min. Cells were then trypsinized and resuspended in PBS at $1 \times 10^6$/ml. The cell suspension (2.5 μl) was applied to the glass slide and lysed with 7 μl of lysis buffer (50 mM EDTA and 0.5% (w/v) SDS in 200 mM Tris/HCl, pH 7.6). Slides were tilted at 15° angle to allow DNA fibers to spread across the slide and air-dried. After this, fibers were fixed with methanol/acetic acid (3:1) for 10 min and then air-dried. Fibers were denatured with 2.5 M HCl for 30 min and washed with PBS. Next, the fixed fibers were blocked with 3% BSA for 30 min at room temperature and incubated overnight with anti-BrdU antibody (BD Biosciences, #347580) specifically recognizing IdU at a 1:50 dilution and anti-BrdU antibody (Abcam, #ab6326) recognizing CldU at a 1:250 dilution respectively in 3% BSA at 4 °C. Slides were then washed with PBS for three times and incubated for 1 h with Alexa-568-conjugated anti-mouse (1:500) and Cy2-conjugated anti-rat (1:250) antibodies (Jackson ImmunoReaserch) in 3% BSA at room temperature in the dark. After secondary antibody incubation, cells were washed and mounted. Immuno-fluorescence images were observed with Axio Observer Z1 microscope (Carl Zeiss). To assess fork speed at least 50 fibers per sample were measured using the line tool in ImageJ software.

**iPOND assay**. iPOND assay was performed according to the protocol described previously[60]. Double thymidine synchronized LN229 cells ($1 \times 10^8$ cells) were treated with 10 μM EdU (Thermo Fisher Scientific, A10044) for 20 min followed with or without 10 μM Thymidine chase for 1 h. Then, cells were immediately crosslinked with 1% formaldehyde (Sigma, F1635) for 10 min at 37 °C and quenched with 0.125 M glycine for 5 min. Cells were collected and washed with 1x PBS for three times. Cell pellets were resuspended in permeabilization buffer containing 0.25% Triton X-100 (Sigma, T8787) at $1 \times 10^7$ cells/ml and incubated at room temperature for 30 min. After washing once with cold PBS containing 0.5% BSA and twice with PBS, cell pellets were incubated by rotation at room temperature for 2 h in Click-it reaction buffer containing 10 mM sodium ascorbate (Sigma A4034), 2 mM CuSO₄ in PBS with or without 1 μM Biotin azide (Thermo Fisher Scientific, B10184) or 1 μM DMSO. Then, cells were lysed in lysis buffer containing 50 mM Tris-HCl (pH 8.0) and 1% SDS supplemented with protease Inhibitor. Chromatin was solubilized with sonication using BRANSON Digital Sonifier at amplitude 10%, 1-s pulse on and 1-s pulse off for 200 cycles. The supernatant was collected after the centrifugation at $16,100 \times g$ for 10 min and further diluted by PBS containing protease inhibitor (1:1). The resulting supernatant was further incubated for 16 h with 100 μl streptavidin agarose beads (Novagen, 69203) at 4 °C. After washing twice with lysis buffer and once with 1 M NaCl, proteins were eluted with 2× Laemmli buffer and boiled at 95 °C for 25 min before detection by immunoblot.

**Chromosome spread assay**. Cells were grown to reach 70% confluence and treated with 1 μg/ml colcemid for 2 hr. Cells were then harvested and suspended in 1 ml KCl (75 mM) for 30 min at 37 °C. After centrifugation, cells were fixed with cold methanol/acetic acid (3:1) buffer and incubated at room temperature for 15 min. Metaphase spreads were made by dropping cells onto the slide and air-dried. The spreads were stained with DAPI and visualized under the microscope.

**Clonogenic survival assay**. Cells were seeded at 100 cell/well onto a six-well plate and cultured at 37 °C for 2 weeks. Colonies were fixed with methanol and stained with 0.1% crystal violet (Sigma). After staining, colonies were gently washed with PBS and counted. Colonies with ≥50 cells were counted for quantification.

**GST pull-down assay**. GST, GST-MIF-WT, and GST-MIF-E22A proteins bound to GSH beads (500 ng) were incubated with MDA-MB-231 cell lysates, or 75 ng PCNA or 5 U PARP1 recombinant proteins respectively in binding buffer containing 0.2% Triton X-100, 50 mM Tris-HCl (pH 7.5), 100 mM NaCl, 15 mM EGTA, 1 mM DTT, and 1 mM PMSF by rotation overnight at 4 °C. After washing with lysis buffer (0.5% Triton X-100, 50 mM Tris-HCl (pH 7.5), 100 mM NaCl, 15 mM EGTA, 1 mM DTT, and 1 mM PMSF) for three times, proteins bound on beads were eluted with 20 μl 1× Laemmli buffer, boiled, and subjected to SDS-PAGE gel, followed by immunoblot assay.

**In vitro PARylation assay**. In vitro PARylation assay was conducted as previously described with some modifications[61]. Briefly, GST and GST-MIF (500 ng) bound to GSH beads were incubated with 1 U recombinant PARP1 protein (Trevigen) in the reaction buffer containing 50 mM Tris-HCl (pH 8.0), 20 mM NaCl, 10 mM MgCl$_2$, 1 mM DTT, 1× activated DNA (BPS Bioscience) and 500 μM NAD$^+$ at 37 °C for 30 min. The reaction was terminated by collecting supernatant and beads separately with the addition of SDS loading buffer. PARylation of interested proteins was detected by immunoblot assay with an anti-PAR antibody.

**Immunoprecipitation and immunoblot assays**. Cells were lysed with NETN lysis buffer (150 mM NaCl, 1 mM EDTA, 10 mM Tris-HCl, pH 8.0, 0.5% IGEPAL, and protease inhibitor cocktail) for 30 min on ice. After centrifugation at 15,000× g for 15 min, the supernatant was incubated overnight with antibodies against to MIF (2.5 μg/ml), PCNA (2.5 μg/ml), poly(ADP-ribose) (2.5 μg/ml) in the presence of protein A/G magnetic beads (Bio-Rad) at 4 °C. On the next day, after washing five times with NETN lysis buffer, proteins bound on beads were boiled in 1× SDS loading buffer, separated on SDS-PAGE gel, and transferred to a nitrocellulose membrane. The blot was incubated with indicated primary antibody followed by HRP-conjugated secondary antibody (Supplementary Table 3). The immune complexes were detected by the ECL prime western blotting detection reagent (GE Healthcare). Images were taken using Image Lab (Version 6.0.1).

**Statistics and reproducibility**. The data were expressed as mean ± SEM (standard error of the mean). Statistical evaluation was performed by unpaired two-tailed Student's $t$ test between two groups and by one-way ANOVA Dunnett's multiple comparisons or two-way ANOVA Tukey's multiple comparisons within multiple groups using GraphPad Prism 8.0 software. *$P < 0.05$, **$P < 0.01$, ***$P < 0.001$, ****$P < 0.0001$ are considered significant and ns is not significant. Kaplan–Meier survival curve was analyzed by log-rank test. Cells with Pearson's correlation coefficient ($r$) > 0.5 is considered as the positive correlation of MIF-PCNA colocalization, $r$ > 0.7 is considered as a strong correlation, and $r$ < 0.4 is considered as a weak or no correlation. All immunoblots were repeated at least three times independently with similar results. The precise sample number ($n$) was provided to indicate the number of independent biological samples in each experiment.

## Data availability

MIF expression data in human breast tumors in the TCGA dataset were downloaded from the UCSC Cancer Browser (https://genome-cancer.ucsc.edu). MIF mRNA expression was queried in adjacent normal breast tissues, and primary and metastatic breast tumors with different subtypes or stages. Kaplan–Meier survival analysis for patients with breast cancer by log-rank test was retrieved from the GEO dataset (GSE1456). MIF genetic alteration analysis in TCGA PanCancer Atlas studies was from cBioPortal database (https://www.cbioportal.org/). All data generated or analyzed during this study are included in this article and its supplementary information files. Any additional data presented in this paper are available from the corresponding author upon request. Source data are provided with this paper.

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

## Acknowledgements

We thank Dr. W. Lee Kraus for providing pTY-U6-hPAPR1 plasmid, Dr. Elliott Ross for the helpful discussion on MIF kinetics analysis and the UTSW proteomic Core for mass spectrometry analysis. This work was supported by grants from the Welch Foundation (I-1939), CPRIT (RP170671), National Institutes of Health (R00NS078049, R35GM124693, and R01AG066166), Darrell K Royal Research Fund, TIBIR pilot Grant, the University of Texas Southwestern Medical Center Startup funds and UT Rising Stars to Y.W.; CPRIT (RR160101) to G.L.; Welch Foundation (I-1879), National Institutes of Health (R37CA226771 and R01CA217333) to D.N.; National Institutes of Health (R00CA168746 and R01CA222393), CPRIT (RR140036 and RP190358), Susan G. Komen® Foundation (CCR16376227), Mary Kay Foundation (08-19), and Welch Foundation (I-1903) to W.L. W.L. is a CPRIT Scholar in Cancer Research.

## Author contributions

Y.W. and W.L. conceived the study, analyzed the data, and wrote the paper; Y.J.W. performed the most experiments, analyzed the data, and wrote the paper; Y.C. and M.Y. performed part of the mouse injection studies; C.W. established PARP1 KO cell lines; Y.N.W. and L.B. contributed to plasmid constructs; J.E.W. contributed to mouse breeding and virus production; B.K. established multiple MIF KO cell lines, E.C. and D.N. contributed to Pol α protein purification; K.C., W.X., J.O., and G.M.L. purified Pol δ, Pol δ D402A, Pol δ D402A/606M proteins, and M13mp18 gap-filling substrate and established DLD1 + MSH6 cell line.

## Competing interests

The authors declare no competing interests.
