## [Peer Review File · Nature Communications]

Editorial Note: Items on pages 14, 15, 16, 17, 34 & 35 reproduced from Wang Y, *et al.* A nuclease that mediates cell death induced by DNA damage and poly(ADP-ribose) polymerase-1. *Science* **354**, (2016). Reprinted with permission from AAAS.

Reviewers' comments:

Reviewer #1 (Remarks to the Author):

The authors of this manuscript demonstrate that MIF is a 3' exo and 3' endonuclease that can cleave few nucleotides. Previously this nuclease was known for its function in cytoplasm in innate immune response. Here they show that the MIF can eliminate nonhomologous sequences at 3' end to promote synthesis by replicative polymerases and it does it possibly in a redundant way with 3' exo activity of Poldelta. Addition of purified MIF to Poldelta mutant with decreased 3' exo activity reduced the level of mutation in gap synthesis assay. MIF mutant cells can't efficiently enter S-phase and show slower replication. All these function of MIF depend on its nuclease activity. MIF translocates to the nucleus in S-phase and localizes at replication forks as checked by iPOND. It interacts with PCNA however this interaction is likely not direct as PIP boxes were dispensable for interaction. They performed mass spec to identify MIF interactors and many proteins were identified including DNA repair and replication proteins. PARP1 was among these proteins and the interaction of MIF and PARP1 was important for MIF-PCNA interaction. MIF depletion leads to spontaneous DNA damage, formation of micronuclei and nuclear bridges. Together the function of MIF in replication is convincing. However, the exact function the authors propose for MIF is questionable.

Critique:

My major critique is related to exact function of MIF. Authors argue that it is needed for correction of mis incorporated nucleotides. While evidence with purified MIF supports the idea that this nuclease promotes high fidelity DNA synthesis, it remains questionable why additional exonuclease is needed besides the one carried by Poldelta and Polepsilon. The data shown in Fig 2G suggest that MIF can compensate the loss of Pold exo activity but to my knowledge Pold exo mutant shows very severe phenotype with respect to increased rate of mutations. Authors should use some mutation reporter assay and check the rates of mutations in MIF deficient, Pold exo deficient cells and the double mutant cells.

Also yeast Poldelta was shown to remove short nonhomologous tails in cells similar to the one studied here. PMID: 9343441 manuscript shows that 3' tails shorter than 30 nt are removed by mechanism that depends at least in part on the 3'-to-5' proofreading activity of DNA polymerase delta. Perhaps the authors could consider alternative possibilities where 3' nuclease can be beneficial in replication.

Are MIF mutant cells sensitive to DNA damage?

Are any patients reported with mutation in MIF?

Is MIF evolutionary conserved and if so, which organisms have it?

Fig. 1H-K – MIF does not look like highly active nuclease toward the substrate tested. It takes 10 minutes to cleave once half of the substrate with an excess of enzyme. How does it compare to other nucleases operating at replication fork like Fen1?

Fig. 2G Please add WT Poldelta with MIF – this is important control to show if the presence of MIF further reduces mutation rates.

In fig 1. Is the hairpin present in all substrates? Is this hairpin needed for nuclease to cleave 3' strands, why the little scheme in Fig 1D does not show hairpin while Fig. 1A SL-7 nt shows it?

Minor

Introduction:

"Although polymerases δ and ϵ have been thought to be responsible for the removal of misincorporated nucleotides in a 3'->5' direction, increasing numbers of germline mutations as well as somatic mutations of polymerases δ and ϵ , which lack 3' exonuclease activity has been identified in human cancers including colorectal cancer, breast cancer and glioblastoma 14."

This sentence is unclear. What do authors want to say here? The fact that polymerases delta and epsilon use their exonuclease activity to remove mis-incorporated nucleotides is not questionable, this is clear from literature and all organisms ever tested. The fact that there are cancer cells that lost these activities does not negate the fact that these exonuclease activities are very important.

"It is a long mystery in the field what 3' flap nuclease helps cancers resolve 3' flap structures to overcome deleterious consequences resulting from high levels of DNA replication stress for their rapid proliferation."

Unclear what the authors want to say. Which 3' flap nuclease they refer to? MIF? Some unknown nuclease? Is it clear that this activity was needed to be identified and if so based on what exactly observations? It is confusing.

"In this study, we identified MIF as a hitherto unrecognized 3' flap nuclease. MIF is translocated from the cytosol into the nucleus in S phase and recruited to DNA replication sites by PARP1, where MIF recognizes and cleaves mis-incorporated nucleotides from Y-shaped dsDNA at the 3' end during DNA replication."

For clarity I suggest separating any discussion of in vitro studies with purified MIF from studies that are done in cells. The authors show that MIF is present at replication forks, and that purified MIF can cleave few nucleotides at 3' strands at Y structures but it is not demonstrated that in cells MIF removes mis-incorporated nucleotides.

Reviewer #2 (Remarks to the Author):

This manuscript describes in vitro and in vivo studies of the migration inhibitory factor (MIF). It is claimed that the studies indicate that MIF is a structure-specific 3' flap nuclease that removes mismatched 3'-nucleotides during DNA replication and that the protein is upregulated in cancer and therefore a new chemotherapy target. MIF is previously well-characterised as a phenylpyruvate tautomerase, so this would be a novel and highly significant finding. Indeed, the initial association of MIF with (at that stage unspecified type of) nuclease activity was published in Science.

I do have potential serious concerns about the conclusions reached in this manuscript. Undoubtedly, the samples of "purified" protein used in this study contain a 3'-nuclease as evidenced by the gels in figure 1. However, the catalytic parameters reported for the enzyme make me nervous that a contaminant is a reasonable explanation for the observed activity. The half-life for the reaction putatively catalysed by MIF is reported to be 10.3 min, whereas typically this figure would be in the order of 10-500 ms for a nuclease, so the reaction observed here is at least 10,000 slower than might normally be expected. As a caveat it needs to be acknowledged that there are examples of exceptional nucleases that do have much longer half-lives; these are generally activated by a protein partner in vivo. There are also examples of others that have slow multiple turnover rates because they bind product strongly; it is not clear from the manuscript whether multiple or single turnover conditions were used for this measurement. Showing lack of other bands on a typically loaded stained gel is not evidence of lack of contaminating activity; this could be present in very small amounts (e.g. 1/10,000) and still give the level of activity reported here.

The whole manuscript hinges on whether this is a bona fide 3'-nuclease as this observation is key to interpretation of all the subsequent experiments. The only evidence that is presented here that in part supports a real enzyme activity for MIF is the lack of (all) activity of a mutant-I would like to see a mutant with reduced activity to be more confident that this really is a nuclease. Moreover, I would like to see some structural analysis that convinces this protein has a nuclease active site. I understand that by sequence alignment it is claimed that MIF has a restriction endonuclease (RE) active site; I would like to see an overlay of the MIF protein and RE active site-the C-alphas should

occupy almost identical positions to conserve catalytic activity. Furthermore, if MIF is a nuclease it would be expected to bind nucleic acid (in the presence of EDTA to prevent reaction); thus, some evidence that this is the case would be preferable. It is also possible to run protein gels containing DNA to detect bands associated with nuclease activity; if MIF is a bona fide nuclease then this should be exclusively the MIF band (example in *Nucleic Acids Research*, 35, 4094–4102, <https://doi.org/10.1093/nar/gkm396>). More evidence needs to be sought that MIF really is a nuclease.

A lesser concern is that I have reservations that some of the kinetic analyses were not carried out under appropriate conditions. To determine Michaelis-Menten parameters, the concentration of enzyme should be significantly lower than all the substrate concentrations. Moreover, analyses that rely on single data points are fraught with errors so a number of data points that fit to a straight line should be obtained to measure the initial rate. The conditions under which half-lives are determined must be specified—usually they would be where $[S] < \text{or} = K_m$ and $[E] \gg K_m$ to give the maximal rate without turnover. Furthermore, as the authors know the concentration of MIF they should have been able to determine k_{cat} and shouldn't need to quote V_{max} ($=k_{cat} \times [E]$). In several places it's not clear what concentrations of reagents have been used and precisely what methods have been employed. It is also not clear what DS-1b is—there are several different ways of attaching 3'-biotin so it is important to know either what reagents were used to create the substrate or its full final structure. A minor point— K_m has a capital K (equilibrium constant).

Reviewer #3 (Remarks to the Author):

In this manuscript, the authors describe the role of macrophage migration inhibitory factor (MIF) in the proofreading process of misincorporated nucleotides during DNA replication. MIF has been shown to possess nuclease activity on stem loop DNA structures and play a key role in the DNA fragmentation process in the cell death pathway initiated by PARP1 activation. The authors show that MIF displays 3'-exo- and 3' endo-nuclease activities in vitro on unpaired 3' DNA ends that mimic nucleotide misincorporation during DNA synthesis. MIF KO cells show DNA replication defects, DNA damage, and abnormal chromosomes. The authors find that MIF interacts with PCNA and PARP1, which they propose to play a role in the recruitment of MIF to replication sites containing misincorporated nucleotides. Finally, the authors use xenograft models to demonstrate that MIF, which is often overexpressed in various tumors, can promote tumor growth.

The proposed model that MIF proofreads misincorporated DNA and thereby mitigates replication stress and promotes tumor growth is novel. However, the study appears to be based on the weak premise that cancer cells experience elevated DNA replication stress due to nucleotide misincorporation. Because DNA misincorporation is generally believed to cause mutagenesis rather than replication stress, the authors need to provide more evidence that the elevated fork stalling in MIF KO cells is caused by nucleotide misincorporation. In addition, the study lacks insight into the molecular mechanisms. The roles of PCNA, PARP1 and PARP1 activity in MIF recruitment to replication sites have not been developed sufficiently. Furthermore, there are a number of technical issues that are likely to impact their interpretation of the data.

In summary, this manuscript presents interesting and potentially important observations. However, the study does not have solid data and proper interpretations to support their conclusions.

Major points

- The proofreading function of MIF was shown only in vitro. The authors need to measure in vivo mutation frequency in MIF KO cells using established methods such as 6-thioguanine selection, which quantitates spontaneous mutations in the HPRT gene.
- Fig. 2A-D: Recombinant PCNA protein needs to be added to the in vitro elongation assays. SDS-PAGE gels of the purified polymerases should be included.
- Fig. 3F-H: The authors provided no data to support that the excessive replication fork stalling in

MIF KO cells is caused by nucleotide misincorporation.

- Fig. 4D: iPOND experiments need thymidine chase samples to prove that MIF is enriched at nascent DNA.
- Fig. 4E and 4F: Is MIF-PARP1 interaction direct? In vitro GST pull-down assays using purified recombinant proteins should be performed.
- Figs. 4G and 4H: Colocalization of MIF with EdU and PCNA is hard to interpret because EdU and PCNA foci are not clearly visible. Pre-extraction of non-chromatin proteins might improve the experiments. Colocalization needs to be properly quantitated.
- The mechanism of MIF recruitment to the unpaired 3' DNA ends is not clear. Fig. 5D suggests that PARP1 might be a sensor of misincorporated nucleotides, but the authors did not examine whether PARP1 binds to unpaired 3'-DNA ends in vitro. The role of PARP1 in the recruitment of MIF to replication sites is difficult to interpret because PARP1 KO appears to affect nuclear translocation of MIF (Fig. 5H).
- Fig. 5J-L: The authors concluded that "PARylation of PARP1 suppresses MIF and PCNA interaction" (Page 16: line 11) based on the increased MIF-PCNA interaction in olaparib-treated cells. However, the authors need to examine the alternative possibility that PARylation of MIF by PARP1 release MIF from PCNA, which is consistent with the increased PCNA-MIF interaction after olaparib treatments.
- Page 23, line 4: The statement "PARP inhibitor Olaparib suppresses PAR synthesis and enhances MIF recruitment to the DNA replication sites" is not fully supported by data. Does the recruitment of MIF to replication foci increase in olaparib-treated cells? The authors need to test this directly by immunofluorescence.
- Page 23, line 4: The statement "PARylation of PARP1 interferes with PARP1-MIF interaction" contradicts with Fig. 5J, which shows no change in the amount of PARP1 co-immunoprecipitated with MIF in olaparib-treated cells.
- The authors need to determine whether cancer cells are more dependent on MIF than normal cells are. Based on the COSMIC database, no mutations have been found in POLD1 or POLE1 in MDA-MB-231, which was used in the experiments. What then causes nucleotide misincorporation that makes MIF critical for DNA replication (Fig. 3) and tumor growth (Fig. 7)?

Minor points

- Fig. 1A: Several different flap sequences need to be tested for sequence specificity of MIF.
- Fig. 2B: MIF-only samples need to be included to eliminate a possibility that the MIF preparation contains polymerase activity (intrinsically or due to contamination).
- Fig. 2F: Does MIF influence the fidelity of WT Pol δ ?
- Fig. S4A: In vitro GST pull-down assays using purified PCNA and GST-MIF need to be done to test whether interaction of MIF and PCNA is direct.
- Fig. 7I: The authors need to show MIF WB to compare the overexpressed levels with endogenous MIF levels.

Response to Reviewers

Referees comments are highlighted in blue and the response is provided below the referee comments.

Reviewer #1:

1. The authors of this manuscript demonstrate that MIF is a 3' exo and 3' endonuclease that can cleave few nucleotides. Previously this nuclease was known for its function in cytoplasm in innate immuneresponse. Here they show that the MIF can eliminate nonhomologous sequences at 3' end to promote synthesis by replicative polymerases and it does it possibly in a redundant way with 3' exo activity of Poldelta. Addition of purified MIF to Poldelta mutant with decreased 3' exo activity reduced the level of mutation in gap synthesis assay. MIF mutant cells can't efficiently enter S-phase and show slower replication. All these functions of MIF depend on its nuclease activity. MIF translocates to the nucleus in S-phase and localizes at replication forks as checked by iPOND. It interacts with PCNA however this interaction is likely not direct as PIP boxes were dispensable for interaction. They performed mass spec to identify MIF interactors and many proteins were identified including DNA repair and replication proteins. PARP1 was among these proteins and the interaction of MIF and PARP1 was important for MIF-PCNA interaction. MIF depletion leads to spontaneous DNA damage, formation of micronuclei and nuclear bridges. Together the function of MIF in replication is convincing. However, the exact function the authors propose for MIF is questionable.

Response: We thank the referee for agreeing that the function of MIF in replication is convincing.

2. My major critique is related to exact function of MIF. Authors argue that it is needed for correction of mis incorporated nucleotides. While evidence with purified MIF supports the idea that this nuclease promotes high fidelity DNA synthesis, it remains questionable why additional exonuclease is needed besides the one carried by Poldelta and Polepsilon. The data shown in Fig 2G suggest that MIF can compensate the loss of Pold exo activity but to my knowledge Pold exo mutant shows very severe phenotype with respect to increased rate of mutations. Authors should use some mutation reporter assay and check the rates of mutations in MIF deficient, Pold exo deficient cells and the double mutant cells.

Response: As the reviewer suggested, we performed HPRT mutation assay in MIF-deficient, Pol δ -deficient, and MIF/Pol δ -double deficient MDA-MB-231 cells. MIF knockout significantly increased the HPRT mutation rate, mainly point mutation/mis-incorporation mutation, in Pol δ -proficient MDA-MB-231 (Fig. 3i, j in the revised manuscript). However, Pol δ -deficient MDA-MB-231 cells are not able to form colonies (Supplementary Fig. 3l-m in the revised manuscript), which does not allow us to study HPRT mutation rate in Pol δ -deficient and MIF/Pol δ -double deficient MDA-MB-231 cells. Alternatively, we performed HPRT mutation assay in a colorectal adenocarcinoma cell line DLD1, which expresses Pol δ R689W and R506H mutations in the conserved DNA polymerase III and exonuclease III motifs that attenuates the proofreading function (Supplementary Fig. 3n, o in the revised manuscript). We found that MIF knockout increased the nucleotide mis-incorporation mutation rate by 13.7-fold in DLD1 cells (Fig. 3k-l in the revised manuscript). MIF KO DLD1 cells expressing exogenous MSH6 (DLD1+MSH6) were also established, as DLD1 cells are deficient in MMR due to the mutation in MSH6 (Supplementary Fig. 3n, o). Restoring MSH6 expression in DLD1 reduced the mutation rate. However, MIF KO still significantly increased the misincorporation rate in these DLD1+MSH6 cells (Fig. 3k). Overall, these results uncover that MIF has a significant impact on nucleotide misincorporation in cancer cells and MIF deficiency elevates the mutation frequency. The new figures were included as Fig. 3i-l and supplementary Fig. 3l-o, and new data have been described in the revised manuscript (page 12-13) as following:

The hypoxanthine-guanine phosphoribosyltransferase (HPRT) gene on the X chromosome has been used as a model gene to investigate mutability in mammalian cell lines^{30, 31}. To further explore the effect of MIF on replication fidelity, we established POLD1 knockdown (KD) and POLD1 KD/MIF KO MDA-MB-231 cells using CRISPR/Cas9 sgRNAs (Supplementary Fig. 3l) and performed HPRT mutation assay following 6-TG treatment (20 μ M, 12 days). We found that the HPRT mutation rate was very low (2.6×10^{-7}) in Pol δ -proficient MDA-MB-231 cells (Fig. 3i), whereas MIF KO increased the HPRT mutation rate (1.43×10^{-6}), which was mainly due to the nucleotide misincorporation (Fig. 3i, j). Since POLD1 KD MDA-MB-231 cells were not able to form colonies for HPRT assay (Supplementary Fig. 3m), we instead established MIF KO in colorectal cancer cell line DLD1 cells, which have Pol δ mutations R689W and R506H in the conserved DNA polymerase III and exonuclease III motifs, respectively³². The overall mutation frequency in Pol δ -deficient DLD1 cells was about 8.68×10^{-5} , which was increased by 1.6-fold in MIF KO cells (1.4×10^{-4} , Fig. 3k). It is of note that the nucleotide misincorporation rate in parental DLD1 cells was quite low (7.1×10^{-6}) and MIF KO significantly increased the misincorporation rate by 13.6-fold (9.65×10^{-5} , Fig. 3l). MIF KO DLD1 cells expressing WT MSH6 (DLD1+MSH6) were also established, as DLD1 cells are deficient in MMR due to the mutation in MSH6 (Supplementary Fig. 3n, o). Restoring MSH6 expression in DLD1 cells reduced the overall mutation rate to 1.55×10^{-5} , which is expected. However, MIF KO still significantly increased the overall mutation frequency (1.0×10^{-4} , Fig. 3k). Notably, the misincorporation rate in the MMR-proficient DLD1 cells was only about 3.5×10^{-6} , which was increased by 19-fold in MIF KO cells (6.67×10^{-5} , Fig. 3k). These findings indicate that MIF has a significant impact on correcting nucleotide misincorporation *in vivo*.

3. Also yeast Poldelta was shown to remove short nonhomologous tails in cells similar to the one studied here. PMID: 9343441 manuscript shows that 3' tails shorter than 30 nt are removed by mechanism that depends at least in part on the 3'-to-5' proofreading activity of DNA polymerase delta. Perhaps the authors could consider alternative possibilities where 3' nuclease can be beneficial in replication.

Response: We thank the referee for constructive suggestion! Our extensive *in vitro* studies including elongation assay and gap-filling assay showed that MIF cooperated with nuclease-deficient Pol δ and Pol α to proofread DNA and facilitated DNA synthesis, which was further supported by *in vivo* HPRT mutation assay in Pol δ -deficient DLD1 cells. In parallel, we also showed that MIF knockout in Pol δ -proficient cancer cells still increased mutation frequency and decreased the replication speed as well as xenograft tumor growth in mice. Pol α plays an important role in initiation of DNA replication and Okazaki fragment on the lagging strand, but it lacks 3' exonuclease activity for proofreading errors. Our data indicate that MIF may cooperate with Pol α to ensure the accuracy and success of the primer elongation in Pol δ and Pol ϵ proficient cancer cells. In addition, polymerase dissociation from DNA occurs when a second structure like R-loop, hairpin, stem-loop, G-quadruplex, fork reversal or slippage is formed during replication (Kaushal S. and Freudenreich C. Genes Chromosomes Cancer, 2019, PMID: 30536896; Maffia A. et al., Int J Mol Sci. 2020, PMID: 32098397; Viguera E. et al., EMBO J. 2001, PMID: 11350948) and Pol δ and Pol ϵ are less stringent in discriminating dNTPs and rNTPs. Misincorporation of rNTPs occurs and causes replication stress (Zeman MK. et al., Nat Cell Biol. 2014, PMID: 24366029). MIF may resolve second structures like hairpin and stem-loop and correct misincorporation, which is supported by our *in vitro* MIF nuclease assays and *in vivo* mutation analysis. Thus, MIF may also cooperate with nuclease-proficient polymerases δ and ϵ and allow them to maintain high speed of DNA replication, which may also explain the discrepancy of polymerase-mediated high rate of nucleotide mis-incorporation *in vitro* but low rate *in vivo*. Future study is required to further explore how MIF cooperates with polymerases in replication. Together, our various *in vitro* and *in vivo* studies including the newly conducted HPRT mutation assay revealed that MIF has DNA proofreading function and is important and favorable for tumor cell growth *in vitro* and *in vivo*. We have included new discussion about the possible functions of MIF as 3' nuclease in DNA replication in the revised manuscript (page 22-23) as following:

Pol δ and Pol ϵ have the 3' exonuclease activity and have been well recognized for proofreading DNA during replication. However, given the fact that increasing numbers of germline and somatic mutations within the

exonuclease domain in human *POLD1* and *POLE* have been identified in human cancers^{1,13,14} and that our *in vitro* nuclease assay, DNA elongation assay and gap filling assay clearly showed that MIF cooperates with Pol α and nuclease-deficient Pol δ to proofread DNA and ensure the success of DNA elongation, MIF is likely to play an important role in proofreading in Pol δ - or Pol ϵ -deficient cancer cells. This is indeed further supported by *in vivo* HPRT mutation analysis showing that MIF KO increases 13.6-fold nucleotide mis-incorporation rate in Pol δ -deficient DLD1 cells. Interestingly, our studies showed that MIF also plays an important role in nuclease-proficient MDA-MB-231 cancer cells that do not have *POLD1* and *POLE* mutations, as MIF KO increases mis-incorporation mutations and genomic instability, delays cell cycle and inhibits DNA synthesis and cancer growth *in vitro* and in mice. Pol α plays an important role in initiation of DNA replication and Okazaki fragment on the lagging strand, but it lacks 3' exonuclease activity for proofreading errors. Our data indicate that MIF may cooperate with Pol α to ensure the accuracy and success of the primer elongation in Pol δ - and Pol ϵ -proficient cancer cells. In addition, polymerase dissociation from DNA occurs when a second structure like R-loop, hairpin, stem-loop, G-quadruplex, fork reversal or slippage is formed during replication^{36, 37, 38}. Pol δ and Pol ϵ are also less stringent in discriminating dNTPs and rNTPs. Misincorporation of rNTPs occurs and causes replication stress³⁹. Under these conditions, MIF may resolve second structures like hairpin and stem-loop and correct misincorporation, which is supported by our *in vitro* MIF nuclease assay and *in vivo* mutation analysis. Thus, MIF may also cooperate with nuclease-proficient Pol δ and Pol ϵ and allow them to maintain high speed of DNA replication. Further investigation is required to explore how MIF cooperates with polymerases during DNA replication.

4. Are MIF mutant cells sensitive to DNA damage?

Response: As suggested, we tested the response of MIF knockout cells to hydroxyurea-induced DNA replication stress by both DNA fiber and colony formation assays (Supplementary Fig. 3h-k in the revised manuscript). In response to hydroxyurea, DNA replication speed was reduced in MDA-MB-231 cells and further significantly reduced in MIF knockout MDA-MB-231 cells. Expression of MIF, but not the nuclease-deficient MIF mutant E22A, in MIF knockout MDA-MB-231 cells reversed the replication speed and the number of the stalled replications forks (Supplementary Fig. 3h-j in the revised manuscript). Moreover, MIF knockout significantly reduced colony survival following hydroxyurea treatment, which can be rescued by the expression of MIF, but not MIF mutant E22A (Supplementary Fig. 3j in the revised manuscript). These results suggest that MIF-deficient cells are sensitive to DNA replication stress. We have included these new results in the revised manuscript (page 12) as following:

To study whether MIF KO cells are more sensitive to DNA replication stress, hydroxyurea (Hu, 2 mM for 2 h) was applied to parental and MIF KO MDA-MD-231 cells. As expected, hydroxyurea treatment decreased the replication frequency and length (21.2 μm) in parental cells, which were further reduced in MIF KO cells (10.6 μm , Supplementary Fig. 3h, i). The stalled replication fork number was significantly increased in MIF KO cells (Supplementary Fig. 3j), which was reversed by the expression of WT but not E22A MIF (Supplementary Fig. 3h-j). In line with this observation, the colony survival of MIF KO cells was significantly reduced in response to hydroxyurea varying at 25-100 μM as compared with those of parental cells, which was rescued by the expression of WT but not E22A MIF (Supplementary Fig. 3k). These data indicate that MIF-deficient cells are more sensitive to DNA replication stress.

5. Are any patients reported with mutation in MIF?

Response: Missense mutations of MIF, including P2R, M3I, N7H, R12P/L, S14F, V15M, G18W, A39T, D45V, R74H, S75F, L84M, A85V, R89P, N98I, Y99C, M102I, had been found in human cancer patients according to the COSMIC database. However, the rate of MIF mutation is low in human cancer patients (0.2%). In contrast, the amplification rate of MIF in human tumors is relatively higher than its mutation rate, which has a strong clinical relevance. We have included this MIF mutation information in the Result section of the revised manuscript (page 21) as following:

To determine the clinical relevance of MIF in human cancers, we analyzed MIF mRNA expression in The Cancer Genome Atlas (TCGA) breast carcinoma dataset. MIF mRNA levels were significantly increased in primary and metastatic breast tumors as compared with normal breast tissues (Fig. 7j). Moreover, MIF mRNA upregulation was detected in breast tumors with histological stages 1-4 and luminal A, luminal B, HER2+, basal-like subtypes (Fig. 7k, 7l). Kaplan-Meier analysis (GSE1456) revealed that overall survival in breast cancer patients with high MIF expression was significantly shorter than that in patients with low MIF expression (Fig. 7m). Missense mutations of MIF, including P2R, M3I, N7H, R12P/L, S14F, V15M, G18W, A39T, D45V, R74H, S75F, L84M, A85V, R89P, N98I, Y99C, M102I, had been found in human cancer patients according to the COSMIC database. However, its overall mutation rate was 0.2% (59 out of 29614 in the COSMIC database). Similarly, the cBioPortal database showed that the overall mutation rate of MIF is relatively lower than its amplification rate in human tumors (Fig. 7n). Taken together, these data indicate that MIF is upregulated in human breast tumors and correlates with the poor clinical outcome in breast cancer patients.

6. Is MIF evolutionary conserved and if so, which organisms have it?

Response: MIF is highly conserved in mammals as shown in the recent study (Figure S3A-B, Wang Y. et al., Science 2016, PMID: 27846469) but less conserved among invertebrate species. MIF-like proteins are found in all species including bacteria, protozoa, helminths, molluscus, arthropods, fish amphibians, birds, and plants (Sparks A. et al., Immunobiology, 2017, PMID: 27780588). We have included this information in Introduction of the revised manuscript (Page 3) as following:

A recent study identified microphage migration inhibitory factor (MIF), a previously known pleiotropic cytokine-like protein highly conserved in mammals, as a novel poly(ADP-ribose) polymerase 1 (PARP1) associated nuclease (PAAN), which possesses a Mg²⁺- and Ca²⁺-dependent 3' exonuclease activity¹⁹.

7. Fig.1H-K – MIF does not look like highly active nuclease toward the substrate tested. It takes 10 minutes to cleave once half of the substrate with an excess of enzyme. How does it compare to other nucleases operating at replication fork like Fen1?

Response: We carefully re-analyzed MIF nuclease activity by performing additional experiments with various concentrations of DNA substrates under short time duration (1 min, 5 min, 10 min). We found that MIF's $t_{1/2}$ is 7.7-10.8 min, K_M varies from 700 nM to 900 nM, and k_{cat} is about 0.02/min. These biochemical parameters are comparable to those of *Thermus thermophilus* MutS2 (ttMutS2), whose K_M is 290 nM and k_{cat} is 0.041/min (Fukui K. et al., Nucleic Acids Res, 2007, PMID: 17215294). MutS2 is known to play an important role in mismatch repair. In addition, MIF's $t_{1/2}$ is also comparable to many other well-recognized nucleases including Fen1, ExoN, ribozyme, ribozyme and EcoRI (Brown KR et al., DNA Repair, 2002, PMID: 12531027; Herschlag D, Cech TR. Nature, 1990, PMID: 1690858; Hicks MR. et al., Biochemistry, 2006, PMID: 16846234; Hutton RD et al., Nucleic Acids Res, 2008, PMID: 18948279; Tsutakawa SE, et al., Cell, 2011, PMID: 21496641; Zhao G. et al., PLoS One, 2013, PMID: 24194862). For example, Fen1's $t_{1/2}$ varies from milliseconds by measuring the Fen1/DNA binding affinity (Negritto MC. et al., Mol Cell Biol. 2001, PMID: 11259584) to 30 min by measuring the actual cleavage rate (Tsutakawa SE, et al., Cell, 2011, PMID: 21496641). EcoRI's $t_{1/2}$ is 9.7 min calculated by UV linear dichroism (Hicks MR. et al., Biochemistry, 2006, PMID: 16846234) and about 8 min monitored by fluorescence changes (Zhao G. et al., PLoS One, 2013, PMID: 24194862). Such difference observed for the same nuclease from different studies are likely due to the difference of the substrates, experiment conditions, functional readout (binding vs cleavage), or the sensitivity of P³²-, fluorescence-, biotin-, or Ethidium Bromide-based detection. Although MIF's enzyme kinetics might be slower than some other nucleases, we do not exclude a possibility that the additional enhancers may help MIF bind and cleave DNA efficiently *in vivo*, which requires further investigation. Overall, MIF is an important nuclease involved in DNA replication. We have included new results in Fig.1h-k and supplementary Fig.1b-e and described in the revised

manuscript (page 6-7). We also included new discussion in the revised manuscript (page 24-25) as following:

We further studied the nuclease kinetics of MIF by incubating purified MIF protein for 1-60 min with 1 nt overhanged Y-shaped dsDNA at concentrations varying from 0.2 μ M to 4 μ M. We found that MIF cleaved away 1 nt overhang at the 3' end of Y-shaped dsDNA (DS-1 nt) in a time-dependent manner with a $t_{1/2}$ of 7.7 min, and also in a concentration-dependent manner with an affinity for the substrate (K_M) of 0.9 μ M and the catalytic efficiency (k_{cat}) of 0.02/min (Fig. 1h-k). Biotin-labelled Y-shaped dsDNA (DS-1b) was similarly tested. MIF cleaved DS-1b in a time- and concentration-dependent manner with a $t_{1/2}$ of 10.8 min, K_M of 0.7 μ M and k_{cat} of 0.02/min (Supplementary Fig. 1b-e).

*Although the kinetic analysis of MIF activity revealed a relatively slow reaction with $t_{1/2}$ at 7.7-10.8 min, it is comparable to some well-recognized nucleases including MutsS2, Fen1, exoribonuclease (ExoN), ribozyme, and EcoRI^{42, 43, 44, 45, 46, 47, 48}. For example, the nuclease activity of *Thermus thermophilus* MutS2 (ttMutS2), which is known to play an important role in MMR, has been reported to have a K_M at 290 nM and k_{cat} at 0.041/min⁴³, which are comparable to those of MIF. The $t_{1/2}$ of Fen1 varies from milliseconds to 30 min depending on measuring the Fen1/DNA binding affinity⁴⁹ or the actual cleavage rate⁴⁷, respectively. The restriction nuclease EcoRI's $t_{1/2}$ is 9.7 min calculated by UV linear dichroism⁴⁵ and about 8 min monitored by fluorescence changes⁴⁸. As with many nucleases, it is possible that the additional enhancer may help MIF bind and cleave DNA efficiently in vivo, which requires further investigation.*

8. Fig. 2G Please add WT Poldelta with MIF – this is important control to show if the presence of MIF further reduces mutation rates.

Response: As suggested, we performed gap-filling assay in the presence of WT Pol δ and MIF and found that MIF did not further increase Pol δ -mediated fidelity of DNA synthesis. This is not a surprise as the mutation rate of WT Pol δ is very low. These results have been shown in the revised Fig. 2f-h and described in the revised manuscript (page 10) as following:

We also examined the effect of MIF-WT on Pol δ -mediated mutagenesis and found that MIF did not further reduce the mutation frequency induced by Pol δ (Fig. 2f-h).

9. In fig 1. Is the hairpin present in all substrates? Is this hairpin needed for nuclease to cleave 3' strands, why the little scheme in Fig 1D does not show hairpin while Fig. 1A SL-7 nt shows it?

Response: Fig. 1a SL-7nt shows single-stranded DNAs with hairpin loop structure as the MIF substrate, which was identified in the previous study (Wang Y. et al., Science 2016, PMID: 27846469). In the present study, we found that MIF can also recognize and cleave the double-stranded DNA with Y-shaped structure as shown in Fig. 1d, which mimics the intermediates of DNA replication. Our *in vitro* nuclease assay showed that the hairpin loop structure in Y-shaped DNA substrates is not required for MIF recognition and cleavage (Fig.1a-e). We have clarified these results in the Result section of the revised manuscript (page 5) as following:

To study whether MIF excises mis-incorporated nucleotides at the 3' end of Y-shaped dsDNA, which mimics the intermediate products during DNA replication, we designed a series of "damaged" dsDNA substrates based on its stem loop (SL) ssDNA substrate identified recently¹⁹ but removed the loop and altered the length of mismatched nucleotides at the 3' end with or without biotin labeling for in vitro nuclease assay (Fig. 1a). We found that MIF cleaved away unpaired nucleotides varying from 1 to 7 nt at the 3' end of non-biotin-labeled dsDNA substrates, which had the Y-shaped structure but no loop structure (Fig. 1b, c).

Minor points

10. “Although polymerases δ and ϵ have been thought to be responsible for the removal of misincorporated nucleotides in a 3'→5' direction, increasing numbers of germline mutations as well as somatic mutations of polymerases δ and ϵ , which lack 3' exonuclease activity has been identified in human cancers including colorectal cancer, breast cancer and glioblastoma 14.” This sentence is unclear. What do authors want to say here? The fact that polymerases delta and epsilon use their exonuclease activity to remove misincorporated nucleotides is not questionable, this is clear from literature and all organisms ever tested. The fact that there are cancer cells that lost these activities does not negate the fact that these exonuclease activities are very important.

Response: As suggested, we acknowledged the importance of the 3' exonuclease activity of Pol δ and ϵ in DNA proofreading. We have rephrased this sentence in the revised manuscript (page 3) to avoid the confusion as following:

Pol δ and Pol ϵ have been well recognized for their functions in removal of mis-incorporated nucleotides in a 3'→5' direction. However, it is unknown if additional 3' nucleases are required to cooperate with nuclease-proficient Pol δ and Pol ϵ to maintain high speed elongation, or to proofread DNA elongated by Pol α or other 3'→5' exonuclease-deficient polymerases like translesion DNA polymerases or mutant Pol δ and Pol ϵ identified in certain human cancers including colorectal cancer, breast cancer and glioblastoma¹⁴.

11. “It is a long mystery in the field what 3' flap nuclease helps cancers resolve 3' flap structures to overcome deleterious consequences resulting from high levels of DNA replication stress for their rapid proliferation.” Unclear what the authors want to say. Which 3' flap nuclease they refer to? MIF? Some unknown nuclease? Is it clear that this activity was needed to be identified and if so based on what exactly observations? It is confusing.

Response: We have rephrased this sentence in the revised manuscript (page 3) to avoid the confusion as following:

Pol δ and Pol ϵ have been well recognized for their functions in removal of mis-incorporated nucleotides in a 3'→5' direction. However, it is unknown if additional 3' nucleases are required to cooperate with nuclease-proficient Pol δ and Pol ϵ to maintain high speed elongation, or to proofread DNA elongated by Pol α or other 3'→5' exonuclease-deficient polymerases like translesion DNA polymerases or mutant Pol δ and Pol ϵ identified in certain human cancers including colorectal cancer, breast cancer and glioblastoma¹⁴.

12. “In this study, we identified MIF as a hitherto unrecognized 3' flap nuclease. MIF is translocated from the cytosol into the nucleus in S phase and recruited to DNA replication sites by PARP1, where MIF recognizes and cleaves mis-incorporated nucleotides from Y-shaped dsDNA at the 3' end during DNA replication.” For clarity I suggest separating any discussion of in vitro studies with purified MIF from studies that are done in cells. The authors show that MIF is present at replication forks, and that purified MIF can cleave few nucleotides at 3' strands at Y structures but it is not demonstrated that in cells MIF removes mis incorporated nucleotides.

Response: We performed in vivo HPRT mutation assays to analyze the effects of MIF knockout on mutation frequency, especially mis-incorporation rate, in Pol δ -proficient cancer cells as well as Pol δ -deficient DLD1 cells, and found that MIF knockout significantly increased the mis-incorporation mutation rate in both Pol δ -proficient and -deficient cancer cells (Fig. 3i-l and supplementary Fig. 3l-o). These results suggest that MIF removes mis-incorporated nucleotides *in vivo*, consistent with our *in vitro*

findings (Fig. 2 and supplementary Fig. 2). On the other hand, we also rephrased the sentence that was pointed out by the reviewer (page 4) as well as the related discussion as suggested to make the conclusion clear and well-supported by our evidence. We have included these new HPRT results (page 12-13) and rephrased the sentence in the revised manuscript as following:

In the present study, we identified MIF as a hitherto unrecognized 3' flap nuclease involved in DNA replication. MIF recognizes and cleaves mis-incorporated nucleotides from Y-shaped dsDNA at the 3' end. Genetic depletion of MIF or inhibition of its nuclease activity in cancer cells significantly increases mutation frequency, reduces DNA synthesis, causes cell cycle delay, inhibits colony survival, and attenuates the growth of breast tumors, glioblastoma and colon tumors in mice. These findings uncover that MIF proofreads mis-incorporated nucleotides during DNA replication as a novel 3' flap nuclease to promote tumor growth.

The hypoxanthine-guanine phosphoribosyltransferase (HPRT) gene on the X chromosome has been used as a model gene to investigate mutability in mammalian cell lines^{30, 31}. To further explore the effect of MIF on replication fidelity, we established POLD1 knockdown (KD) and POLD1 KD/MIF KO MDA-MB-231 cells using CRISPR/Cas9 sgRNAs (Supplementary Fig. 3l) and performed HPRT mutation assay following 6-TG treatment (20 μ M, 12 days). We found that the HPRT mutation rate was very low (2.6×10^{-7}) in Pol δ -proficient MDA-MB-231 cells (Fig. 3i), whereas MIF KO increased the HPRT mutation rate (1.43×10^{-6}), which was mainly due to the nucleotide misincorporation (Fig. 3i, j). Since POLD1 KD MDA-MB-231 cells were not able to form colonies for HPRT assay (Supplementary Fig. 3m), we instead established MIF KO in colorectal cancer cell line DLD1 cells, which have Pol δ mutations R689W and R506H in the conserved DNA polymerase III and exonuclease III motifs, respectively³². The overall mutation frequency in Pol δ -deficient DLD1 cells was about 8.68×10^{-5} , which was increased by 1.6-fold in MIF KO cells (1.4×10^{-4} , Fig. 3k). It is of note that the nucleotide misincorporation rate in parental DLD1 cells was quite low (7.1×10^{-6}) and MIF KO significantly increased the misincorporation rate by 13.6-fold (9.65×10^{-5} , Fig. 3l). MIF KO DLD1 cells expressing WT MSH6 (DLD1+MSH6) were also established, as DLD1 cells are deficient in MMR due to the mutation in MSH6 (Supplementary Fig. 3n, o). Restoring MSH6 expression in DLD1 cells reduced the overall mutation rate to 1.55×10^{-5} , which is expected. However, MIF KO still significantly increased the overall mutation frequency (1.0×10^{-4} , Fig. 3k). Notably, the misincorporation rate in the MMR-proficient DLD1 cells was only about 3.5×10^{-6} , which was increased by 19-fold in MIF KO cells (6.67×10^{-5} , Fig. 3k). These findings indicate that MIF has a significant impact on correcting nucleotide misincorporation in vivo.

Reviewer #2

1. This manuscript describes *in vitro* and *in vivo* studies of the migration inhibitory factor (MIF). It is claimed that the studies indicate that MIF is a structure-specific 3' flap nuclease that removes mismatched 3'-nucleotides during DNA replication and that the protein is upregulated in cancer and therefore a new chemotherapy target. MIF is previously well-characterised as a phenylpyruvate tautomerase, so this would be a novel and highly significant finding. Indeed, the initial association of MIF with (at that stage unspecified type of) nuclease activity was published in Science.

Response: We thank the referee for the positive comments “a novel and highly significant finding”.

2. I do have potential serious concerns about the conclusions reached in this manuscript. Undoubtedly, the samples of “purified” protein used in this study contain a 3'-nuclease as evidenced by the gels in figure 1. However, the catalytic parameters reported for the enzyme make me nervous that a contaminant is a reasonable explanation for the observed activity. The half-life for the reaction putatively catalysed by MIF is reported to be 10.3 min, whereas typically this figure would be in the order of 10-500 ms for a nuclease, so the reaction observed here is at least 10,000 slower than might normally be expected. As a caveat it needs to be acknowledged that there are examples of excepted nucleases that do have much longer half-lives; these are generally activated by a protein partner *in vivo*. There are also examples of others that have slow multiple turnover rates because they bind product strongly; it is not clear from the manuscript whether multiple or single turnover conditions were used for this measurement. Showing lack of other bands on a typically loaded stained gel is not evidence of lack of contaminating activity; this could be present in very small amounts (e.g. 1/10,000) and still give the level of activity reported here.

Response: To address the reviewer’s concern regarding the possible contamination, we have utilized multiple different approaches to confirm the purity of our MIF protein preparation. Besides Coomassie blue staining (Fig. 1f), we performed mass spectrometry to analyze our MIF protein preparations used in our *in vitro* nuclease assay. As shown in Supplementary Table 1 in the revised manuscript, no known nucleases were identified by mass spectrometry, although 4 additional proteins with very low abundance close to the background were detected. The negative controls GST and MIF E22A mutant, which were purified using the exactly same system and method as MIF shown in Fig. 1g, do not show any nuclease activities, which also strongly supports the good purity of our MIF protein and excludes a possible nuclease contamination. Our *in vitro* nuclease assay were conducted with many different batches of MIF protein preparations and we observed the reproducible results. Moreover, the previous study (Wang Y et al., 2016 Science, PMID: 27846469) and current study both showed that MIF selectively cleaves single-stranded DNA with hairpin loop structure as well as double-stranded DNA with Y-shaped structure. Even if contaminated, non-specific protein unlikely cleaves various substrates with different structures with such strong specificity and selectivity. Moreover, MIF proteins purchased from three different companies (Shenandoah, Novus, R&D) all showed similar 3' nuclease activity to cleave the 3' unpaired nucleotide labeled with fluorescein (Fig. 1 in this letter). The cleavage was detected by both Ethidium Bromide (EtBr) and more obviously by the fluorescence staining, which was not detected in GST or DNA substrate only groups (Fig. 1 in this letter). The variable MIF nuclease activity among different sources might be due to the use of different buffer and/or purification system (Fig. 1 in this

letter). Overall, we are very confident that the nuclease activity we observed is specifically attributed to MIF.

On the other hand, we have carefully re-analyzed MIF's kinetic parameters according to the reviewer's comments with various concentrations of DNA substrates under short time duration (1 min, 5 min, 10 min). We found that MIF's $t_{1/2}$ is 7.7-10.8 min, K_M varies from 700 nM to 900 nM, and k_{cat} is about 0.02/min. These biochemical parameters are comparable to those of *Thermus thermophilus* MutS2 (ttMutS2), whose K_M is 290 nM and k_{cat} is 0.041/min (Fukui K et al., Nucleic Acids Res, 2007, PMID: 17215294). MutS2 is known to play an important role in mismatch repair. In addition, MIF's $t_{1/2}$ is also comparable to many other well-recognized nucleases including Fen1, ExoN, ribozyme, ribozyme and EcoRI (Brown KR et al., DNA Repair, 2002, PMID: 12531027; Herschlag D, Cech TR. Nature, 1990, PMID: 1690858; Hicks MR. et al., Biochemistry, 2006, PMID: 16846234; Hutton RD et al., Nucleic Acids Res, 2008, PMID: 18948279; Tsutakawa SE, et al., Cell, 2011, PMID: 21496641; Zhao G. et al., PLoS One, 2013, PMID: 24194862). For example, Fen1's $t_{1/2}$ varies from milliseconds by measuring the Fen1/DNA binding affinity (Negritto MC. et al., Mol Cell Biol. 2001, PMID: 11259584) to 30 min by measuring the actual cleavage rate (Tsutakawa SE, et al., Cell, 2011, PMID: 21496641). EcoRI's $t_{1/2}$ is 9.7 min calculated by UV linear dichroism (Hicks MR. et al., Biochemistry, 2006, PMID: 16846234) and about 8 min monitored by fluorescence changes (Zhao G. et al., PLoS One, 2013, PMID: 24194862). These differences observed for the same nuclease from different studies may be due to the difference of the substrates, experiment conditions, or the sensitivity of P³²-, fluorescence-, biotin- or Ethidium Bromide-based detection. Although MIF's enzyme kinetics might be slower than some other nucleases, as the reviewer pointed out, the additional enhancer may help MIF bind and cleave DNA efficiently *in vivo*, which requires further investigation. Overall, MIF is an important nuclease involved in DNA replication. We have included new results in Fig.1h-k and supplementary Fig.1b-e and described in the revised manuscript (page 6-7). The new discussion has been also included in the Discussion section of the revised manuscript (page 24-25) as following:

We further studied the nuclease kinetics of MIF by incubating purified MIF protein for 1-60 min with 1 nt overhanged Y-shaped dsDNA at concentrations varying from 0.2 μ M to 4 μ M. We found that MIF cleaved away 1 nt overhang at the 3' end of Y-shaped dsDNA (DS-1 nt) in a time-dependent manner with a $t_{1/2}$ of 7.7 min, and also in a concentration-dependent manner with an affinity for the substrate (K_M) of 0.9 μ M and the catalytic efficiency (k_{cat}) of 0.02/min (Fig. 1h-k). Biotin-labelled Y-shaped dsDNA (DS-1b) was similarly tested. MIF cleaved DS-1b in a time- and concentration-dependent manner with a $t_{1/2}$ of 10.8 min, K_M of 0.7 μ M and k_{cat} of 0.02/min (Supplementary Fig. 1b-e).

*Although the kinetic analysis of MIF activity revealed a relatively slow reaction with $t_{1/2}$ at 7.7-10.8 min, it is comparable to some well-recognized nucleases including Muts2, Fen1, exoribonuclease (ExoN), ribozyme, and EcoRI^{42, 43, 44, 45, 46, 47, 48}. For example, the nuclease activity of *Thermus thermophilus* MutS2 (ttMutS2), which is known to play an important role in MMR, has been reported to have a K_M at 290 nM and k_{cat} at 0.041/min⁴³, which are comparable to those of MIF. The $t_{1/2}$ of Fen1 varies from milliseconds to 30 min depending on measuring the Fen1/DNA binding affinity⁴⁹ or the actual cleavage rate⁴⁷, respectively. The restriction nuclease EcoRI's $t_{1/2}$ is 9.7 min calculated by UV linear dichroism⁴⁵ and about 8 min monitored by fluorescence changes⁴⁸. As with many nucleases, it is possible that the additional enhancer may help MIF bind and cleave DNA efficiently *in vivo*, which requires further investigation.*

3. The whole manuscript hinges on whether this is a bona fide 3'-nuclease as this observation is key to interpretation of all the subsequent experiments. The only evidence that is presented here that in part supports a real enzyme activity for MIF is the lack of (all) activity of a mutant-I would like to see a mutant with reduced activity to be more confident that this really is a nuclease. Moreover, I would like to see some structural analysis that convinces this protein has a nuclease active site. I understand that by sequence alignment it is claimed that MIF has a restriction endonuclease (RE) active site; I would like to

see an overlay of the MIF protein and RE active site-the C-alphas should occupy almost identical positions to conserve catalytic activity. Furthermore, if MIF is a nuclease it would be expected to bind nucleic acid (in the presence of EDTA to prevent reaction); thus, some evidence that this is the case would be preferable. It is also possible to run protein gels containing DNA to detect bands associated with nuclease activity; if MIF is a bona fide nuclease then this should be exclusively the MIF band (example in *Nucleic Acids Research*, 35, 4094–4102, <https://doi.org/10.1093/nar/gkm396>). More evidence needs to be sought that MIF really is a nuclease.

Response: As suggested, we actually included the data from a catalytically-inactive MIF mutant E22A that reduces MIF nuclease activity (but not completely abolish its nuclease activity) throughout our manuscript (Fig. 1g, Fig. 2d-h, Fig. 3b-h, Fig. 6b-I, Fig. 7a-I, and Supplementary Fig. 2d,e,h,I, S Fig. 3f-k, Fig. 4a, Fig. 6a-d in the revised manuscript). We have also performed systematically detailed structural analysis of MIF (Figs. 2-3 in this letter) and overlapped it to other classic type II restriction endonucleases (Fig. 2 in this letter). We found that the MIF trimer indeed contains the typical nuclease core structure and it can be nicely overlapped with multiple classic endonuclease including EcoRI, EcoRV, ExoII and PvuII (Figs. 3 in this letter). We also tested the binding of MIF to DNA by *in vivo* ChIP-seq (Fig. 4 in this letter) and *in vitro* EMSA assay (Figs. 5 and 6 in this letter). We found that MIF binds to the 5' free arm of the hairpin loop structure, which can be blocked by either MIF antibody (Fig. 5 in this letter) or removing 5' free arm (Fig. 6 in this letter). Since Y-shaped DNA shares very similar features with the hairpin structured DNA except the loop that is not required for DNA binding or cleavage as shown in our recent study (Wang Y et al., 2016 *Science*, PMID: 27846469) as well as the current study. Therefore, we did not further pursue MIF-DNA binding. All the structural analysis has been shown in our recent publication when MIF was first discovered as a novel nuclease (Wang Y et al., 2016 *Science*, PMID: 27846469). For the reviewer's convenience, we have also included part of the detailed data from our recent publication (PMID: 27846469) in order to address the reviewer's concerns in this response letter as following:

The core PD-D/E(X)K topology structure of nucleases including EcoRI and EcoRV consists of 4 β -strands next to two-helices (19). Two of the β -strands are parallel to each other whereas the other two are antiparallel. Previous 3-D crystal structures of MIF indicate that it exists as a trimer (22-24). The trimeric structure of MIF enables the interaction of the β -strands of one monomer with the other monomers resulting in a PD-D/E(X)K structure that consists of 4 β -strands next to 2 α -strands (Fig. 2 and Fig. 3, D to G). Two of the β -strands (β -4 and β -5) are parallel whereas the other two strands (β -6 and β -7) (from the adjacent monomer) are anti-parallel (Fig. 3, D to G). The topology structure of PD-D/E(X)K motifs with orientations of the beta-strands relative to the alpha helices in the MIF trimer are very similar to EcoRV, a well characterized endonuclease (Fig. 3, H to K). Importantly, the PD-D/E(X)K motif based on the trimer structure of MIF is structurally similar to type II ATP independent restriction endonucleases, such as EcoRI and EcoRV, as well as, ExoIII family purinic/apyrimidinic (AP) endonucleases, such as ExoIII (Fig. 2 and fig. 3, L to N). Moreover, MIF also has a similar topology to the PvuII endonuclease and its β -7 strand is of similar size to PvuII endonuclease β -strand at the same position in its PD-D/E(x)K motif (fig. 3O). These 3-D modeling results taken together indicate that MIF belongs to the PD-D/E(X)K nuclease-like superfamily (25, 26).

Fig. 3. MIF contains PD-D/E(x)K nuclease motif. (A-B) Alignments of MIF's nuclease and CxxCxxHx(n)C domains. (C) Conserved topology of the active site in PDD/E(x)K nucleases modified from Kosinski et al., (19). (D) Crystal structure of MIF trimer (pdb:1GD0). (E) Topology of MIF trimer illustrating the orientations of the various domains similar to PD-D/E(x)K motif. (F) Crystal structure of the MIF monomer containing the PD-D/E(x)K domain derived from the trimer (broken red line in D). (G) Topology of a MIF monomer in the MIF trimer. (H) Illustration that each monomer has a PD-D/E(x)K domain, which is made of two parallel β -strands ($\beta 4$ and $\beta 5$) from one monomer and two anti-parallel strands ($\beta 6$ and $\beta 7$) from the adjacent monomer. (I) A schematic diagram of the similarity in topology of the MIF and EcoRV illustrating similar orientations of the various domains in their nuclease domains. (J) Topology of EcoRV monomer. (K) Alignment of MIF and EcoRV monomer (red). (L-O) Alignments of PD-D/E(x)K motif in MIF and other well-known nucleases including EcoRI (magenta, pdb: 1QC9), EcoRV (light blue, pdb: 1SX8), ExoIII (red, pdb: 1AK0), and PvuII (orange, pdb 1PVU). All five motifs show similar orientations of the four beta strands in the beta-sheet against the alpha helices as observed in a typical PD-D/E(x)K motif active site. Adopted from Fig. S3 (Wang Y et al., 2016 Science).

To determine the characteristics of DNA sequences bound by MIF in an unbiased manner, HeLa cells were treated with dimethyl sulfoxide (DMSO) or MNNG (50 mM, 15 min), followed by anti-MIF chromatin immunoprecipitation (ChIP) assays and deep sequencing (Fig. 4). The quality of sheared genomic DNA and the specificity of ChIP using the MIF antibody was tested and confirmed (Fig. 4, A and B). After excluding overlapped peaks in the DMSO-treated samples, 0.1% of total mapped reads exhibit MIF peaks after MNNG treatment (Fig. 4C). MIF preferentially binds to the promoter and 5' UTR regions after MNNG treatment (Fig. 4D). The representative IGV visualization of MIF enrichment on the genome is shown in two different window sizes (250 kb (Fig. 4E) and 50 kb (Fig. 4F)). The average distance intervals between MIF peaks are about 15 to 60 kb, which is consistent with size of DNA fragments observed via pulse-gel electrophoresis during parthanatos. ChIP-qPCR further confirms that MIF binds to the peak regions at 55101, 66005, 65892, 36229, 46426 and 62750 but it does not bind to the non-peak regions after MNNG treatment (Fig. 4G). We used the multiple Em for motif elicitation (MEME) program, which performs comprehensive motif analysis on large sets of nucleotide sequences (32), and we identified two classes of MIF-binding motifs (Fig. 5A). The first class (sequences 1 through 3) represents a highly related family of overlapping sequences (Fig. 5A). The sequence features of this family are best captured in sequence 1 with 30 nucleotides and designated PS30, the most statistically significant motif identified, as determined by the MEME program (E -value = $1.4e-051$) (Fig. 5A).

We performed 3D modeling to determine likely points of DNA interaction with MIF's PD-D/E(X)K motif. Within the PD-D/E(X)K motif, P16 and D17 on MIF are predicted to be positioned close to double-stranded DNA (dsDNA), whereas E22 is close to ssDNA (unpaired arm), indicating MIF might bind ssDNA, dsDNA, or both (Fig. 5B). We

examined both single-stranded and double stranded forms of MIF DNA substrates for MIF binding and cleavage specificity. We synthesized the ssPS30 sequence with a 5' biotin label and subjected it to an electrophoretic mobility shift assay (EMSA) (Fig. 5C). MIF bound to the biotin-labeled ssPS30, forming one major complex in the presence of 10 mM Mg²⁺ (Fig. 5C), which was completely disrupted by the addition of excess unlabeled DNA substrate (PS30) or a polyclonal antibody to MIF (Fig. 5C). MIF E22Q, E22A, P16A, P17A, and P17Q mutants still formed MIF/ssPS30 complexes (Fig. 5C). Because ssPS30 has the potential to form a stem loop structure with unpaired bases at the 5' and 3' ends, we tested whether MIF binds to ssDNA with sequence or structure specificity. We used 5' biotin labeled ssPS30 and sequence-related substrates with different structures created by removing unpaired bases at the 5' end, 3' end, or both 5' and 3' ends, or by eliminating the stem loop in the EMSA (Fig. 6). Completely removing the 3' unpaired bases (5'bLF) had no effect on the DNA MIF complex formation (Fig. 6). In contrast removing the 5' unpaired bases (5'bRF) reduced but did not abolish DNA-MIF binding. Similar results are observed when both 5' and 3' unpaired bases were removed (5'bSL). Thus, MIF appears to mainly bind to 5' unpaired bases in ssDNA with stem-loop structures. We also used a poly(A) sequence that has no stem loop (5'bPA30) and a short poly(A) sequence at the 5' end of a stem-loop structure (5'b3F1) as the substrates. MIF failed to bind to 5'bPA30 but did bind to 5'b3F1. These results indicated that 5' free arm is required for MIF-DNA binding (Fig. 6). We also tested a substrate unrelated in sequence but that had a stem loop-like structure (5'bL3). MIF bound weakly to 5'bL3. But its binding efficiency was much lower than that of 5'bPS30. These data indicate that MIF

Fig. 4. Characterization of MIF-DNA binding by ChIP-seq. (A) Sonicated fragments of chromatin for ChIP-seq in the DMSO and MNNG (50 μM) treated cells. (B) Representative immunoblot images of MIF ChIP. (C) Number and coverage of the reads from four different libraries. (D) MIF ChIP-peak distribution across different genomic regions in MNNG treated cells. (E-F) Representative IGV visualization of MIF enrichment on the genome. The top two lines show the tdf file. The third and fourth lines show the bed files. (G) MIF chromatin enrichment in DMSO and MNNG treated cells confirmed by qPCR. Adopted from Fig. S6 (Wang Y et al., 2016 Science).

Fig. 5. MIF binds to single stranded DNA. (A) Alignment of MIF DNA binding motif. (B) Images of MIF trimer (PDB accession 1FIM) surface showing a groove/binding pocket (arrows) (Top panel). Models of MIF trimer with dsDNA in the groove (Middle panel). Right image in the middle panel shows the side view of the overlay of MIF-dsDNA (PDB accession 1BNA) with MIF-ssDNA (PDB accession 2RPD) models. i-iii, Cartoon images showing residues P16 and D17 close to dsDNA and ssDNA whereas E22 is close to the ssDNA but not the dsDNA. (C) Binding of MIF to the single strand 5' biotin labeled DNA binding motif (PS30) as determined by EMSA. Left panel, binding of MIF in the presence or absence of Mg²⁺. Middle panel, binding of MIF in the presence of MIF antibody and controls. Right panel, binding of MIF in comparison to MIF mutants. Arrow indicates DNA/MIF protein complex. Asterisk (*) indicates nonspecific bands. Experiments were replicated for 4 times using MIF protein purified from 3 independent preparations. Adopted from Fig. S7 (Wang Y et al., 2016 Science).

preferentially binds to ssDNA with a stem loop and that its specificity is not entirely determined by the sequence. We also tested whether MIF bound to dsDNA with PS30; poly(A); substrates with sequence similarity to PS30 (5'bPS30, 5'bSL, 5'bLF, 5'bRF, 5'bPA30, and 5'bPA5E); and others with nonrelated sequences (PCS and 5'bL3) (Fig. 6). MIF failed to bind to any of these double-stranded substrates (Fig. 6).

4. A lesser concern is that I have reservations that some of the kinetic analyses were not carried out under appropriate conditions. To determine Michaelis Menten parameters, the concentration of enzyme should be significantly lower than all the substrate concentrations. Moreover, analyses that rely on single data points are fraught with errors so a number of data points that fit to a straight line should be obtained to measure the initial rate. The conditions under which half-lives are determined must be specified-usually they would be where $[S] < \text{or} = K_m$ and $[E] \gg K_m$ to give the maximal rate without turnover. Furthermore, as the authors know the concentration of MIF they should have been able to determine k_{cat} and shouldn't need to quote V_{max} ($=k_{cat} \times [E]$). In several places it's not clear what concentrations of reagents have been used and precisely what methods have been employed. It is also not clear what DS-1b is-there are several different ways of attaching 3'-biotin so it is important to know either what reagents were used to create to substrate or its full final structure. A minor point- K_m has a capital K (equilibrium constant).

Response: We thank the referee for the detailed comments, which helps improve our manuscript. We have also consulted with Dr. Elliott Ross, an expert in enzyme kinetics, to improve our assay conditions. According to the suggestions, we have carefully modified our experiment conditions including adjusting concentrations of the substrate and MIF and increasing numbers of the short time points. We have also calculated $t_{1/2}$ and k_{cat} as suggested and provided detailed methods and substrate and enzyme concentrations throughout the revised manuscript. We have also corrected k_m to K_M in the revised manuscript. DS-1b DNA is a biotin-labeled DS-1nt whose sequence and structure were shown on Fig. 1a Biotin was attached to the last unpaired nucleotide at the 3' end using a C6 spacer linker performed by IDT. We have included this information in Method (page 32). We have also included new results in Fig. 1h-k and supplementary Fig. 1b-e and described in the revised manuscript (page 6-7) as following:

We further studied the nuclease kinetics of MIF by incubating purified MIF protein for 1-60 min with 1 nt overhanged Y-shaped dsDNA at concentrations varying from 0.2 μM to 4 μM . We found that MIF cleaved away 1 nt overhang at the 3' end of Y-shaped dsDNA (DS-1 nt) in a time-dependent manner with a $t_{1/2}$ of 7.7 min, and also in a concentration-dependent manner with an affinity for the substrate (K_M) of 0.9 μM and the catalytic efficiency (k_{cat}) of 0.02/min (Fig. 1h-k). Biotin-labelled Y-shaped dsDNA (DS-1b) was similarly tested. MIF cleaved DS-1b in a time- and concentration-dependent manner with a $t_{1/2}$ of 10.8 min, K_M of 0.7 μM and k_{cat} of 0.02/min (Supplementary Fig. 1b-e).

Reviewer #3

1. In this manuscript, the authors describe the role of macrophage migration inhibitory factor (MIF) in the proofreading process of misincorporated nucleotides during DNA replication. MIF has been shown to possess nuclease activity on stem loop DNA structures and play a key role in the DNA fragmentation process in the cell death pathway initiated by PARP1 activation. The authors show that MIF displays 3' exo- and 3' endo-nuclease activities in vitro on unpaired 3' DNA ends that mimic nucleotide misincorporation during DNA synthesis. MIF KO cells show DNA replication defects, DNA damage, and abnormal chromosomes. The authors find that MIF interacts with PCNA and PARP1, which they propose to play a role in the recruitment of MIF to replication sites containing misincorporated nucleotides. Finally, the authors use xenograft models to demonstrate that MIF, which is often overexpressed in various tumors, can promote tumor growth. The proposed model that MIF proofreads misincorporated DNA and thereby mitigates replication stress and promotes tumor growth is novel.

Response: We thank the referee for the detailed summary and the positive comment that “The proposed model that MIF proofreads misincorporated DNA and thereby mitigates replication stress and promotes tumor growth is novel”.

2. However, the study appears to be based on the weak premise that cancer cells experience elevated DNA replication stress due to nucleotide misincorporation. Because DNA misincorporation is generally believed to cause mutagenesis rather than replication stress, the authors need to provide more evidence that the elevated fork stalling in MIF KO cells is caused by nucleotide misincorporation. In addition, the study lacks insight into the molecular mechanisms. The roles of PCNA, PARP1 and PARP1 activity in MIF recruitment to replication sites have not been developed sufficiently. Furthermore, there are a number of technical issues that are likely to impact their interpretation of the data. In summary, this manuscript presents interesting and potentially important observations. However, the study does not have solid data and proper interpretations to support their conclusions.

Response: We thank the referee for agreeing that the manuscript presents interesting and potentially important observations. In this revision, we have performed HPRT mutation assays in Pol δ -proficient MDA-MB-231 and Pol δ -deficient DLD1 cells as suggested (Fig. 3i-l, supplementary Fig. 3i-o) and found that MIF indeed has a significant impact on nucleotide misincorporation in cancer cells (please see detailed data description in the response to Points 3 and 5). We also performed substantial new experiments to further help understand the molecular mechanism of MIF recruitment to replication sites. The new experiments addressed if MIF directly interacts with PARP1 or PCNA (Supplementary Fig. 5f) (see the response to Points 7 and 17), which PARP1 domain binds to MIF (Supplementary Fig. 5g, h), if MIF is PARylated (Supplementary Fig. 5i, see point 10), and how PARylation affects MIF-PARP1 and MIF-PCNA interaction (Fig. 5h-l and Supplementary Fig. 5i, j, see point 11), which hopefully strengthened our understanding of the molecular mechanism of MIF recruitment to the replication sites. We also addressed technical issues raised by the reviewer. Please refer to details in the following responses to the specific points raised by the reviewer.

Besides performing the new experiments, here we also provided additional interpretation/discussion based on the literatures to strength the premise of this study, which was summarized as following:

Although nucleotide misincorporation is able to cause mutagenesis as the reviewer mentioned, it has also been well recognized as one of the main causes for replication stress, which has been well summarized in many review articles including “Causes and consequences of replication stress” by Zeman MK and Cimprich KA (Nat Cell Biol, 2014), “The impact of replication stress on replication dynamics and DNA

damage in vertebrate cells” by Techer H et al. (Nature, 2017); and “Exploiting DNA replication stress for cancer treatment” by Ubhi T and Brown GW (Cancer Research, 2019). Pol δ and Pol ϵ are less stringent in discriminating dNTPs and rNTPs. Misincorporation of rNTPs causes replication stress (Zeman MK and Cimprich KA, Nat Cell Biol, 2014). Disruption of the balance between purine and pyrimidine enhances misincorporation and may induce moderate levels of fork slowing (Techer H et al., Nature, 2017). It could be lethal for the cell if misincorporation cannot be corrected properly. However, if cells can survive from replication stress without fixing it, nucleotide misincorporation may eventually cause mutagenesis. Thus, nucleotide misincorporation represents an intermediate status and has a strong link with replication stress.

Major points

3. The proofreading function of MIF was shown only *in vitro*. The authors need to measure *in vivo* mutation frequency in MIF KO cells using established methods such as 6-thioguanine selection, which quantitates spontaneous mutations in the HPRT gene.

Response: As suggested, we performed HPRT mutation assay in Pol δ -proficient MDA-MB-231 cells and Pol δ -deficient (R689W and R506H) DLD1 cells. We found that MIF knockout increased the HPRT mutation rate, especially mis-incorporation mutation, in Pol δ -proficient MDA-MD-231 (Fig. 3i, j in the revised manuscript). Similarly, MIF knockout also increased the overall mutation rate by 1.6-fold and mis-incorporation mutation by 13.7-fold in Pol δ -deficient DLD1 cells (Fig. 3k, l in the revised manuscript). These results indicate that MIF has a proofreading function *in vivo*. These new results have been shown in Fig.3i-3l and sFig.3l-o and described in the revised manuscript (page 12-13) as following:

*The hypoxanthine-guanine phosphoribosyltransferase (HPRT) gene on the X chromosome has been used as a model gene to investigate mutability in mammalian cell lines^{30, 31}. To further explore the effect of MIF on replication fidelity, we established POLD1 knockdown (KD) and POLD1 KD/MIF KO MDA-MB-231 cells using CRISPR/Cas9 sgRNAs (Supplementary Fig. 3l) and performed HPRT mutation assay following 6-TG treatment (20 μ M, 12 days). We found that the HPRT mutation rate was very low (2.6×10^{-7}) in Pol δ -proficient MDA-MB-231 cells (Fig. 3i), whereas MIF KO increased the HPRT mutation rate (1.43×10^{-6}), which was mainly due to the nucleotide misincorporation (Fig. 3i, j). Since POLD1 KD MDA-MB-231 cells were not able to form colonies for HPRT assay (Supplementary Fig. 3m), we instead established MIF KO in colorectal cancer cell line DLD1 cells, which have Pol δ mutations R689W and R506H in the conserved DNA polymerase III and exonuclease III motifs, respectively³². The overall mutation frequency in Pol δ -deficient DLD1 cells was about 8.68×10^{-5} , which was increased by 1.6-fold in MIF KO cells (1.4×10^{-4} , Fig. 3k). It is of note that the nucleotide misincorporation rate in parental DLD1 cells was quite low (7.1×10^{-6}) and MIF KO significantly increased the misincorporation rate by 13.6-fold (9.65×10^{-5} , Fig. 3l). MIF KO DLD1 cells expressing WT MSH6 (DLD1+MSH6) were also established, as DLD1 cells are deficient in MMR due to the mutation in MSH6 (Supplementary Fig. 3n, o). Restoring MSH6 expression in DLD1 cells reduced the overall mutation rate to 1.55×10^{-5} , which is expected. However, MIF KO still significantly increased the overall mutation frequency (1.0×10^{-4} , Fig. 3k). Notably, the misincorporation rate in the MMR-proficient DLD1 cells was only about 3.5×10^{-6} , which was increased by 19-fold in MIF KO cells (6.67×10^{-5} , Fig. 3k). These findings indicate that MIF has a significant impact on correcting nucleotide misincorporation *in vivo*.*

4. Fig. 2A-D: Recombinant PCNA protein needs to be added to the *in vitro* elongation assays. SDS-PAGE gels of the purified polymerases should be included.

Response: As suggested, we included PCNA protein in *in vitro* elongation assays and found that PCNA did not affect DNA elongation. The results have been included in the Supplementary Fig.2c in the revised manuscript. We also included SDS-PAGE gels of the purified Pol α and Pol δ proteins in the revised manuscript (Supplementary Fig. 2f, g). These new results have been described in the revised manuscript (page 8, 9) as following:

*Proliferating cell nuclear antigen (PCNA), which has been shown previously to enhance the nucleotide incorporation rate*²⁷, did not obviously alter the effect of MIF on DNA elongation (Supplementary Fig. 2c).

The protein purity of POLA1 (Pol α) and POLD (Pol δ) was confirmed by Coomassie blue staining (Supplementary Fig. 2f, g).

5. Fig. 3F-H: The authors provided no data to support that the excessive replication fork stalling in MIF KO cells is caused by nucleotide misincorporation.

Response: We performed HPRT mutation assay in Pol δ -proficient MDA-MB-231 cells and Pol δ -deficient DLD1 cells to study the effects of MIF on nucleotide misincorporation in vivo. We found that the HPRT mutation rate is very low in Pol δ -proficient MDA-MB-231 cells (2.6×10^{-7} , Fig. 3i), whereas MIF knockout increased the HPRT mutation rate (1.43×10^{-6}), which is mainly due to the nucleotide misincorporation (Fig. 3i, j). As POLD1 knockdown MDA-MB-231 cells were not able to form colonies for HPRT mutation assay (Supplementary Fig. 3m), we further established MIF knockout DLD1 cells, which express mutant Pol δ (R689W and R506H in the conserved DNA polymerase III and exonuclease III motifs). Using the established method 6-TG selection, we quantified the nucleotide misincorporation in the HPRT gene and found that MIF knockout increased the nucleotide misincorporation rate by 13.7-fold in DLD1 cells. Restoring MSH6 expression in DLD1 cells reduced the overall mutation rate as expected (Fig. 3k). However, MIF knockout still significantly increased the nucleotide misincorporation mutation in these MMR-proficient DLD1 cells (Fig. 3k). Moreover, our in vitro gap-filling assay also supported that nuclease-deficient MIF-E22A significantly increased nucleotide misincorporation rate as compared with MIF (Fig. 2h and Supplementary Fig. 2i). Taken together, our in vitro nuclease assay, elongation assay, gap-filling assay and in vivo HPRT mutation, DNA fiber assay/replication stalling analysis, and analysis of MIF recruitment to the replication foci reveal that MIF plays an important role in DNA proofreading and MIF knockout increases nucleotide misincorporation leading to the replication fork stalling in cancer cells, which can be reversed by expression of WT MIF, but not nuclease-deficient MIF-E22A mutant. These new results have been shown in Fig.3i-3l and sFig.3l-3o and described in the revised manuscript as following:

The hypoxanthine-guanine phosphoribosyltransferase (HPRT) gene on the X chromosome has been used as a model gene to investigate mutability in mammalian cell lines^{30, 31}. To further explore the effect of MIF on replication fidelity, we established POLD1 knockdown (KD) and POLD1 KD/MIF KO MDA-MB-231 cells using CRISPR/Cas9 sgRNAs (Supplementary Fig. 3l) and performed HPRT mutation assay following 6-TG treatment (20 μ M, 12 days). We found that the HPRT mutation rate was very low (2.6×10^{-7}) in Pol δ -proficient MDA-MB-231 cells (Fig. 3i), whereas MIF KO increased the HPRT mutation rate (1.43×10^{-6}), which was mainly due to the nucleotide misincorporation (Fig. 3i, j). Since POLD1 KD MDA-MB-231 cells were not able to form colonies for HPRT assay (Supplementary Fig. 3m), we instead established MIF KO in colorectal cancer cell line DLD1 cells, which have Pol δ mutations R689W and R506H in the conserved DNA polymerase III and exonuclease III motifs, respectively³². The overall mutation frequency in Pol δ -deficient DLD1 cells was about 8.68×10^{-5} , which was increased by 1.6-fold in MIF KO cells (1.4×10^{-4} , Fig. 3k). It is of note that the nucleotide misincorporation rate in parental DLD1 cells was quite low (7.1×10^{-6}) and MIF KO significantly increased the misincorporation rate by 13.6-fold (9.65×10^{-5} , Fig. 3l). MIF KO DLD1 cells expressing WT MSH6 (DLD1+MSH6) were also established, as DLD1 cells are deficient in MMR due to the mutation in MSH6 (Supplementary Fig. 3n, o). Restoring MSH6 expression in DLD1 cells reduced the overall mutation rate to 1.55×10^{-5} , which is expected. However, MIF KO still significantly increased the overall mutation frequency (1.0×10^{-4} , Fig. 3k). Notably, the misincorporation rate in the MMR-proficient DLD1 cells was only about 3.5×10^{-6} , which was increased by 19-fold in MIF KO cells (6.67×10^{-5} , Fig. 3k). These findings indicate that MIF has a significant impact on correcting nucleotide misincorporation in vivo.

6. Fig. 4D: iPOND experiments need thymidine chase samples to prove that MIF is enriched at nascent DNA.

Response: As suggested, we have performed iPOND experiments with thymidine chase and found that MIF was indeed associated with newly synthesized DNA at the replication forks (Fig. 4d). However, unlike PCNA, MIF persistently bound to newly replicated chromatin even after a thymidine chase, indicating that MIF is not a replisome protein, but a chromatin-bound protein possibly involved in some post-replicative or damage repair-related events. These results have been shown in Fig.4d and described in the revised manuscript (Page 14) as following:

Next, we investigated whether MIF is recruited to the DNA replication site after its nuclear translocation. We performed iPOND (isolation of proteins on nascent DNA) assay and found that MIF was indeed associated with newly synthesized DNA at the replication forks (Fig. 4d). However, unlike PCNA, MIF persistently bound to newly replicated chromatin even after a thymidine chase, indicating that MIF is not a replisome protein, but a chromatin-bound protein possibly involved in some post-replicative or damage repair-related events.

7. Fig. 4E and 4F: Is MIF-PARP1 interaction direct? In vitro GST pull-down assays using purified recombinant proteins should be performed.

Response: As suggested, we performed GST pull-down assay using purified MIF and PARP1 and found that MIF and PARP1 directly interacted with each other. Moreover, we also mapped their binding domains by co-immunoprecipitation assay and found that MIF bound to the N-terminal DNA binding domain of PARP1. These new results have been shown in supplementary Fig.5f-h and described in the revised manuscript (page 17) as following:

To further confirm if PARP1 directly binds to MIF, GST-MIF-bound Sepharose beads were incubated with purified PARP1, PCNA, or GST protein. We found that PARP1, but not PCNA or GST, directly bound to GST-MIF (Supplementary Fig. 5f). V5-tagged full-length (FL) PARP1 and a series of PARP1 truncates including Δ Zn, Δ Zn-BRCT, Δ WGR-CAT, Δ CAT were generated to systematically map MIF's binding domain on PARP1 by co-IP (Supplementary Fig. 5g). We observed that MIF mainly bound to the N-terminal zinc finger domain, as deletion of the zinc finger domain abolished PARP1-MIF interaction (Supplementary Fig. 5g, h).

8. Figs. 4G and 4H: Colocalization of MIF with EdU and PCNA is hard to interpret because EdU and PCNA foci are not clearly visible. Pre-extraction of non-chromatin proteins might improve the experiments. Colocalization needs to be properly quantitated.

Response: As suggested, we repeated the experiment and improved the image quality to better visualize the EdU and PCNA foci. In the new images, we can observe clearer colocalization of MIF-EdU and MIF-PCNA. These new results have been shown in the revised Fig. 4g and 4h and the colocalization data are quantified in Fig.4i.

9. The mechanism of MIF recruitment to the unpaired 3' DNA ends is not clear. Fig. 5D suggests that PARP1 might be a sensor of misincorporated nucleotides, but the authors did not examine whether PARP1 binds to unpaired 3'-DNA ends in vitro. The role of PARP1 in the recruitment of MIF to replication sites is difficult to interpret because PARP1 KO appears to affect nuclear translocation of MIF (Fig. 5H).

Response: It is known that PARP1 binds to DNA without sequence specificity via its N-terminal DNA binding domain, which has been confirmed by EMSA assays as well as structure analysis in many different studies (Zandarashvili L et al., Science, 2020, PMID: 32241924; Melikishvili M et al., Cell Discovery, 2017, PMID: 29387452; Pascal JM. DNA repair. 2018, PMID: 30177435; Matta E. et al., Sci Rep. 2020;

PMID: 32111879). It was also shown in many studies that PARP1 binds to damaged DNA leading to PARP1 activation and PAR production (Gupte R. et al., Genes Dev. 2017. PMID: 28202539; Thomas C. et al., PNAS. 2019. PMID:31028139). In another word, PAR is a good indicator to determine if PARP1 binds to DNA especially damaged DNA. Indeed, our data showed that double-stranded DNA with 3' unpaired ends caused PARP1 activation to produce more PAR as compared with double-stranded DNA without 3' unpaired ends (Fig. 5d and supplementary Fig. 5d,e), indicating that PARP1 binds to these DNAs and its binding to double-stranded DNA with 3' unpaired ends is relatively stronger. Although EMSA assay can determine if PARP1 binds to our DNA substrates (which is known), it cannot tell if PARP1 binds to the unpaired nucleotides at the 3' end or paired nucleotides next to the 3' end. It is technically challenging to study if PARP1 directly binds to a specific single nucleotide at the 3' end. Our in vitro PAR synthesis assay will be a good assay to give functional indication that PARP1 is more sensitive to double-stranded DNA with 3' unpaired ends. Moreover, we mapped PARP1-MIF interaction domain and found MIF binds to the N-terminal domain of PARP1. These data indicate that PARP1 binds to DNA via its N-terminal domain and further recruits MIF to the unpaired 3' DNA ends.

PARP1 should not be the one recruiting MIF from the cytosol to the nuclei as PARP1 mainly localizes in the nuclei and MIF mainly locates in the cytosol. In order to further validate our data, we performed new immunostaining experiments to compare the effects of PARP1 KO and inhibitor on MIF recruitment to the replication sites. Consistent with what we observed in co-IP study, PARP1 KO reduces MIF recruitment to the replication sites whereas PARP inhibitor increases its recruitment to DNA replication site. Interestingly, PARP1 KO did not block MIF nuclear translocation but indeed reduced nuclear translocation of MIF to a certain degree. Previously, it has been shown that mitochondrial protein apoptosis-inducing factor (AIF) interacts with MIF and recruits MIF to the nuclei under severe DNA damage conditions/ischemic stroke/brain injury (Wang Y. et al., 2016, Science, PMID: 27846469). Under normal conditions, AIF is not released from mitochondria, thus MIF nuclear translocation in cancer cells likely involves additional mechanisms. Studies from others have indicated that the posttranslational modifications contribute to the nuclear translocation of the nuclease DNA2 (Chen X. et al., Nat Struct Mol Biol. et al., 2012, PMID: 21841787; Meng Y et al., Nucleic Acids Res., 2019, PMID: 31216032). Our data suggested that PARP1 does not contribute to MIF nuclear translocation but may play a role to secure/lock MIF in the nuclei once MIF is translocated to the nuclei. Without PARP1, MIF may travel back to the cytosol. Future studies are required to explore the molecular mechanism underlying MIF nuclear translation and how PARP1 prevents MIF travelling back to the cytosol.

Taken together, we provided new evidence to support our proposed model: PARP1 recognizes the nucleotide misincorporation as indicated by in vitro PAR activity assay. The N-terminal domain of PARP1 directly interacts with MIF and recruits MIF to the replication sites as indicated by PCNA and Edu staining, where MIF cleaves the misincorporated nucleotides and proofreads DNA synthesis. MIF loss increased the mutations rates, especially the point mutation rate supported by gap-filling and HPRT mutation assays. Eventually PAR formation and PARylation of PARP1 disrupt the complex of PARP1/MIF/PCNA. PARP inhibitor stabilizes the PARP1/MIF/PCNA complex supported by our immunostaining and co-IP studies. The substantial new results have been shown in Fig.3i-l, Fig.4g-i, Fig. 5h-l and supplementary Fig. 3l, Fig. 5f-j throughout the manuscript. The new discussion has been included in the revised manuscript (Page 25-26) as following:

We showed that MIF primarily locates in the cytosol but is translocated to the nucleus when cancer cells enter S phase, which is supported by the previous clinical data showing that nuclear MIF protein is detected in a large number (79.7%) of human tumors including glioblastoma, bladder tumors, and lung adenocarcinoma^{50, 51, 52}. Our biochemical and cell biology studies revealed that PARP1 is required for MIF recruitment to the DNA replication sites through their physical protein-protein interaction after MIF is translocated to the nucleus in S phase. As a DNA damage sensor, PARP1 dynamically binds to DNA and recognizes DNA damage, thereby recruiting nucleases

and DNA repair proteins to the sites of DNA damage to facilitate DNA repair¹⁸. Recent studies revealed that PARP1 is activated by unligated Okazaki fragments in S phase and promotes Okazaki fragment maturation³⁴. In line with this report, our study illustrated that PARP1 senses mis-incorporated nucleotides and is required for MIF recruitment to the replication fork as deletion of PARP1 blocks MIF colocalization with DNA replication protein PCNA. However, PARylation of PARP1 disrupts the PARP1-MIF interaction and decreases the amount of MIF protein at the replication sites, which is supported by our data that PARP inhibitor treatment increases MIF-PCNA interaction and colocalization in the replication sites. Increase of MIF at the DNA replication sites is likely to increase replication speed as MIF loss decreases the DNA replication speed (Fig. 3f-h). Consistent with these findings, a recent study showed that PARylation controls the velocity of replication forks and that inhibition of PARylation by PARP inhibitors increases the speed of fork elongation³⁵. Although our data support that PARP1 plays a pivotal role in recruiting MIF to DNA replication sites, the primary nuclear localization of PARP1 suggests that PARP1 may not function to mediate MIF translocation from the cytosol to the nucleus. Future studies are required to investigate the molecular mechanism underlying MIF nuclear translocation. Taken together, our findings support a working model that PARP1 interacts with MIF and monitors replication errors. When PARP1 detects mis-incorporated nucleotides during DNA replication, it recruits MIF to the damage sites to excise the mis-incorporated nucleotides, facilitating DNA synthesis and promoting cancer cell growth (Fig. 7o).

10. Fig. 5J-L: The authors concluded that “PARylation of PARP1 suppresses MIF and PCNA interaction” (Page 16: line 11) based on the increased MIF-PCNA interaction in olaparib-treated cells. However, the authors need to examine the alternative possibility that PARylation of MIF by PARP1 release MIF from PCNA, which is consistent with the increased PCNA-MIF interaction after olaparib treatments.

Response: As suggested, we tested if MIF is PARylated by an in vitro PARylation assay, and found that MIF itself was not PARylated (Supplementary Fig.5i in the revised manuscript). In addition, we also performed co-immunoprecipitation using PAR and MIF antibodies. We also found that MIF was not PARylated (Fig. 5j-l and Supplementary Fig. 5j). These results exclude an alternative possibility that PARylation of MIF disrupts MIF and PCNA interaction. These new results have been shown in Supplementary Fig. 5i, j and described in the revised manuscript (page 17-18) as following:

To rule out the possibility that MIF is PARylated leading to inhibition of MIF-PCNA interaction, we performed an in vitro PARylation assay and found that MIF was not PARylated by incubating with the purified PARP1 protein in the presence of NAD⁺ and activated DNA (Supplementary Fig. 5i). As expected, PARP1 itself was successfully PARylated. However, PARylated PARP1 was mainly in the supernatant and very little if any bound to GST-MIF-bound beads (Supplementary Fig. 5i). In line with this observation, the treatment of MNNG (50 μ M, 15 min) increased PAR levels but decreased MIF binding to PARylated-proteins, which was determined by immunoprecipitation of PAR antibody (Supplementary Fig. 5j). In addition, MIF itself was not PARylated (Supplementary Fig. 5j).

11. Page 23, line 4: The statement “PARP inhibitor Olaparib suppresses PAR synthesis and enhances MIF recruitment to the DNA replication sites” is not fully supported by data. Does the recruitment of MIF to replication foci increase in olaparib-treated cells? The authors need to test this directly by immunofluorescence.

Response: As suggested, we performed immunostaining assay to study the recruitment of MIF to replication foci in Olaparib-treated cells and found that the recruitment of MIF to the replication foci was indeed increased after Olaparib treatment. These new results have been shown in Fig. 5h, i and described in the revised manuscript (page 17, 26) as following:

To our surprise, the treatment of PARP inhibitor Olaparib (5 μ M, 2 h) did not block MIF nuclear localization. Conversely, it even increased MIF-PCNA nuclear colocalization (Fig. 5h, i). In line with this, the treatment of

Olaparib (5 μ M, 2 h) following double thymidine synchronization clearly increased MIF-PCNA interaction in MDA-MB-231 cells (Fig. 5j-l).

However, PARylation of PARP1 disrupts the PARP1-MIF interaction and decreases the amount of MIF protein at the replication sites, which is supported by our data that PARP inhibitor treatment increases MIF-PCNA interaction and colocalization in the replication sites.

12. Page 23, line 4: The statement “PARylation of PARP1 interferes with PARP1-MIF interaction” contradicts with Fig. 5J, which shows no change in the amount of PARP1 co-immunoprecipitated with MIF in olaparib-treated cells.

Response: We thank the reviewer for pointing this out. Since PARP1 is highly abundant in cells, the increase of PARP1 interaction with MIF after PARP1 inhibitor treatment is relatively difficult to be detected. We have re-checked all the experiments and used the most representative images in the revised manuscript. On the other hand, we also performed substantial new experiments to further support our conclusion that PARylation of PARP1 interferes with PARP1-MIF interaction. First, we studied MIF-PCNA colocalization in the nuclei after PARP inhibitor treatment (Supplementary Fig. 5h-i in the revised manuscript). We found that MIF-PCNA colocalization is increased by Olaparib treatment, which is consistent with the observation of increased MIF-PCNA interaction after PARP inhibitor treatment shown in Fig. 5j. Second, we performed additional new experiments to study if PARP1 directly binds to MIF, if PARylated PARP1 binds to MIF, and if MIF itself is PARylated thereby interfering PARP1-MIF interaction by GST pulldown, in vitro PARylation assay and co-IP (Supplementary Fig. 5f-j in the revised manuscript). We found that PARP1 directly binds to MIF (Supplementary Fig. 5f). Little or no PARylated PARP1 binds to GST-MIF (Supplementary Fig. 5f) and MIF itself is not obviously PARylated (Supplementary Fig. 5i-5j). Third, alkylating agent MNNG was used to treat cancer cells to increase PAR levels. We found that the treatment of MNNG decreased MIF binding to PARylated proteins (Supplementary Fig. 5j). Together, these data strongly support our conclusion that PARylation of PARP1 interferes with PARP1-MIF interaction. These new results have been shown in Fig. 5h-j and Supplementary Fig. 5f-j and described in the revised manuscript (page 17-18) as following:

To further confirm if PARP1 directly binds to MIF, GST-MIF-bound Sepharose beads were incubated with purified PARP1, PCNA, or GST protein. We found that PARP1, but not PCNA or GST, directly bound to GST-MIF (Supplementary Fig. 5f). V5-tagged full-length (FL) PARP1 and a series of PARP1 truncates including Δ Zn, Δ Zn-BRCT, Δ WGR-CAT, Δ CAT were generated to systematically map MIF's binding domain on PARP1 by co-IP (Supplementary Fig. 5g). We observed that MIF mainly bound to the N-terminal zinc finger domain, as deletion of the zinc finger domain abolished PARP1-MIF interaction (Supplementary Fig. 5g, h). Immunostaining further showed that PARP1 KO blocked MIF-PCNA colocalization at the DNA replication site in MDA-MB-231 cells (Fig. 5h, i). To our surprise, the treatment of PARP inhibitor Olaparib (5 μ M, 2 h) did not block MIF nuclear localization. Conversely, it even increased MIF-PCNA nuclear colocalization (Fig. 5h, i). In line with this, the treatment of Olaparib (5 μ M, 2 h) following double thymidine synchronization clearly increased MIF-PCNA interaction in MDA-MB-231 cells (Fig. 5j-l).

To rule out the possibility that MIF is PARylated leading to inhibition of MIF-PCNA interaction, we performed an in vitro PARylation assay and found that MIF was not PARylated by incubating with the purified PARP1 protein in the presence of NAD⁺ and activated DNA (Supplementary Fig. 5i). As expected, PARP1 itself was successfully PARylated. However, PARylated PARP1 was mainly in the supernatant and very little if any bound to GST-MIF-bound beads (Supplementary Fig. 5i). In line with this observation, the treatment of MNNG (50 μ M, 15 min) increased PAR levels but decreased MIF binding to PARylated-proteins, which was determined by immunoprecipitation of PAR antibody (Supplementary Fig. 5j). In addition, MIF itself was not PARylated (Supplementary Fig. 5j). Taken together, these data indicate that PARP1 is required for MIF recruitment to the

DNA replication site and PARylation of PARP1 suppresses MIF and PCNA interaction, thereby allowing MIF to dissociate from the DNA replication sites

13. The authors need to determine whether cancer cells are more dependent on MIF than normal cells are. Based on the COSMIC database, no mutations have been found in POLD1 or POLE1 in MDA-MB-231, which was used in the experiments. What then causes nucleotide misincorporation that makes MIF critical for DNA replication (Fig. 3) and tumor growth (Fig. 7)?

Response: As the reviewer suggested, we first performed new experiments to determine the effects of MIF on proliferation and survival of cancer cells vs normal cells. We found that MIF knockout inhibited cancer cell growth, but did not obviously affect growth of non-tumorigenic mammary epithelial cell line MCF-10A or mouse embryonic fibroblasts (Supplementary Fig. 7a, i-k). MIF knockout mice are relatively normal without obvious defects. These results indicate that MIF is more favorable for growth of cancer cells rather than normal cells. Second, we also studied the effects of MIF on nucleotide misincorporation rate in Pol δ -proficient cells vs Pol δ -deficient cells (Fig. 3i-l and Supplementary Fig. 3h-o) and further discussed possible causes for nucleotide misincorporation that makes MIF critical for DNA replication. Our HPRT mutation assay showed that MIF knockout increased mis-incorporation mutations in Pol δ -proficient MDA-MB-231 cells and this effect became more robust in Pol δ -deficient MDA-MB-231 cells (Fig. 3i-l). Pol α plays an important role in initiation of DNA replication and Okazaki fragment on the lagging strand, but it lacks 3' exonuclease activity for proofreading errors. Our data indicate that MIF may cooperate with Pol α to ensure the accuracy and success of the primer elongation in Pol δ and Pol ϵ proficient cancer cells. In addition, polymerase dissociation from DNA occurs when a second structure like R-loop, hairpin, stem-loop, G-quadruplex, fork reversal or slippage is formed during replication (Kaushal S. and Freudenreich C. *Genes Chromosomes Cancer*, 2019, PMID: 30536896; Maffia A. et al., *Int J Mol Sci*. 2020, PMID: 32098397; Viguera E. et al., *EMBO J*. 2001, PMID: 11350948) and Pol δ and Pol ϵ are less stringent in discriminating dNTPs and rNTPs. Misincorporation of rNTPs occurs and causes replication stress (Zeman MK. et al., *Nat Cell Biol*. 2014, PMID: 24366029). MIF may resolve second structures like hairpin and stem-loop and correct misincorporation, which is supported by our *in vitro* MIF nuclease assays and *in vivo* mutation analysis. Thus, MIF may also cooperate with nuclease-proficient polymerases δ and ϵ and allow them to maintain high speed of DNA replication, which may also explain the discrepancy of polymerase-mediated high rate of nucleotide mis-incorporation *in vitro* but low rate *in vivo*. The new figures were included as Fig. 3i-l, Supplementary Fig. 3h-o and Supplementary Fig. 7a, i-k in the revised manuscript. The new Results (page 12-13, 19-20) and Discussion (page 22-23, 27) of the revised manuscript have also been shown as following:

To determine if MIF promotes cancer cell growth, parental and MIF KO MDA-MB-231 cells were subjected to cell proliferation assay in vitro. MIF loss in Pol δ -proficient MDA-MB-231 cancer cells significantly reduced cell proliferation (Supplementary Fig. 7a). MIF KO similarly suppressed Pol δ -deficient cancer cell growth, although loss of Pol δ already significantly inhibited MDA-MB-231 cell growth (Supplementary Fig. 7a). Clonogenic assay showed that MIF KO significantly decreased colony survival of MDA-MB-231, LN229 and HCT116 cells (Supplementary Fig. 7b-d). Consistently, MIF KD by any of three independent short hairpin RNAs also reduced colony survival of HCT116 cells (Supplementary Fig. 7e-h). Notably, expression of WT MIF partially restored reduced colony formation conferred by MIF KO2, whereas E22A MIF mutant failed to do so (Fig. 7a-b). These findings reveal that MIF promotes cancer cell survival and growth in vitro through its nuclease activity.

We next determined the role of MIF in normal cell growth. MIF was knocked out in a non-tumorigenic mammary epithelial cell line MCF-10A cells using two independent sgRNAs (Supplementary Fig. 7i) and the growth pattern was compared. We discovered that MIF KO1 or KO2 did not obviously alter MCF-10A cell proliferation, although they repressed MCF-10A colony survival (Supplementary Fig. 7j, k). Similar results were observed in mouse

embryonic fibroblasts. These data indicate that MIF is more vulnerable for growth of cancer cells than normal cells.

However, MIF loss does not obviously alter the growth of non-tumorigenic epithelial cells MCF-10A or mouse embryonic fibroblasts and MIF knockout mice are normal without obvious defects, suggesting that the expression of MIF is more favorable for cancer cell growth rather than normal cells.

The hypoxanthine-guanine phosphoribosyltransferase (HPRT) gene on the X chromosome has been used as a model gene to investigate mutability in mammalian cell lines^{30, 31}. To further explore the effect of MIF on replication fidelity, we established POLD1 knockdown (KD) and POLD1 KD/MIF KO MDA-MB-231 cells using CRISPR/Cas9 sgRNAs (Supplementary Fig. 3l) and performed HPRT mutation assay following 6-TG treatment (20 μ M, 12 days). We found that the HPRT mutation rate was very low (2.6×10^{-7}) in Pol δ -proficient MDA-MB-231 cells (Fig. 3i), whereas MIF KO increased the HPRT mutation rate (1.43×10^{-6}), which was mainly due to the nucleotide misincorporation (Fig. 3i, j). Since POLD1 KD MDA-MB-231 cells were not able to form colonies for HPRT assay (Supplementary Fig. 3m), we instead established MIF KO in colorectal cancer cell line DLD1 cells, which have Pol δ mutations R689W and R506H in the conserved DNA polymerase III and exonuclease III motifs, respectively³². The overall mutation frequency in Pol δ -deficient DLD1 cells was about 8.68×10^{-5} , which was increased by 1.6-fold in MIF KO cells (1.4×10^{-4} , Fig. 3k). It is of note that the nucleotide misincorporation rate in parental DLD1 cells was quite low (7.1×10^{-6}) and MIF KO significantly increased the misincorporation rate by 13.6-fold (9.65×10^{-5} , Fig. 3l). MIF KO DLD1 cells expressing WT MSH6 (DLD1+MSH6) were also established, as DLD1 cells are deficient in MMR due to the mutation in MSH6 (Supplementary Fig. 3n, o). Restoring MSH6 expression in DLD1 cells reduced the overall mutation rate to 1.55×10^{-5} , which is expected. However, MIF KO still significantly increased the overall mutation frequency (1.0×10^{-4} , Fig. 3k). Notably, the misincorporation rate in the MMR-proficient DLD1 cells was only about 3.5×10^{-6} , which was increased by 19-fold in MIF KO cells (6.67×10^{-5} , Fig. 3k). These findings indicate that MIF has a significant impact on correcting nucleotide misincorporation in vivo.

Pol δ and Pol ϵ have the 3' exonuclease activity and have been well recognized for proofreading DNA during replication. However, given the fact that increasing numbers of germline and somatic mutations within the exonuclease domain in human POLD1 and POLE have been identified in human cancers^{1, 13, 14} and that our in vitro nuclease assay, DNA elongation assay and gap filling assay clearly showed that MIF cooperates with Pol α and nuclease-deficient Pol δ to proofread DNA and ensure the success of DNA elongation, MIF is likely to play an important role in proofreading in Pol δ - or Pol ϵ -deficient cancer cells. This is indeed further supported by in vivo HPRT mutation analysis showing that MIF KO increases 13.6-fold nucleotide mis-incorporation rate in Pol δ -deficient DLD1 cells. Interestingly, our studies showed that MIF also plays an important role in nuclease-proficient MDA-MB-231 cancer cells that do not have POLD1 and POLE mutations, as MIF KO increases mis-incorporation mutations and genomic instability, delays cell cycle and inhibits DNA synthesis and cancer growth in vitro and in mice. Pol α plays an important role in initiation of DNA replication and Okazaki fragment on the lagging strand, but it lacks 3' exonuclease activity for proofreading errors. Our data indicate that MIF may cooperate with Pol α to ensure the accuracy and success of the primer elongation in Pol δ - and Pol ϵ -proficient cancer cells. In addition, polymerase dissociation from DNA occurs when a second structure like R-loop, hairpin, stem-loop, G-quadruplex, fork reversal or slippage is formed during replication^{36, 37, 38}. Pol δ and Pol ϵ are also less stringent in discriminating dNTPs and rNTPs. Misincorporation of rNTPs occurs and causes replication stress³⁹. Under these conditions, MIF may resolve second structures like hairpin and stem-loop and correct misincorporation, which is supported by our in vitro MIF nuclease assay and in vivo mutation analysis. Thus, MIF may also cooperate with nuclease-proficient Pol δ and Pol ϵ and allow them to maintain high speed of DNA replication. Further investigation is required to explore how MIF cooperates with polymerases during DNA replication.

Minor points

14. Fig. 1A: Several different flap sequences need to be tested for sequence specificity of MIF.

Response: As suggested, we have generated 3 different types of the flap sequences with free T, A, or C at the 3' end as the substrates and found that MIF cleaved all 3 substrates at the 3' end regardless of its

sequence, suggesting that MIF's 3' flap exonuclease activity is independent of the substrate sequence. The new results have been shown in Supplementary Fig. 1a and described in the revised manuscript (page 5-6) as following:

Moreover, MIF cleaved the unpaired 3' end nucleotide regardless of its sequence as "T", "A" or "C" (Supplementary Fig. 1a). These data indicate that MIF recognizes Y-shaped dsDNA as the substrate and possesses both 3' exonuclease activity and 3' flap endonuclease activity to selectively cleave away the short flap at the 3' end, which depends on the substrate structure but not sequence.

15. Fig. 2B: MIF-only samples need to be included to eliminate a possibility that the MIF preparation contains polymerase activity (intrinsically or due to contamination).

Response: As suggested, we included MIF-only sample in *in vitro* elongation assay (new Figure 2b, lane 8) and did not observe the polymerase activity of MIF. These data suggest that MIF has the nuclease activity, but not the polymerase activity. The new results have been shown in Figure 2b and described in the revised manuscript (page 8) as following:

MIF itself did not possess the polymerase activity (Fig. 2b, lane 8).

16. Fig. 2F: Does MIF influence the fidelity of WT Pol δ ?

Response: As suggested, we performed gap-filling assay in the presence of WT Pol δ and MIF and found that MIF did not further increase Pol δ -mediated fidelity of DNA synthesis. This is not a surprise as the mutation rate of WT Pol δ is very low. The data have been included in the revised Fig. 2f-h and described in the revised manuscript (page 10) as following:

We also examined the effect of MIF-WT on Pol δ -mediated mutagenesis and found that MIF did not further reduce the mutation frequency induced by Pol δ (Fig. 2f-h).

17. Fig. S4A: In vitro GST pull-down assays using purified PCNA and GST-MIF need to be done to test whether interaction of MIF and PCNA is direct.

Response: As suggested, we performed GST pull-down assay using purified PCNA, PARP1 and MIF. GST was used as a negative control. We found that MIF cannot directly bound to PCNA, which was consistent with the fact that MIF does not contain the PCNA binding PIP-box motif. Under the same condition, MIF directly pulled down PARP1. These new results have been shown in revised Supplementary Fig. 5f-h and described in the revised manuscript (page 17) as following:

To further confirm if PARP1 directly binds to MIF, GST-MIF-bound Sepharose beads were incubated with purified PARP1, PCNA, or GST protein. We found that PARP1, but not PCNA or GST, directly bound to GST-MIF (Supplementary Fig. 5f). V5-tagged full-length (FL) PARP1 and a series of PARP1 truncates including Δ Zn, Δ Zn-BRCT, Δ WGR-CAT, Δ CAT were generated to systematically map MIF's binding domain on PARP1 by co-IP (Supplementary Fig. 5g). We found that MIF mainly bound to the N-terminal zinc finger domain, as deletion of the zinc finger domain abolished PARP1-MIF interaction (Supplementary Fig. 5g, h).

18. Fig. 7I: The authors need to show MIF WB to compare the overexpressed levels with endogenous MIF levels.

Response: As suggested, we performed MIF western blot assay and compared the overexpressed MIF levels with endogenous MIF levels. The new results have been shown in the revised Figure 7i.

Reviewers' comments:

Reviewer #1 (Remarks to the Author):

The revised manuscript is improved. The authors made a tremendous effort to revise it and address the concerns. Of particular importance is the experiment showing increased mutagenesis in MIF deficient cells.

Minor comments on introduction:

"DNA polymerases (Pol) α , δ and ϵ are three key polymerases contributing to DNA replication in mammals 2, 7, 8."

Please change mammals to eukaryotes as this applies to all eukaryotes.

"It is not yet completely understood how the DNA proofreading process is controlled in cancer cells while DNA polymerases keep incorporating nucleotides at such an amazingly high speed."

This statement needs a citation. What is the rate of replication in cancer cells when compared to noncancer cells?

"Furthermore, there is a clear discrepancy of polymerase-mediated nucleotide mis-incorporation rate between in vitro and in vivo studies. The nucleotide mis-incorporation rate of Pol δ and Pol ϵ in vivo is about 1/108-1010, which is much lower than their in vitro rate (1/104-105)"

This statement is misleading. MMR reduces replication error rate by about 100-1000 folds and in vitro experiments do not include MMR proteins.

Reviewer #2 (Remarks to the Author):

I am afraid I am still not convinced by the data that MIF is a bona fide nuclease and the activity is not a result of a contaminant. Several of the arguments made to justify this in the rebuttal are spurious. For example, it is stated several times in that Fen1 has a comparable half-life to MIF but that is untrue-the half-life of FEN1 substrates, which are typical of replicative nucleases, are in the region 10-40 milliseconds so the difference is several orders of magnitude. There is an excellent way of determining whether a co-purifying nuclease activity actually belongs to an enzyme using DNA substrate gels. For example, Molecular interactions of Escherichia coli ExoIX and identification of its associated 3'-5' exonuclease activity Nucleic Acids Research, Volume 35, 2007, 4094-410. It would be good to see these sort of studies carried out here.

Reviewer #3 (Remarks to the Author):

Conceptually, it is difficult to understand why cells need to recruit another nuclease to misincorporated nucleotides when the exact polymerase that caused misincorporation is equipped with an exonuclease to correct such misincorporation. Recruiting MIF via PARP1 to misincorporated nucleotides seems to be too slow for the job of correcting misincorporation.

Authors' proposed model is still speculative. The authors tend to overstate the in vivo functions of MIF and PARP1 based on speculation from the biochemical studies in vitro. There is no evidence that PARP1 actually recognizes misincorporated nucleotides in vivo. There is no evidence that nucleotide misincorporation caused excessive replication fork stalling in MIF KO cells. Importantly, localization of MIF to the replication sites is no longer supported by iPOND experiments. I remain skeptical about the final model proposed in this manuscript.

Major points:

1) The authors provided no data to support that the excessive replication fork stalling in MIF KO cells is caused by nucleotide misincorporation. They responded to the critique by citing the increased mutation frequency in MIF KO cells, but increased mutations do not necessarily mean replication forks were stalled when misincorporations occurred.

2) The new thymidine chase experiments in iPOND failed to provide evidence for MIF localization at replication forks. MIF proteins were detected in the EdU pull-down even after thymidine chase. This data should not be interpreted to indicate that "MIF is [...] possibly involved in some post-replicative or damage repair-related events" (Page 14, line 14) without additional evidence because it is possible that MIF is ubiquitously distributed on chromatin.

3) The molecular events at the misincorporated nucleotides are still defined poorly. The authors appear to propose that PARylation of PARP1 promotes dissociation of MIF from PARP1, but what brings the released MIF to PCNA is completely absent in the proposed model.

4) Tumor specific replication stress that makes MIF important for cell survival is undefined. The authors' response and revised Discussion contain too much speculation.

Minor point:

1) Concerns remain whether in vitro extension assays were performed under an optimal condition. The stoichiometry of Pol delta subunits seems to be off as POLD3 and POLD4 appear to be overrepresented.

Response to Reviewers

Referees comments are highlighted in blue and the response is provided below the referee comments.

Reviewer #1:

1. The revised manuscript is improved. The authors made a tremendous effort to revise it and address the concerns. Of particular importance is the experiment showing increased mutagenesis in MIF deficient cells.

Response: We thank the referee for agreeing that the authors made a tremendous effort and the revised manuscript is improved.

2. Minor comments on introduction: “DNA polymerases (Pol) α , δ and ϵ are three key polymerases contributing to DNA replication in mammals 2, 7, 8.” Please change mammals to eukaryotes as this applies to all eukaryotes.

Response: As the reviewer suggested, we have changed “mammals” to “eukaryotes” in the revised manuscript.

3. “It is not yet completely understood how the DNA proofreading process is controlled in cancer cells while DNA polymerases keep incorporating nucleotides at such an amazingly high speed.” This statement needs a citation. What is the rate of replication in cancer cells when compared to noncancer cells?

Response: As the reviewer suggested, we have cited the literatures in the revised manuscript (Ganai RA and Johansson E. Mol Cell, 2016, PMID: 27259205; Schoonen PM et al., Adv Protein Chem Struct Biol. 2019, PMID: 30798931; Ubhi T and Brown GW. Cancer Res, 2019, PMID: 30967400). Since the growth speed of many different types of cancer and noncancer cells varies, it is hard to provide the exact rate of replication in cancer cells vs. noncancer cells. Cancer cells are not necessary to grow or divide faster than normal cells. However, one key feature of cancer cells distinct from normal cells is that cancer cells lack the control of cell growth and keep growing and dividing that brings higher replication stress than normal cells (Schoonen PM et al., Adv Protein Chem Struct Biol. 2019, PMID: 30798931; Ubhi T and Brown GW. Cancer Res, 2019, PMID: 30967400). This information has been already included in Introduction in the last submission and also included as following:

DNA replication is a central event for cell proliferation and malfunction of the DNA replication machinery causes DNA replication stress. While DNA replication stress-induced genomic instability in normal cells has been thought as a key driver of tumorigenesis^{1, 2, 3, 4}, the continuous abnormal proliferation-induced DNA replication stress may cause profound genomic instability in cancer cells and bring a threat to cancer cell viability^{5, 6}.

4. “Furthermore, there is a clear discrepancy of polymerase-mediated nucleotide mis-incorporation rate between in vitro and in vivo studies. The nucleotide mis-incorporation rate of Pol δ and Pol ϵ in vivo is about $1/10^8$ - 10^{10} , which is much lower than their in vitro rate ($1/10^4$ - 10^5). This statement is misleading. MMR reduces replication error rate by about 100-1000 folds and in vitro experiments do not include MMR proteins.

Response: As suggested, we revised the statement by pointing out the MMR functions in reducing replication error in the revised manuscript (page 3) as following:

Furthermore, there is a clear discrepancy of polymerase-mediated nucleotide mis-incorporation rate between in vitro and in vivo studies. The nucleotide mis-incorporation rate of Pol δ and Pol ϵ in vivo is about $1/10^8$ - 10^{10} , which is much lower than their in vitro rate ($1/10^4$ - 10^5). Mismatch repair has been counted as one of main contributors to correct in vivo replication errors that escape proofreading. However, the efficiency of mismatch repair varies at different positions in the genome². It is not known if additional mechanisms may be involved in replication to correct replication errors in vivo.

Reviewer #2

I am afraid I am still not convinced by the data that MIF is a bona fide nuclease and the activity is not a result of a contaminant. Several of the arguments made to justify this in the rebuttal are spurious. For example, it is stated several times in that Fen1 has a comparable half-life to MIF but that is untrue—the half-life of FEN1 substrates, which are typical of replicative nucleases, are in the region 10-40 milliseconds so the difference is several orders of magnitude. There is an excellent way of determining whether a co-purifying nuclease activity actually belongs to an enzyme using DNA substrate gels. For example, Molecular interactions of Escherichia coli ExoIX and identification of its associated 3'–5' exonuclease activity Nucleic Acids Research, Volume 35, 2007, 4094–410. It would be good to see these sort of studies carried out here.

Response: We thank the referee for the positive comment “a novel and highly significant finding” on our original submission. We also thank the referee for the constructive suggestion to help improve our manuscript.

Regarding Reviewer's comments on Fen1, since Fen1's half-life and kinetics analysis is not always consistent from different literatures depending on the sensitivity of different substrates and assays used (see Figure 6, Tsutakawa SE et al., Cell, 2011; Negritto MC et al., Mol. Cell. Biol, 2001; Zaher MS et al., Nucleic Acids Res. 2018; Liu R et al., Nucleic Acids Res. 2006), we have revised the statement by removing Fen1 in the revised manuscript as following:

Although the kinetic analysis of MIF activity revealed a relatively slow reaction with $t_{1/2}$ at 7.7-10.8 min, K_M at 0.7-0.9 μM and k_{cat} at 0.02/min, it is comparable to some well-recognized nucleases including MutsS2, exoribonuclease (ExoN), ribozyme, and EcoRI^{42, 43, 44, 45, 46}.

We have also deleted the following statement:

The $t_{1/2}$ of Fen1 varies from milliseconds to 30 min depending on measuring the Fen1/DNA binding affinity⁴⁹ or the actual cleavage rate⁴⁷, respectively.

To address the reviewer's concern regarding the possible nuclease contamination in purified MIF protein, we have performed new experiments using DNA substrate gels as the reviewer suggested and confirmed that MIF's 3' nuclease activity was not caused by a contaminated nuclease (Fig. 1 below). DNase I, bovine serum albumin (BSA, Cat#3912, Sigma), GST, and restriction endonuclease BamHI were used as controls. We found that DNase I caused Salmon sperm DNA degradation and showed a specific black band at the expected DNase I position, indicating that our assay works well. Under the same experimental conditions, a black band around 100 kDa above the BSA band was shown up in the BSA lane, indicating a possible nuclease contamination. This finding is consistent with the previous reports that BSA might be contaminated with DNase I and RNase (Odunuga OO & Shazhko Alina, 2013; Slaaby R et al., 2008). However, there was no obvious nuclease contamination in MIF, E22A, and GST groups. We also acknowledge that this assay has its limitation. The assay can detect a nuclease that fully degrades DNA but not an endonuclease that only cuts DNA without degradation. For example, the endonuclease activity of BamHI was not detected in our studies. Note, the band in BamHI group was attributed to nucleases contaminated in BSA as the enzyme were diluted in BSA buffer. Similarly, because MIF cleaves away 3' flap nucleotides only without DNA degradation, it was not surprising that there was no band at MIF protein position. Nevertheless, this assay confirms that MIF protein preparation we used in our nuclease assays did NOT show obvious contamination with nucleases like DNaseI or E. coli ExoIX, which contains 3'→5' exonuclease activity and can be detected by DNA substrate gels as reported previously (Hodskinson MR et al., Nucleic Acids Res. 2007, PMID: 17567612).

Together, in the last two revisions, we have utilized multiple different approaches to confirm the purity of our MIF protein preparation and to support that MIF indeed contains a unique nuclease activity. These results are summarized as following:

- 1) Coomassie blue staining shows the purity of MIF protein (Fig. 1f in the revised manuscript).

- 2) We performed mass spectrometry to analyze our MIF protein preparations used in our *in vitro* nuclease assay. As shown in Supplementary Table 1 in the revised manuscript, no known nucleases were

Table 1. Mass spectrometry analysis of MIF protein preparations

Accession	Description	Coverage (%)	Spectrum counts			
			Buffer	Sample 1	Sample 2	Sample 3
P14174	Macrophage migration inhibitory factor	42	0	127	346	985
P0A6Y8	Chaperone protein DnaK	89	0	15	99	26
P0C8J8	D-tagatose-1,6-bisphosphate aldolase subunit GatZ	75	0	5	66	27
Q8WXH0	Nesprin-2	0	0	5	12	4
P0A6F5	60 kDa chaperonin	85	0	0	56	8

identified by mass spectrometry, although 4 additional chaperone proteins with very low abundance close to the background were detected. (Samples 1 and 3 were mainly used in our *in vitro* assays).

- 3) The negative controls GST and MIF E22A mutant, which were purified using the exact same system and method as MIF shown in Fig. 1g, do not show obvious nuclease activities, which also strongly supports the good purity of our MIF protein and excludes a possible nuclease contamination.

- 4) Our *in vitro* nuclease assays were conducted with many different batches of MIF protein preparations and we observed the reproducible results. Moreover, our previous study (Fig. 3 from Wang Y et al., 2016 Science, PMID: 27846469) and current study (Fig. 1d from the revised manuscript) both showed that MIF selectively cleaves single-stranded DNA with hairpin loop structure and double-stranded DNA with Y-shaped structure. Both substrates share a common Y-shaped structure. If it is due to the contamination, non-specific proteins unlikely cleave various substrates with different structures with such strong specificity and selectivity.

Fig. 3. MIF cleavage of unpaired bases at the 3' end of the stemloop of 5' or 3' biotin-labeled small DNA substrates with various structures or sequences in a nuclease assay (see fig. S8 for illustrations of substrates, and tables S1 and S2 for sequences). Experiments were replicated four times with three independent preparations of MIF protein. Adopted from Fig. 3C (Wang Y et al., 2016 Science).

5) MIF trimer contains a nuclease core structure and it can be nicely overlapped with multiple classic endonuclease including EcoRI, EcoRV, ExoII and PvuII (Fig. S3). Although MIF cleaves at the 3' overhang, our *in vitro* EMSA assay (Fig. 3B) showed that MIF binds to the 5' free arm of the hairpin loop structure, which can be blocked by either MIF antibody (Fig. 3C) or removing 5' free arm (Fig. 3B). These data further provide structure base of MIF as a nuclease and also confirms the specificity of MIF binding to DNA.

6) Our *in vitro* elongation assay and gap-filling assays in the presence/absence of MIF (Fig. 2 in the revised manuscript) as well as the extensive *in vivo* studies (Figs. 3-7) further support that MIF has 3' flap nuclease activity.

7) Moreover, MIF proteins purchased from three different companies (Shenandoah, Novus, R&D) all showed similar 3' nuclease activity to cleave the 3' unpaired nucleotide labeled with fluorescein (Fig. 1 in this letter). The cleavage was detected by both Ethidium Bromide (EtBr) and more obviously by the fluorescence staining, which was not detected in GST or DNA substrate only groups (Fig. 3 in this letter). The variable MIF nuclease activity among different sources might be due to the use of different buffer and/or purification system (Fig. 3 in this letter).

Overall, we are very confident that the nuclease activity we observed is specifically attributed to MIF.

Reviewer #3

1. Conceptually, it is difficult to understand why cells need to recruit another nuclease to misincorporated nucleotides when the exact polymerase that caused misincorporation is equipped with an exonuclease to correct such misincorporation. Recruiting MIF via PARP1 to misincorporated nucleotides seems to be too slow for the job of correcting misincorporation.

Response: We thank the referee for the positive comments on our original submission that “The proposed model that MIF proofreads misincorporated DNA and thereby mitigates replication stress and promotes tumor growth is novel” and “the manuscript presents interesting and potentially important observations”.

Overlapped DNA replication and repair mechanisms are quite common in cancer cells, which plays an important role in prevention of lethal and pathogenic DNA mutations and also mediates resistance to chemotherapy drugs (Bullock CR et al., 2020). Mismatch repair (MMR) is a good example to help understand the importance of such a redundant repair mechanism in cancer cells. MMR corrects replication errors in polymerase-proficient cells, although polymerase itself such as Pol δ/ϵ has high nucleotide selectivity and also accurate proofreading exonuclease activity. Likewise, our extensive *in vitro* and *in vivo* studies including elongation assay, gap-filling assay, HPRT mutation assay, and DNA fiber assay showed that MIF cooperated with nuclease-proficient and -deficient Pol δ and Pol α to proofread DNA and facilitate DNA synthesis. MIF may represent another important redundant mechanism during DNA replication in cancer cells and targeting MIF-mediated redundant DNA replication correction mechanism may be a novel potential strategy for cancer therapy.

On the other hand, in order to specifically address the importance of recruiting MIF to the DNA replication sites, we have included the discussion about the possible functions of MIF as 3' nuclease in DNA replication in the revised manuscript (page 22-23) as following:

*Pol δ and Pol ϵ have the 3' exonuclease activity and have been well recognized for proofreading DNA during replication. However, given the fact that increasing numbers of germline and somatic mutations within the exonuclease domain in human POLD1 and POLE have been identified in human cancers^{1, 13, 14} and that our *in vitro* nuclease assay, DNA elongation assay and gap filling assay clearly showed that MIF cooperates with Pol α and nuclease-deficient Pol δ to proofread DNA and ensure the success of DNA elongation, MIF is likely to play an important role in proofreading in Pol δ - or Pol ϵ -deficient cancer cells. This is indeed further supported by *in vivo* HPRT mutation analysis showing that MIF KO increases 13.6-fold nucleotide mis-incorporation rate in Pol δ -deficient DLD1 cells. Interestingly, our studies showed that MIF also plays an important role in nuclease-proficient MDA-MB-231 cancer cells that do not have POLD1 and POLE mutations, as MIF KO increases mis-incorporation mutations and genomic instability, delays cell cycle and inhibits DNA synthesis and cancer growth *in vitro* and *in mice*. Pol α plays an important role in initiation of DNA replication and Okazaki fragment on the lagging strand, but it lacks 3' exonuclease activity for proofreading errors. Our data indicate that MIF may cooperate with Pol α to ensure the accuracy and success of the primer elongation in Pol δ - and Pol ϵ -proficient cancer cells. In addition, polymerase dissociation from DNA occurs when a second structure like R-loop, hairpin, stem-loop, G-quadruplex, fork reversal or slippage is formed during replication^{36, 37, 38}. Pol δ and Pol ϵ are also less stringent in discriminating dNTPs and rNTPs. Misincorporation of rNTPs occurs and causes replication stress³⁹. Under these conditions, MIF may resolve second structures like hairpin and stem-loop and correct misincorporation, which is supported by our *in vitro* MIF nuclease assay and *in vivo* mutation analysis. Thus, MIF may also cooperate with nuclease-proficient Pol δ and Pol ϵ and allow them*

to maintain high speed of DNA replication. Further investigation is required to explore how MIF cooperates with polymerases during DNA replication.

Regarding “Recruiting MIF via PARP1 to misincorporated nucleotides seems to be too slow for the job of correcting misincorporation.”, we have clarified our model in the revised manuscript that PARP1 interacts with MIF and forms a complex prior to binding to the damage sites. PARP1 as the DNA damage sensor detects DNA damage including 3’ flap structures during DNA replication. Then MIF-PARP1 complex is recruited to the damage sites at the replication fork to clear the 3’ flap structure, facilitating DNA synthesis and promoting cancer cell growth (Fig. 7o). PARP1 activation is a very fast event within a minute (Kruger A et al., 2020). Since PARP-MIF complex is formed before PARP1 detects the abnormal sites, the speed of this process should not be of much concern. Detailed explanation about our model has been provided in Response to Point 2 below.

2. Authors’ proposed model is still speculative. The authors tend to overstate the *in vivo* functions of MIF and PARP1 based on speculation from the biochemical studies *in vitro*. There is no evidence that PARP1 actually recognizes misincorporated nucleotides *in vivo*. There is no evidence that nucleotide misincorporation caused excessive replication fork stalling in MIF KO cells. Importantly, localization of MIF to the replication sites is no longer supported by iPOND experiments. I remain skeptical about the final model proposed in this manuscript.

Response: The key finding of this manuscript is that MIF is a novel 3’ flap nuclease and facilitates DNA replication process to promote tumor growth. The model we proposed has been clarified as following: MIF is translocated to the nucleus during S phase. In the nucleus, PARP1 interacts with MIF and forms a complex prior to binding to the damage sites. PARP1 as the DNA damage sensor detects DNA damage including 3’ flap structures during DNA replication. Then MIF-PARP1 complex is recruited to the damage sites at the replication fork to clear the 3’ flap structure, facilitating DNA synthesis and promoting cancer cell growth (Fig. 7o). Each individual process proposed in this model has been strongly supported by our extensive *in vitro* and *in vivo* evidence listed as following.

- 1) MIF is a 3’ flap nuclease and cleaves 3’ flap nucleotides to facilitate DNA elongation, which is supported by our *in vitro* nuclease assay (Fig. 1), elongation assay (Fig. 2a-d) and gap-filling elongation assay (Fig. 2e-h).
- 2) MIF is translocated to the nucleus during S phase, which is supported by the immunostaining of MIF localization at different phases in cancer cells (Fig. 4a, b).
- 3) PARP1 and MIF interacts with each other and forms a complex prior to binding to the damage sites, which is supported by GST-pulldown (supplementary Fig. 5f), co-immunoprecipitation in cells (supplementary Fig. 5a,g,h), mass spectrometry analysis (Fig. 5a-c, e-f, j,i) and PARylation that is produced after binding to DNA damage sites reduced the recruitment of MIF-PARP1 complex to the replication sites (Fig. 5j-l).
- 4) MIF is located at DNA replication sites, which is strongly supported by multiple approaches including iPOND, co-immunoprecipitation and immune-colocalization with PCNA and Edu (Fig. 4). (Our iPOND experiment does not support the reviewer’s statement that “localization of MIF to the replication sites is no longer supported by iPOND experiments”. Detailed explanation please see Response to Point #4).
- 5) MIF facilitates DNA replication process, DNA synthesis and tumor growth, whereas MIF KO or its nuclease deficiency increases DNA replication stalling, DNA damage, and genome instability, which is not only supported by *in vitro* nuclease/elongation/gap-filling assays (Fig. 1 and Fig.2), but also substantial *in vivo* approaches, including classic DNA fiber assays prepared from MIF WT, KO, KO-MIF (rescue cells) and KO-E22A cancer cells treated with or without hydroxyurea (Fig. 3f-h and supplementary Fig. 3h-k), BrdU-labeled DNA synthesis assay in MIF WT, MIF KO, KO-MIF (rescue

cells) and KO-E22A cancer cells, HPRT mutation assays with MIF and MIF KO in polymerase-proficient cancer cells as well as polymerase-deficient cancer cells (Fig. 3i-l), DNA damage and micronuclei assays in cancer cells, mitosis with chromatin bridges and metaphase spread assays in MIF WT, MIF KO, KO-MIF (rescue cells) and KO-E22A (Fig.6), cancer cell colony formation assays with MIF WT, MIF KO, KO-MIF (rescue cells) and KO-E22A and their tumor growth in mouse (Fig. 7, supplementary Fig. 7).

Regarding the concerns whether PARP1 can recognize nucleotide misincorporation in vivo and whether nucleotide misincorporation caused excessive replication fork stalling in MIF KO cells, we first need to clarify that MIF cleaves 3' flap, which can be caused by nucleotide misincorporation but not limited to nucleotide misincorporation, although we used misincorporated nucleotides to create 3' flap structure as an in vitro model. We apologize that we did not make our statement clear enough. Therefore, in this revision, we have revised "nucleotide misincorporation" to "3' flap structure" to make it accurate and pointed out that 3' flap structure can be induced by different factors, including but not limited to nucleotide misincorporation, ribonucleotide misincorporation, DNA damage or DNA second structures. Second, whether PARP1 can recognize 3' flap structure/misincorporation in vivo and whether 3' flap structure/nucleotide misincorporation caused excessive replication fork stalling observed in MIF KO cells should not be a concern. PARP1 is a well-defined DNA damage sensor and recognizes DNA damage, including single base modifications like addition of alkyl groups, nucleotide damage and unligated Okazaki fragments (Chaudhuri AR et al., 2017, Hanzlikova H et al., 2018). PARP1 detects 3' flap structures and is activated to produce PAR, which is directly proved by our in vitro PARP1 activity assay (Fig. 5d and Supplemental Fig. 5d-e) and also detected at DNA replication fork in vivo by another recent study (Hanzlikova H et al., 2018). On the other hand, as the reviewer agreed, our HRPT mutation analysis provided evidence that MIF KO increased misincorporated mutations in vivo and DNA fiber assays provided direct evidence that excessive DNA replication stalling was indeed observed in MIF KO cells. Although currently no mature or technically feasible assay is available in the field, which can allow us to directly model 3' flap structure/misincorporation in vivo and to test if it causes replication stalling in MIF KO cells, it is no doubt that unresolved 3' flap structure will cause DNA replication stress, as 3' flap structures or Y shape DNA structures represent the intermediate status of DNA replication and occur during DNA replication, which can be caused by misincorporation of dNTP, rNTP, disruption of the balance between purine and pyrimidine, DNA damage or DNA second structure and is already known to cause replication stress and fork stalling (Zeman MK and Cimprich KA, 2014; Techer H et al., 2017; Ubhi T and Brown GW, 2019; Zeman MK and Cimprich KA, 2014; Techer H et al., 2017). As for how 3' flap structure/Y shape structure is formed in vivo and how it causes replication fork stalling is beyond the scope of the current study.

Taken together, in this revised manuscript, we have corrected our description regarding "nucleotide misincorporation", revised our title and clarified our model to make it well supported by our data. We have clarified our model in detail on page 26 as following:

Taken together, our findings support a working model that PARP1 interacts with MIF during S phase and forms a complex prior to binding to the damage sites. PARP1 as the DNA damage sensor detects DNA damage including 3' flap structures during replication. Then MIF-PARP1 complex is recruited to the damage sites at the replication fork to excise the 3' flap structures, facilitating DNA synthesis and promoting cancer cell growth (Fig. 7o).

Major points

3. The authors provided no data to support that the excessive replication fork stalling in MIF KO cells is caused by nucleotide misincorporation. They responded to the critique by citing the increased mutation frequency in MIF KO cells, but increased mutations do not necessarily mean replication forks were stalled when misincorporations occurred.

Response: Regarding the concern whether nucleotide misincorporation caused excessive replication fork stalling in MIF KO cells, we have summarized the current biggest challenge that lacks mature tool in the field to directly model 3' flap structure/misincorporation in vivo in Response to Point #2 and pointed out that it is no doubt that unresolved 3' flap structure/misincorporation will cause DNA replication stress, which will not affect us to draw our main conclusion. Please see the detailed explanation as following:

The key finding of our manuscript is that MIF recognizes and cleaves the unpaired 3' flap on the Y-shaped DNA, which mimics the intermediate status of DNA replication, to facilitate DNA elongation and cancer cell growth. This conclusion was strongly supported by our in vitro nuclease assay (Fig. 1), elongation assay (Fig. 2a-d), gap-filling assay (Fig.2e-h), BrdU-labeled DNA synthesis assay, DNA fiber assay, and cell cycle analysis (Fig. 3 and Fig. 4a-c), colocalization and co-immunoprecipitation studies of MIF-PARP1-PCNA complex at DNA replication site (Fig. 4d-i), as well as effects of MIF and its nuclease activity on cancer cell colony formation and tumor growth in mouse (Fig. 7). Although we used misincorporated nucleotides to create 3' flap structure as an in vitro model, in vivo 3' flap structure, which can be produced not only by misincorporation of nucleotide, but also other factors like misincorporation of ribonucleotide, disruption of the balance between purine and pyrimidine, DNA damage or DNA secondary structure, is already known to cause replication stress and fork stalling (Zeman MK and Cimprich KA, 2014; Techer H et al., 2017; Ubhi T and Brown GW, 2019). DNA fiber assay is one of the best assays to visualize the replication fork stalling in the cell in vivo. Our DNA fiber assay clearly showed that MIF knockout increased the replication fork stalling, which can be reversed by the expression of MIF, but not the nuclease-deficient mutant (Fig. 3f-h), suggesting that loss of MIF nuclease activity contributes to the replication fork stalling in vivo. Moreover, as the reviewer suggested, we also performed HPRT mutation analysis in the revision to further confirm that MIF knockout increased mutation rate especially the nucleotide misincorporation rate in vivo (Fig. 3i-l). So far, there is no other mature or technically feasible assay in the field, which can allow us to directly model 3' flap structure/mis-incorporation and to study if mis-incorporation causes the excessive replication stalling in MIF KO cells. On the other hand, how 3' flap structure/Y shape structure is formed in vivo either induced by misincorporation or others and how 3' flap structure/misincorporation causes replication fork stalling is beyond the scope of the current study. Nevertheless, the results presented in the manuscript strongly support our conclusion that MIF and its nuclease activity facilitate DNA replication and cancer cell growth as MIF KO causes DNA replication fork stalling, increases DNA mutations and inhibits tumor growth. In order to make our statement more precise, we have rephrased “nucleotide misincorporation” to “3' flap structure” throughout the revised manuscript.

4. The new thymidine chase experiments in iPOND failed to provide evidence for MIF localization at replication forks. MIF proteins were detected in the EdU pull-down even after thymidine chase. This data should not be interpreted to indicate that “MIF is [...] possibly involved in some post-replicative or damage repair-related events” (Page 14, line 14) without additional evidence because it is possible that MIF is ubiquitously distributed on chromatin.

Response: Our iPOND experiments showed that MIF was indeed associated with newly synthesized DNA (Fig. 4d) and we also confirmed that MIF was present at the replication forks by co-immunoprecipitation and immunofluorescence colocalization with PCNA and EdU (Fig. 4e-I and Fig. 5e-l). Further we found that MIF KO and loss of MIF nuclease activity inhibited DNA synthesis, delayed cell cycle and decreased

replication speed (Figs. 3 and 4). These data clearly indicate that MIF locates at replication sites and promotes replication fork progression. Although MIF persistently bound to newly replicated chromatin after thymidine chase, these results cannot deny its location and functions at replication sites. Instead, our results indicate that MIF's nuclease activity may be also involved in other events besides its role in replication. Therefore, we cannot agree with the reviewer that "The new thymidine chase experiments in iPOND failed to provide evidence for MIF localization at replication forks." It would be arbitrary to exclude MIF location and functions at replication sites based on these thymidine chase data, which actually can be supported by another work published on *Nature* (Coquel F. et al., 2018). By using the iPOND method, Coquel F et al. showed that SAMHD1, which is similar to MIF present at replication forks and persisted on newly replicated chromatin after a thymidine chase (Fig. 2b and Extended Data Fig. 2b), acts at stalled replication forks to promote normal fork progression. On the other hand, we have removed this statement "MIF is [...] possibly involved in some post-replicative or damage repair-related events" (Page 14, line 14) from the revised manuscript to make it well supported by our data.

5. The molecular events at the misincorporated nucleotides are still defined poorly. The authors appear to propose that PARylation of PARP1 promotes dissociation of MIF from PARP1, but what brings the released MIF to PCNA is completely absent in the proposed model.

Response: PARP1 is a DNA damage sensor and detects 3' flap structures, which is directly supported by our PARP1 activity assay in response to 3' flap structures (Fig. 5d and Supplemental Fig. 5d-e). On the other hand, as explained in Response to Point #2 and #3, we have revised "misincorporated nucleotides" to "3' flap structures" regardless it is caused by misincorporation or others. MIF and PARP1 interact with each other and form a complex prior to PARP1 detection of damaged DNA. Once PARP1 recognizes damaged DNA, it brings the PARP1-MIF complex to replication sites, where PCNA is localized. PCNA is a DNA clamp that acts as a processivity factor for DNA Pol δ and also acts as a scaffold to recruit proteins involved in DNA replication, DNA repair, chromatin remodeling, and epigenetics (Moldovan GL et al., 2007. *Cell*). Previous studies have shown that PARP1 dynamically interacts with PCNA (Prosperi Ennio and Scovssi Ivana, *Landes Bioscience*, 2006; Frouin Isabelle et al., 2003; Prosperi E. et al., 2013). Our study showed that knockout of PARP1 inhibits MIF-PCNA interaction and blocks MIF recruitment to sites of DNA damage (Fig. 5e-5i). Activation of PARP1 subsequently promotes dissociation of MIF from PARP1, which is supported by the finding that PARP inhibitor enhances PARP1-MIF-PCNA interaction and colocalization (Fig. 5h-l). Therefore, we have provided substantial data to support that PARP1 recruits MIF to PCNA in the revised manuscript and PARylation terminates this process. In this revised manuscript, we have explained our model in detail on page 26 as following:

PARylation of PARP1 disrupts the PARP1-MIF interaction and decreases the amount of MIF protein at the replication sites, which is supported by our data that PARP inhibitor treatment increases MIF-PCNA interaction and colocalization in the replication sites.

Taken together, our findings support a working model that PARP1 interacts with MIF during S phase and forms a complex prior to binding to the damage sites. PARP1 as the DNA damage sensor detects DNA damage including 3' flap structures during replication. Then MIF-PARP1 complex is recruited to the damage sites at the replication fork to excise the 3' flap structures, facilitating DNA synthesis and promoting cancer cell growth (Fig. 7o).

6. Tumor specific replication stress that makes MIF important for cell survival is undefined. The authors' response and revised Discussion contain too much speculation.

Response: The main focus of this manuscript is to study if MIF has a 3' flap nuclease activity and the effects of MIF's nuclease activity on DNA replication and cell growth in cancer cells, since MIF expression is upregulated in human cancers. During the 1st round review, reviewer raised a question about MIF's effects on tumor vs. normal cells. Although this question is beyond the topic of our current study, we still tried to address it by investigating the effect of MIF on growth and survival of cancer cells vs. nontumorigenic epithelial MCF-10A cells and mouse embryonic fibroblasts in the last revision. We agree with the reviewer that it might be premature to conclude that "*These data indicated that MIF is more vulnerable for growth of cancer cells than normal cells*" since we did not perform extensive studies in normal cells and normal tissues in MIF WT and KO mouse models, which could be another independent comprehensive study and requires tremendous time and efforts. Nevertheless, we have provided substantial evidence to support our conclusion in this manuscript that MIF is a novel 3' flap nuclease and promotes DNA elongation and cell growth in cancer cells. MIF knockout significantly inhibited cancer cell growth and colony formation in both Pol δ -proficient and -deficient cancer cells (Supplementary Fig. 3k-o). Considering the fact that studying the effect of MIF in normal cells is beyond the scope of the current study, we revised our statement to precisely interpret our data in this revision and also pointed out that additional comprehensive studies are required to address this question in the future. The revised conclusion is shown as following:

Additional studies are required to explore if MIF is more vulnerable for growth of cancer cells than normal cells in future.

Minor point:

1) Concerns remain whether in vitro extension assays were performed under an optimal condition. The stoichiometry of Pol delta subunits seems to be off as POLD3 and POLD4 appear to be overrepresented.

Response: Pol δ was purified as a complex. We performed multiple times of Pol δ protein purification in our studies and got highly consistent results. The image shown in the manuscript represents one batch of the purified Pol δ complex. The intensity of Coomassie staining cannot be simply equal to the protein concentration due to the fact that the dye binds to proteins primarily through basic amino acids (arginine/R, lysine/K and histidine/H). The enrichment of R/K/H in POLD4 and POLD3 is about two times higher than those in POLD2 and POLD1.

REVIEWERS' COMMENTS

Reviewer #2 (Remarks to the Author):

The authors have now made great efforts to investigate the role of the MIF protein and have carried out a number of important controls and verifications. The findings described in the manuscript are novel and exciting. Although it is possible to suggest alternatives to some of the details of the models proposed by the authors it would be useful if this manuscript now received a more general audience and scrutiny. The manuscript certainly makes some interesting proposals.

Reviewer #3 (Remarks to the Author):

While the authors addressed some of the previous concerns about speculative statements, the major concerns about in vivo evidence for the role of misincorporated nucleotides (now they call it 3' flap structure) still remain. In their responses, the authors listed in vivo experiments such as (1) colocalization of MIF with replication sites, (2) localization of PARP1 to replication forks (by other group), (3) increased mutagenesis and excessive replication fork stalling in MIF KO cells. These experiments were performed nicely but it does mean that 3' flap structure is the cause of these observations. Thus, I do not believe that the authors have provided enough evidence to support their claim.

Response to Reviewers

Referees comments are highlighted in blue and the response is provided below the referee comments.

Reviewer #2 :

1. The authors have now made great efforts to investigate the role of the MIF protein and have carried out a number of important controls and verifications. The findings described in the manuscript are novel and exciting. Although it is possible to suggest alternatives to some of the details of the models proposed by the authors it would be useful if this manuscript now received a more general audience and scrutiny. The manuscript certainly makes some interesting proposals.

Response: We thank the referee for the positive comments, including “the authors have made great efforts to investigate the role of the MIF protein and have carried out a number of important controls and verifications.” and “The findings described in the manuscript are novel and exciting.”

Reviewer #3

While the authors addressed some of the previous concerns about speculative statements, the major concerns about in vivo evidence for the role of misincorporated nucleotides (now they call it 3' flap structure) still remain. In their responses, the authors listed in vivo experiments such as (1) colocalization of MIF with replication sites, (2) localization of PARP1 to replication forks (by other group), (3) increased mutagenesis and excessive replication fork stalling in MIF KO cells. These experiments were performed nicely but it does not mean that 3' flap structure is the cause of these observations. Thus, I do not believe that the authors have provided enough evidence to support their claim.

Response: To address the reviewer's concern, we have revised the manuscript in order to make sure that our in vitro and in vivo conclusions are strongly based on the extensive experiments. Meanwhile, we have also toned down our in vivo conclusion/model in the revised manuscript by stating it as “a possible model” or “possible mechanism” and pointed out that future in vivo studies are required to further support this model. The new changes were made on Page 2-Abstract, Page 5-Introduction and Page 27-Discussion parts, and have also been included here as following:

Here, we show that macrophage migration inhibitory factor (MIF) is a 3' flap nuclease and translocated to the nucleus in S phase. Poly(ADP-ribose) polymerase 1 recruits MIF to the DNA replication fork, where MIF nuclease activity is required to resolve replication stress and facilitates tumor growth. (Page 2)

These findings uncover a possible mechanism of MIF-mediated tumor growth by removing the 3' flap structures during DNA replication. (Page 5)

Taken together, our findings support a possible working model that PARP1 interacts with MIF during S phase and forms a complex prior to binding to the damage sites. PARP1 as the DNA damage sensor detects DNA damage including 3' flap structures during replication. Then MIF-PARP1 complex is recruited to the damage sites at the replication fork to resolve the replication stress, facilitating DNA synthesis and promoting cancer cell growth (Fig. 7o). Future in vivo studies are required to further support the proposed model. (Page 27)